# Label Correction of Crowdsourced Noisy Annotations with an Instance-Dependent Noise Transition Model

**Hui Guo**
Department of Computer Science
University of Western Ontario
hguo288@uwo.ca

**Boyu Wang** *
Department of Computer Science
University of Western Ontario
bwang@csd.uwo.ca

**Grace Y. Yi** *
Department of Statistical and Actuarial Sciences
Department of Computer Science
University of Western Ontario
gyi5@uwo.ca

## Abstract

The predictive ability of supervised learning algorithms hinges on the quality of annotated examples, whose labels often come from multiple crowdsourced annotators with diverse expertise. To aggregate noisy crowdsourced annotations, many existing methods employ an annotator-specific instance-independent *noise transition matrix* to characterize the labeling skills of each annotator. Learning an *instance-dependent* noise transition model, however, is challenging and remains relatively less explored. To address this problem, in this paper, we formulate the noise transition model in a Bayesian framework and subsequently design a new label correction algorithm. Specifically, we approximate the instance-dependent noise transition matrices using a Bayesian network with a hierarchical spike and slab prior. To theoretically characterize the distance between the noise transition model and the true instance-dependent noise transition matrix, we provide a posterior-concentration theorem that ensures the posterior consistency in terms of the Hellinger distance. We further formulate the label correction process as a hypothesis testing problem and propose a novel algorithm to infer the true label from the noisy annotations based on the pairwise likelihood ratio test. Moreover, we establish an information-theoretic bound on the Bayes error for the proposed method. We validate the effectiveness of our approach through experiments on benchmark and real-world datasets.

## 1 Introduction

Deep neural networks (DNNs) have achieved remarkable performance in various tasks [1, 2], and they have proven to be useful in handling sizable labeled data. Acquiring large accurately annotated datasets, however, is usually expensive and time consuming. To enhance the efficiency of annotation, in many applications, *crowdsourcing* [3] is employed as an alternative way for data labeling, where the labels are provided by multiple annotators with varying and imperfect labeling skills, and thus, the collected labels suffer from unavoidable noise. As deep models have a strong memorization power, using these noisy labels as the ground truth deteriorates the performance of DNNs [4, 5], and most importantly, yields erroneous learning results. Further, potentially substantial disagreement among

---

*Corresponding authors.

37th Conference on Neural Information Processing Systems (NeurIPS 2023).

the annotators for each instance presents extra challenges in the application of traditional supervised learning algorithms. Hence, in the crowdsourcing scenario, to effectively train DNNs on noisy labeled datasets, a fundamental question is how to aggregate the noisy crowdsourced annotations and infer the latent true labels [6].

One naive approach to aggregate the crowdsourced labels is simply by computing the majority vote, which can be ineffective when the number of annotators is not large enough or the labeling task is difficult [7, 8]. Recent research has developed more powerful techniques for inferring the ground truth labels [7, 9–11], among which the *annotator-specific noise transition matrix*, aka *annotator confusion*, plays an important role by modeling the labeling process for each individual annotator. To estimate the transition matrix, available research [8, 11–13] usually makes the *instance-independent* assumption that for annotator $r$, given the true label $\mathrm{y}$, the corruption process is independent of the input $\mathbf{x}$, i.e., $\mathbb{P}(\tilde{\mathrm{y}}^{(r)} = l | \mathrm{y} = k, \mathbf{x}) = \mathbb{P}(\tilde{\mathrm{y}}^{(r)} = l | \mathrm{y} = k)$, where $\mathbf{x}$ denotes the random variable for instance/feature, $\tilde{\mathrm{y}}^{(r)}$ represents the noisy label given by annotator $r$, and $\mathrm{y}$ is the underlying ground truth label. This assumption, however, is often violated in applications. *Instance-dependent* annotation noise is more realistic and appropriate for real-world datasets, as suggested by the example that factors such as the quality of ultrasound images and the domain expertise of human annotators can greatly influence the actual diagnostic process in medical analysis [14, 15]. For annotator $r$, the transition matrix $\tau^{(r)}(\mathbf{x})$ is a matrix-valued function, with the $(k, l)$ element defined as $\tau_{kl}^{(r)}(\mathbf{x}) = \mathbb{P}(\tilde{\mathrm{y}}^{(r)} = l | \mathrm{y} = k, \mathbf{x})$. Unfortunately, the case of instance-dependent annotation noise remains challenging and less explored. Most existing works considering instance-dependent noise are designed for the single annotator case [16–18]. For the case with multiple annotators, existing methods investigate the human annotation process and use different models to estimate instance-dependent noise matrices. Approaches in [3, 19–22] use traditional classification models such as logistic regression, while others [23–25] cater to large datasets and deep models. Methods in [3, 19–22] and [24, 25] are heuristic in nature and lack theoretical guarantees in estimating instance-dependent noise matrices. [23] makes some theoretical progress in justifying the use of the trace regularisation, and extends the work of [8] which establishes the theory only for settings with an instance-independent noise matrix. The theory in [23] is constrained to individual samples rather than the population setting. Importantly, the theoretical characterization of the distance of the noise model and the true annotator confusion remains absent from the literature.

In this paper, we address this notable problem by framing it within the Bayesian paradigm. We **formulate the instance-dependent annotator-specific label transition matrix**, and further propose **a novel algorithm to infer the underlying ground truth by aggregating the noisy annotations**. To model the noise transition matrix, we invoke the Bayesian generalized linear mixed effects model (GLMM), which can be learned by deploying anchor points within the deep learning framework [12, 26, 27]. To facilitate the fact that the number of anchor points learned from the noisy training data is relatively small compared to the sample size, we employ a hierarchical spike and slab prior on the network parameters. This approach offers an interpretable mechanism for variable selection and allows us to establish the theoretical result within the deep learning setup. Our study reveals that the proposed noise transition model is close to the underlying true transition matrix with respect to the Hellinger distance in the Bayesian framework. Such a result is established for independently, nonidentically distributed (i.n.i.d.) observations, substantially extending the existing sparse Bayesian theories within the deep learning paradigm. Further, we develop a label correction method using the pairwise likelihood ratio test to aggregate and infer the ground truth from the noisy crowdsourced annotations. This development is carried out by formulating the label correction process as a hypothesis testing problem and utilizing the proposed Bayesian model in place of the unknown transition matrix in the pairwise likelihood ratio test (LRT). More importantly, with the posterior consistency result, we also derive information-theoretic bounds on the Bayes error for the proposed algorithm even without access to the underlying true noise transition matrix.

This research brings forth several noteworthy advancements: (1) We formulate the annotator-specific noise transition matrix in the Bayesian framework (Section 3.1). This method offers a practical and flexible framework to address real-world problems with noisy annotators. (2) We theoretically characterize the closeness of the proposed model and the underlying annotator confusions with respect to the Hellinger distance. (Section 3.2). (3) We develop a novel label correction algorithm by aggregating the noisy annotations using the pairwise likelihood ratio test, and identify information-theoretic bounds on the Bayes error (Section 3.3). The effectiveness of the proposed algorithm is

confirmed by the application to both synthetic and real-world noisy datasets (Section 5). Code is available at https://github.com/hguo1728/BayesianIDNT.

## 2 Problem Setup

**Objective and Data.** Consider a classification task with a feature space $\mathcal{X} \subset \mathbb{R}^p$ and a label space $\mathcal{Y} = [K]$, where $p$ is the dimension of the features, $K$ is the number of classes, and $[k]$ represents $\{1, ..., k\}$ for any positive integer $k$. Our goal is to develop a classifier $h : \mathcal{X} \mapsto \mathcal{Y}$, which can accurately predict the true label for a test instance. However, in applications, the true label $y \in \mathcal{Y}$ is often not observed for each input vector $\mathbf{x} \in \mathcal{X}$. Instead, we receive a set of noisy crowdsourced labels $\tilde{\mathbf{y}} = \{\tilde{y}^{(1)}, .., \tilde{y}^{(R)}\}$ from $R$ distinct annotators, where $\tilde{y}^{(r)} \in \mathcal{Y}$ represents the label given by the $r$th annotator for $r \in [R]$. Thus, a noisy dataset $\mathcal{D}$ of size $N$ is defined as $\mathcal{D} = \{\mathbf{x}_i, \tilde{y}_i^{(1)}, .., \tilde{y}_i^{(R)}\}_{i=1}^N$, where for each instance $\mathbf{x}_i$, the true label $y_i$ is unobserved. Under this setting, we aim to learn a reliable classifier $h$ by utilizing the noisy crowdsourced dataset $\mathcal{D}$.

In practice, on commercial crowdsourcing platforms, large-scale labels can often be collected from independent human annotators. We thereby make a common assumption that the $R$ annotators independently label the instances [7, 8]. The conditional probability of the $R$ noisy labels, given an instance, can then be formulated as

$$\mathbb{P}(\tilde{y}^{(1)}, .., \tilde{y}^{(R)}|\mathbf{x}) = \prod_{r=1}^R \mathbb{P}(\tilde{y}^{(r)}|\mathbf{x}) = \prod_{r=1}^R \sum_{k \in \mathcal{Y}} \left\{ \mathbb{P}(\tilde{y}^{(r)}|y = k, \mathbf{x}) P(y = k|\mathbf{x}) \right\}, \qquad (1)$$

where for $k \in \mathcal{Y}$, $\mathbb{P}(y = k|\mathbf{x})$, called the *base model* [28], denotes the conditional probability of the latent true label $y$ given $\mathbf{x}$, which can be modeled by the output of a DNN parameterized by a parameter vector, say $\boldsymbol{\vartheta}$; and $\mathbb{P}(\tilde{y}^{(r)}|y = k, \mathbf{x})$ is the *noise transition model* for the $r$th annotator [8], satisfying $\sum_{l=1}^K \mathbb{P}(\tilde{y}^{(r)} = l|y = k, \mathbf{x}) = 1$ for any $\mathbf{x} \in \mathcal{X}$ and $k \in [K]$. For ease of theoretical presentation, we assume the accessibility to all the annotations from the $R$ workers for now, and consider more general situations in the experimental part in Section 5; extensions to accommodating the case where each instance is only annotated by a subset of annotators are straightforward.

**Notation.** In this paper, sets are denoted by calligraphic upper case letters, and vectors and matrices are denoted by bold lower and upper case letters, respectively. For a vector $\boldsymbol{v}$, $v_j$ denote its $j$th element, and $\boldsymbol{v}^\top$ denotes its transpose. For $\boldsymbol{v} = (v_1, ..., v_d)^\top$, we denote $\|\boldsymbol{v}\|_q = (\sum_{j=1}^d |v_j|^q)^{1/q}$ for $q > 0$, $\|\boldsymbol{v}\|_\infty = \max_j |v_j|$, and $\|\boldsymbol{v}\|_0 = \sum_{j=1}^d \mathbf{1}(v_j \neq 0)$, with $\mathbf{1}(\cdot)$ denoting the indicator function. The $L_2$ norm of $\boldsymbol{v}$ is also denoted by $\|\boldsymbol{v}\|$ for simplicity. For a matrix $\boldsymbol{V}$, we use $V_{i,j}$ to represent its $(i, j)$ element. Let $(\Omega, \mathcal{G}, \mu)$ denote the measure space under consideration, where $\Omega$ is a set, $\mathcal{G}$ is the $\sigma$-field of subsets of $\Omega$, and $\mu$ is the associated measure. For a measurable function $f : \Omega \to \mathbb{R}^d$, we write $\|f\|_q \triangleq \|f\|_{L^q(\Omega)}$ when there is no ambiguity of the domain, where $\|f\|_{L^q(\Omega)} = \left( \int_\Omega \sum_{j=1}^d |f_j(x)|^q d\mu \right)^{1/q}$ for $q > 0$. For two sequences, $\{a_n\}$ and $\{b_n\}$, we write $a_n \preceq b_n$ if there exists a positive constant $C$ such that $a_n \leq Cb_n$ for large enough $n$, and we write $a_n \asymp b_n$ if $a_n \preceq b_n$ and $b_n \preceq a_n$.

## 3 Main Results

### 3.1 Instance-dependent transition matrix with multiple annotators

**Annotator-specific instance-dependent noise transition model.** Given an instance $\mathbf{x}$, the conditional probability mass function of noisy annotations can be characterized by $R$ instance-dependent matrices of dimension $K \times K$, termed *transition matrices* or *annotator confusions* [8, 13], with the $k$th row of the $r$th matrix denoted $\left( \mathbb{P}(\tilde{y}^{(r)} = 1|y = k, \mathbf{x}), \ldots, \mathbb{P}(\tilde{y}^{(r)} = K|y = k, \mathbf{x}) \right)$. Thus, the distribution of noisy annotation depends on the instance in different ways due to the differences in the annotator $r$ and the underlying true label $y$, which can be characterized by a Bayesian generalized linear mixed effects model (GLMM) [29, 30] in the deep learning framework.

Specifically, conditioned on the true label $\mathrm{y} = k$ and the feature vector $\mathbf{x}$, we treat the noisy label $\tilde{\mathrm{y}}^{(r)}$ from annotator $r$ as a random variable generated from the distribution:

$$\tilde{\mathrm{y}}^{(r)}|\{\mathrm{y} = k, \mathbf{x}\} \sim \mathrm{Cat}(\boldsymbol{s}^{(k,r)}), \tag{2}$$

where $\boldsymbol{s}^{(k,r)} = (s_1^{(k,r)}, ..., s_K^{(k,r)})^\top \in \mathcal{S}^{K-1}$ with $\mathcal{S}^{K-1} = \{(s_1, ..., s_K)^\top \in \mathbb{R}^K : s_j \geq 0 \text{ for } j \in [K] \text{ and } \sum_{j=1}^K s_j = 1\}$ representing the $(K-1)$-dimensional simplex, and $\mathrm{Cat}(\boldsymbol{s}^{(k,r)})$ represents a categorical distribution specified by the parameter vector $\boldsymbol{s}^{(k,r)}$. We extend existing works on mixed effects neural networks (MNN) [31, 32] by employing two nonlinear transformations $\psi_1$ and $\psi_2$ to incorporate different effects in the instance-dependent noise transition model, and set

$$\boldsymbol{s}^{(k,r)} = G(\boldsymbol{\omega}_0^{(k,r)}) \text{ with } \boldsymbol{\omega}_0^{(k,r)} = \mathbf{A}_0^{(r)} \psi_1(\mathbf{x}) + \mathbf{B}_0^{(k)} \psi_2(\mathbf{x}), \tag{3}$$

where $\mathbf{A}_0^{(r)} = (\boldsymbol{\alpha}_{10}^{(r)}, ..., \boldsymbol{\alpha}_{K0}^{(r)})^\top$ and $\mathbf{B}_0^{(k)} = (\boldsymbol{\beta}_{10}^{(k)}, ..., \boldsymbol{\beta}_{K0}^{(k)})^\top$ are the regression weights; $\psi_1(\mathbf{x})$ and $\psi_2(\mathbf{x})$ can be modeled by some suitable networks; and $G$ is a function mapping $\mathbb{R}^K$ to $\mathcal{S}^{K-1}$, which, in practice, is chosen to be the softmax function in the final layer. Utilizing two different network components $\psi_1$ and $\psi_2$ enables us to flexibly reflect possibly different effects of the annotator expertise ($r$) and the ground truth ($k$) in the annotation process, which can be interpreted as the input in mixed effects models.

**Approximating the transition matrices.** The proposed instance-dependent noise transition model can be learned by leveraging anchor points [12, 26, 27], or instances that are similar to anchor points learned from noisy training data [33]. An instance $\mathbf{x}$ is defined to be an anchor point of class $k$ if it belongs to the $k$th class almost surely, that is, $\mathbb{P}(\mathrm{y} = k|\mathbf{x}) = 1$, and hence, $\mathbb{P}(\tilde{\mathrm{y}}^{(r)}|\mathbf{x}) = \mathbb{P}(\tilde{\mathrm{y}}^{(r)}|\mathrm{y} = k, \mathbf{x})$. For $k \in [K]$, let $\overline{\mathcal{D}}_{0,k}$ be the set of anchor points of the $k$th class and the associated noisy annotations, i.e., $\overline{\mathcal{D}}_{0,k} = \{\{\mathbf{x}_i, \tilde{\mathbf{y}}_i\} : \mathbb{P}(\mathrm{y}_i = k|\mathbf{x}_i) = 1\}$. Define $\overline{\mathcal{D}}_0 = \overline{\mathcal{D}}_{0,1} \cup \overline{\mathcal{D}}_{0,2} \cup \ldots \cup \overline{\mathcal{D}}_{0,K}$, and let $n$ denote the subsample size of the learned anchor points, i.e., the cardinality of $\overline{\mathcal{D}}_0$. Paired variables $\{\mathbf{x}_i, \tilde{\mathbf{y}}_i\}$ in $\overline{\mathcal{D}}_0$ are independent, but **not necessarily identically** distributed (i.n.i.d). We write the input dimension $p$ as $p_n$ from now on to emphasize that its dependence on $n$ is allowed.

In applications, overfitting can occur when the subsample size $n$ of the learned anchor points is relatively small compared to the main sample size $N$. To address this issue, we propose to learn $\psi_j(\mathbf{x})$ with a sparse Bayesian DNN, denoted $\psi_j(\mathbf{x}; \boldsymbol{\theta}^{(j)})$, where $\boldsymbol{\theta}^{(j)}$ represents the vector of all involved parameters in the network with $j = 1, 2$. Furthermore, invoking the sparse Bayesian setting allows us to rigorously characterize the distance between the proposed model and the underlying true transition matrices, as presented in Theorem 1 of Section 3.2.

## 3.2 Bayesian analysis and posterior consistency result

**Prior specification.** To implement sparse Bayesian analysis, we utilize the spike and slab prior [34] on the network parameters, offering an interpretable mechanism for variable selection. The spike and slab model is formulated by constructing a prior hierarchy of the involved parameters and selects nonzero coefficients according to the posterior inclusion probability. Marginally, these priors are mutually independent and have a mixture distribution consisting of a flat distribution (slab) and a distribution concentrated at zero (spike). Parameters with a small posterior mean will be set to zero to achieve sparsity.

Specifically, for network $\psi_j(\mathbf{x}; \boldsymbol{\theta}^{(j)})$, we write $\boldsymbol{\theta}^{(j)}$ as $\boldsymbol{\theta}^{(j)} = (\theta_1^{(j)}, ..., \theta_{J_j}^{(j)})^\top$ with $J_j$ denoting the length of $\boldsymbol{\theta}^{(j)}$ for $j = 1, 2$. For $k \in [J_j]$, we treat $\theta_k^{(j)}$ as a random variable generated from the following prior hierarchy:

$$\gamma_k^{(j)} \sim \mathrm{Bernoulli}(\lambda_{nj}), \tag{4a}$$

$$\theta_k^{(j)}|\gamma_k^{(j)} \sim \gamma_k^{(j)}\pi_1(\theta_k^{(j)}; \sigma_{nj}^2) + (1 - \gamma_k^{(j)})\pi_0(\theta_k^{(j)}; c_{nj}\sigma_{nj}^2), \tag{4b}$$

where $\gamma_k^{(j)} \in \{0, 1\}$ indicates whether or not $\theta_k^{(j)}$ is nonzero, $c_{nj}$ is specified as a very small positive number, $\sigma_{nj}^2$ and $c_{nj}\sigma_{nj}^2$ are the parameters related to the variances of distributions $\pi_1(\cdot)$ and $\pi_0(\cdot)$, respectively, and $\lambda_{nj} \in (0, 1)$ determines the ratio of the mixture distribution. As $c_{nj} \to 0$,

$\pi_0(\theta_k^{(j)}; c_{nj}\sigma_{nj}^2)$ becomes the degenerate distribution at zero. The marginal distribution of $\theta_k^{(j)}$ is then determined by

$$\theta_k^{(j)} \sim \lambda_{nj}\pi_1(\theta_k^{(j)}; \sigma_{nj}^2) + (1 - \lambda_{nj})\pi_0(\theta_k^{(j)}; c_{nj}\sigma_{nj}^2), \tag{5}$$

which is presented as $\pi^{(j)}(\cdot)$ for short; and this is taken as the prior distribution of $\theta_k^{(j)}$.

To further incorporate the effects of the true label information and the randomness from different annotators in (3), we place the following probabilistic structure on the generic weights for $\mathbf{A}_0^{(r)}$ and $\mathbf{B}_0^{(k)}$ in (3), $\mathbf{A}^{(r)} = (\boldsymbol{\alpha}_1^{(r)}, ..., \boldsymbol{\alpha}_K^{(r)})$ and $\mathbf{B}^{(k)} = (\boldsymbol{\beta}_1^{(k)}, ..., \boldsymbol{\beta}_K^{(k)})$:

$$\boldsymbol{\alpha}_j^{(r)} \sim \mathcal{N}(\mathbf{0}, \boldsymbol{\Sigma}_\alpha^{(r)}) \text{ and } \boldsymbol{\beta}_j^{(k)} \sim \mathcal{N}(\mathbf{0}, \boldsymbol{\Sigma}_\beta^{(k)}). \tag{6}$$

for $j, k \in [K]$ and $r \in [R]$, where $\boldsymbol{\Sigma}_\alpha^{(r)}$ and $\boldsymbol{\Sigma}_\beta^{(k)}$ are nonnegative definite matrices. We use $\pi_A^{(r)}(\cdot)$ and $\pi_B^{(k)}(\cdot)$ to denote the prior distribution of $\mathbf{A}^{(r)}$ and $\mathbf{B}^{(k)}$ in (6). Here the regression weights $\mathbf{A}^{(r)}$ and $\mathbf{B}^{(k)}$ can be seen as fully connected layers on top of $\psi_1(\mathbf{x}; \boldsymbol{\theta}^{(1)})$ and $\psi_2(\mathbf{x}; \boldsymbol{\theta}^{(2)})$, respectively. The conditions on the aforementioned priors are given in Appendix A.3.

**Prior and posterior probability measure.** Let $\boldsymbol{\theta} = \left(\boldsymbol{\theta}^{(1)\top}, \boldsymbol{\theta}^{(2)\top}, \text{vec}(\mathbf{B}^{(1)})^\top, \ldots, \text{vec}(\mathbf{B}^{(K)})^\top,\right.$ $\left.\text{vec}(\mathbf{A}^{(1)})^\top, \ldots, \text{vec}(\mathbf{A}^{(R)})^\top\right)^\top \in \Theta$ stand for the vector of all involved parameters in the noise transition model, with $\Theta$ denoting the parameter space. We use $\boldsymbol{\theta}_0$ to represent the true value of $\boldsymbol{\theta}$, which is an interior point of $\Theta$. The foregoing specification of the prior distribution places a prior probability measure, denoted $\Pi(\cdot)$, on $\boldsymbol{\theta}$. With the data $\overline{\mathcal{D}}_0$, the posterior probability measure $\Pi(\cdot|\overline{\mathcal{D}}_0)$ is given by

$$\Pi(G|\overline{\mathcal{D}}_0) = \frac{\int_G p_{\boldsymbol{\theta}}^n(\overline{\mathcal{D}}_0)d\Pi(\boldsymbol{\theta})}{\int_\Theta p_{\boldsymbol{\theta}}^n(\overline{\mathcal{D}}_0)d\Pi(\boldsymbol{\theta})} \text{ for any } G \in \mathcal{G}, \tag{7}$$

where $\mathcal{G}$ is the $\sigma$-field on $\Theta$, and $p_{\boldsymbol{\theta}}^n$ is the joint probability density or mass function for the observations in $\overline{\mathcal{D}}_0$ under $\boldsymbol{\theta}$. Let $\mathbb{P}_{\boldsymbol{\theta}}^n(\cdot)$ denote the probability measure associated with $p_{\boldsymbol{\theta}}^n(\cdot)$, and write $\mathbb{P}_0^n(\cdot) \triangleq \mathbb{P}_{\boldsymbol{\theta}_0}^n(\cdot)$. Hence, the data $\overline{\mathcal{D}}_0$ is generated from $\mathbb{P}_0^n(\cdot)$ in our setup.

Let $f$ denote the unknown density of $\mathbf{x}$. For $\{\mathbf{x}, \tilde{\mathbf{y}}\} \in \overline{\mathcal{D}}_{0,k}$, let $f_0^{(k,r)}$ and $f_0^{(k)}$ respectively represent the underlying true distributions for $\tilde{y}^{(r)}$ and $\tilde{\mathbf{y}}$, given $\{y = k, \mathbf{x}\}$, determined by (2) and (3); and let $f_{\boldsymbol{\theta}}^{(k,r)}$ and $f_{\boldsymbol{\theta}}^{(k)}$ denote the corresponding distributions characterized by the model indexed by $\boldsymbol{\theta}$. We let $p_{\boldsymbol{\theta},i} \triangleq f_{\boldsymbol{\theta}}^{(k_i)} f$ denote the probability density or mass function of the $i$th component in $\overline{\mathcal{D}}_0$ under $\boldsymbol{\theta}$, with $k_i \in [K]$ denoting the class that the instance belongs to almost surely. Then, the joint probability density or mass function $p_{\boldsymbol{\theta}}^n$ is calculated as $p_{\boldsymbol{\theta}}^n \triangleq \prod_{i=1}^n p_{\boldsymbol{\theta},i}$. The following theorem describes the closeness of the proposed noise transition model and the true annotator confusions with respect to the Hellinger distance within the Bayesian framework.

**Theorem 1.** *Suppose Conditions A.1-A.4 in Appendix A.2 and B.1-B.3 in Appendix A.3 are satisfied. Let $d(\cdot, \cdot)$ and $d_n(\cdot, \cdot)$ denote the Hellinger distance given in Definition 1 and the semimetric defined in (16) in Appendix B.1, respectively. Then there exists a sequence of constants $\{\epsilon_n^2\}_{n=1}^\infty$ with $\epsilon_n^2 = O(\varpi_{n1} + \varpi_{n2} + \zeta_n)$ and $\log(1/\epsilon_n^2) \prec n\epsilon_n^2$, satisfying $0 < \epsilon_n^2 < 1$, $\epsilon_n \to 0$ and $n\epsilon_n^2 \to \infty$ as $n \to \infty$, such that [2],*

$$\Pi\left\{\boldsymbol{\theta} \in \Theta : d_n(\boldsymbol{\theta}, \boldsymbol{\theta}_0) > M_n\epsilon_n|\overline{\mathcal{D}}_0\right\} \to 0 \tag{8}$$

*in $\mathbb{P}_0^n$ probability for every $M_n \to \infty$, where $\{\varpi_{jn}\}$ is a sequence of nonnegative numbers converging to 0 as $n \to \infty$ for $j = 1, 2$ as given in (14), and $\{\zeta_n\}_{n=1}^\infty$ is a sequence given in Appendix A.3*

---

[2] The anchor point assumption can be relaxed and Theorem 1 can be extended to a more general setting. For any given $\delta \in (0, 1)$, we define the $\delta$-pseudo anchor point for class $k$ as $\mathbb{P}(y = k|\mathbf{x}) \geq 1 - \delta$ and denote $\bar{\mathcal{D}}_\delta$ as the set of all $\delta$-pseudo anchor point accordingly. Then, the following result holds: $\Pi\{\theta \in \Theta : d_n(\theta, \theta_0) > M_n\epsilon_n + C\delta|\bar{\mathcal{D}}_\delta\} \to 0$ in $\mathbb{P}_0^n$ probability for every $M_n \to \infty$, where $C$ is a positive constant. From the modified theory, as $\delta$ approaches 0 slowly, the Hellinger distance of the transition model and the true transition probability converges to zero at a slow rate. In other words, the transition model will still converge even if the collection of a set of anchor points is not guaranteed, albeit at a slow rate.

*depending on the structures of $\psi_1(\cdot|\boldsymbol{\theta}^{(1)})$ and $\psi_2(\cdot|\boldsymbol{\theta}^{(1)})$ with $\zeta_n \to 0$ as $n \to \infty$. If we further assume that $|\overline{\mathcal{D}}_{0,k}|/|\overline{\mathcal{D}}_0| > \varsigma_1$ for some positive constant $\varsigma_1$, with $|\cdot|$ representing the cardinality of a set, then for any $k \in [K]$ and $r \in [R]$*

$$\Pi\left\{\boldsymbol{\theta} \in \Theta : d(f_{\boldsymbol{\theta}}^{(k,r)}, f_0^{(k,r)}) > M_n \epsilon_n | \overline{\mathcal{D}}_0\right\} \to 0, \tag{9}$$

*in $\mathbb{P}_0^n$ probability for any $M_n \to \infty$.*

Intuitively, Theorem 1 reveals that the sparse noise transition model is close to the underlying true transition matrix with respect to the Hellinger distance under mild conditions. Notably, our posterior consistency result extends the existing theories in sparse Bayesian learning [35–37] to the setup of i.n.i.d observations. Moreover, this result on the convergence rate of the posterior measure allows us to infer the underlying true label from the noisy annotations with a theoretical guarantee on the bounds of the Bayes error, which will be discussed in detail in the following section.

### 3.3 Pairwise likelihood ratio test for label correction

The asymptotic result (9) in Theorem 1 indicates that for each annotator, the underlying true instance-dependent transition matrix can be accurately modeled under the Bayesian framework. This enables us to aggregate and infer the ground truth label from noisy crowdsourced annotations.

**A novel label correction algorithm.** To highlight the idea, we first assume that the noise transition matrix $\mathbb{P}(\tilde{\mathrm{y}}^{(r)}|\mathrm{y} = k, \mathbf{x})$, or $f_0^{k,r}(\cdot)$, is known. To simplify the notation, for each $\mathbf{x}_i$ in the noisy dataset $\mathcal{D}$, denote $\tau_{i,kl}^{(r)} \triangleq \tau_{kl}^{(r)}(\mathbf{x}_i) \triangleq \mathbb{P}(\tilde{\mathrm{y}}^{(r)} = l|\mathrm{y} = k, \mathbf{x}_i)$ for $i \in [N]$ and $k, l \in [K]$. We assign class prior [38] $\hbar_i = (\hbar_{i,1}, ..., \hbar_{i,K})^\top$ for the ground truth label for the $i$th task, where the $\hbar_{i,k}$ for $k \in [K]$ are nonnegative weights satisfying $\sum_{k=1}^K \hbar_{i,k} = 1$. For each instance, with the class prior and the noise transition matrices, the label correction process can be formulated as a hypothesis testing problem, where different hypotheses are generated from different choices of the true label values. Specifically, selecting the label for the instance $\mathbf{x}_i$ from $\{g, g'\}$, with $1 \leq g < g' \leq K$, is equivalent to choosing from the two competitors $\mathbb{P}(\tilde{\mathbf{y}}|\mathrm{y} = g, \mathbf{x}_i)$ and $\mathbb{P}(\tilde{\mathbf{y}}|\mathrm{y} = g', \mathbf{x}_i)$. We thereby consider the following hypothesis testing problem: $H_g : \tilde{\mathbf{y}}_i|\{\mathrm{y}_i, \mathbf{x}_i\} \sim \mathbb{P}(\tilde{\mathbf{y}}|\mathrm{y} = g, \mathbf{x}_i)$ versus $H_{g'} : \tilde{\mathbf{y}}_i|\{\mathrm{y}_i, \mathbf{x}_i\} \sim \mathbb{P}(\tilde{\mathbf{y}}|\mathrm{y} = g', \mathbf{x}_i)$. By the Neyman-Pearson Lemma [39], the Bayes testing error is minimized by the likelihood ratio test, and the decision region for hypothesis $H_g$ is given by

$$\left\{\tilde{\mathbf{y}} : \frac{\hbar_{i,g}\mathbb{P}(\tilde{\mathbf{y}}|\mathrm{y} = g, \mathbf{x}_i)}{\hbar_{i,g'}\mathbb{P}(\tilde{\mathbf{y}}|\mathrm{y} = g', \mathbf{x}_i)} = \frac{\hbar_{i,g} \prod_{r=1}^R \prod_{l=1}^K \left\{\tau_{i,gl}^{(r)}\right\}^{\mathbb{I}(\tilde{y}^{(r)}=l)}}{\hbar_{i,g'} \prod_{r=1}^R \prod_{l=1}^K \left\{\tau_{i,g'l}^{(r)}\right\}^{\mathbb{I}(\tilde{y}^{(r)}=l)}} > 1\right\}.$$

Building from the abovementioned reformulation of the label correction process, we now propose an algorithm to infer the underlying ground truth by aggregating noisy crowdsourced annotations with the help of the annotator confusions. Formally, we propose the following label correction method by setting the estimated label of $\mathbf{x}_i$ to be $\overline{\mathrm{y}}_i \triangleq g$ if

$$\frac{\hbar_{i,g} \prod_{r=1}^R \prod_{l=1}^K \left\{\tau_{i,gl}^{(r)}\right\}^{\mathbf{1}(\tilde{y}_i^{(r)}=l)}}{\hbar_{i,g'} \prod_{r=1}^R \prod_{l=1}^K \left\{\tau_{i,g'l}^{(r)}\right\}^{\mathbf{1}(\tilde{y}_i^{(r)}=l)}} > \Omega \text{ for any } g' \neq g, \tag{10}$$

where $\Omega \geq 1$ is a pre-specified threshold.

**Information-theoretic bounds on the Bayes error.** To theoretically justify the effectiveness of the proposed label correction method (10), we derive information-theoretic bounds on the Bayes error, given the instances. Let $\overline{\mathcal{D}} = \{\mathbf{x}_i, \overline{\mathrm{y}}_i\}_{i=1}^{\bar{n}}$ denote the collection of instances with estimated labels, where $\bar{n}$ represents the size of $\overline{\mathcal{D}}$. We write $\overline{\mathbf{y}} = \{\overline{\mathrm{y}}_i\}_{i=1}^{\bar{n}}$ and the corresponding true label is denoted $\mathbf{y} = \{\mathrm{y}_i\}_{i=1}^{\bar{n}}$. A loss measured by the accuracy of the estimated labels is given by $\mathcal{L}(\overline{\mathbf{y}}, \mathbf{y}) = \frac{1}{\bar{n}} \sum_{i=1}^{\bar{n}} \mathbf{1}(\overline{\mathrm{y}}_i \neq \mathrm{y}_i)$. Let $\mathbb{P}(\cdot|\mathbf{y}; \boldsymbol{\tau})$ denote the joint probability distribution of $\{\tilde{\mathbf{y}}_i\}$, given $\mathbf{y}$ and $\boldsymbol{\tau}$, and let $\mathbb{E}(\cdot|\mathbf{y}; \boldsymbol{\tau})$ denote the associated expectation operator, where $\boldsymbol{\tau} \triangleq \{\boldsymbol{\tau}_i\}_{i=1}^{\bar{n}} \triangleq \{\boldsymbol{\tau}_i^{(r)} :$

$r \in [R]\}_{i=1}^{\bar{n}}$ represents the collection of the corresponding transition matrices $\boldsymbol{\tau}_i^{(r)}$ having $\tau_{i,kl}^{(r)}$ as its $(k, l)$ element. Then, the Bayes risk is defined as [40]

$$\Re_{\text{Bayes}}(\hbar, \mathcal{L}) = \inf_{\overline{\mathbf{y}}} \left[ \sum_{\mathbf{y} \in [K]^{\bar{n}}} \hbar(\mathbf{y}) \mathbb{E}\{\mathcal{L}(\overline{\mathbf{y}}, \mathbf{y}) | \mathbf{y}; \boldsymbol{\tau}\} \right], \tag{11}$$

or $\Re_{\text{Bayes}}$ for short, where $\hbar(\mathbf{y})$ is the joint prior probability of $\mathbf{y}$ calculated from $\hbar \triangleq \{\hbar_i\}_{i=1}^{\bar{n}}$. The following theorem identifies bounds for the Bayes risk.

**Theorem 2.** *Let $D_{KL}\left(\boldsymbol{\tau}_{i,g*}^{(r)} \| \boldsymbol{\tau}_{i,g'*}^{(r)}\right)$ denote the Kullback-Leibler (KL) divergence for discrete distributions $\boldsymbol{\tau}_{i,g*}^{(r)}$ and $\boldsymbol{\tau}_{i,g'*}^{(r)}$, where, for $i \in [\bar{n}]$, $r \in [R]$, and $g, g' \in [K]$, $\boldsymbol{\tau}_{i,g*}^{(r)}$ and $\boldsymbol{\tau}_{i,g'*}^{(r)}$ stand for the $g$th and $g'$th rows of $\boldsymbol{\tau}_i^{(r)}$, respectively. For $\hbar = \{\hbar_i\}_{i=1}^{\bar{n}}$ and $\boldsymbol{\tau} = \{\boldsymbol{\tau}_i\}_{i=1}^{\bar{n}}$, define*

$$\overline{D}_{KL}(\hbar, \boldsymbol{\tau}) = \frac{1}{\bar{n}} \sum_{i=1}^{\bar{n}} \sum_{r=1}^{R} \sum_{g=1}^{K} \sum_{g'=1}^{K} \hbar_{i,g} \hbar_{i,g'} D_{KL}\left(\boldsymbol{\tau}_{i,g*}^{(r)} \| \boldsymbol{\tau}_{i,g'*}^{(r)}\right) \quad \text{and}$$

$$C_{gg'}^{(i)} = - \min_{0 \le \lambda \le 1} \frac{1}{R} \left[ -\lambda \log \left( \frac{\Omega \hbar_{i,g}}{\hbar_{i,g'}} \right) + \sum_{r=1}^{R} \log \left\{ \sum_{l=1}^{K} \left( \tau_{i,gl}^{(r)} \right)^{1-\lambda} \left( \tau_{i,g'l}^{(r)} \right)^{\lambda} \right\} \right].$$

*For $i \in \bar{n}$ and $g \in [K]$, let $I_\Omega^{(g)}(\hbar_i, \boldsymbol{\tau}_i) = \min_{g' \ne g} C_{gg'}^{(i)}$. Then the Bayes error defined in (11) is bounded as follows:*

$$\frac{1}{\bar{n}} \left[ 1 - \frac{\overline{D}_{KL}(\hbar, \boldsymbol{\tau}) + \frac{1}{\bar{n}} \log(2 - \prod_{i=1}^{\bar{n}} \max_{k \in [K]} \hbar_{i,k})}{\left\{ \sum_{i=1}^{\bar{n}} \log(\max_{k \in [K]} \hbar_{i,k}) \right\} / \bar{n}} \right]$$

$$\le \Re_{Bayes}(\hbar, \mathcal{L}) \le \frac{K-1}{\bar{n}} \sum_{i=1}^{\bar{n}} \sum_{g=1}^{K} \hbar_{i,g} \exp \left\{ -R I_\Omega^{(g)}(\hbar_i, \boldsymbol{\tau}_i) \right\}.$$

*Remark* 1. Theorem 2 establishes information-theoretic bounds on the Bayes error $\Re_{\text{Bayes}}$ for arbitrary priors $\hbar_i$ with $i \in [\bar{n}]$, which theoretically quantifies the combined impact of the prior knowledge and annotators' expertise on label accuracy using algorithm (10). The lower bound is proved in light of the concept of $f$-informativity [40–42], and is stronger than the commonly-used Bayes lower bound based on Fano's inequality [43, 44]. The proof of the upper bound considers the inference procedure of $\overline{\mathbf{y}}$ and applies Markov's inequality. The details are given in Appendix B.

*Remark* 2. The quantity $C_{gg'}^{(i)}$ in Theorem 2 reflects how the identified upper bound of the Bayes error may be influenced by the prior $\hbar_i$ and the ability of the $R$ annotators to distinguish between labels $g$ and $g'$ for instance $i$. If we set $\hbar_i$ to be the uniform prior and let $\Omega = 1$, $C_{gg'}^{(i)}$ will degenerates to the average of the Chernoff information between $\{\boldsymbol{\tau}_{i,g*}^{(r)}\}_{r=1}^{R}$ and $\{\boldsymbol{\tau}_{i,g'*}^{(r)}\}_{r=1}^{R}$[45, 46], which is a statistical divergence measuring the deviation between two probability distributions.

**Result under the sparse Bayesian model.** The label correction method (10) is not immediately applicable if we have no access to the underlying true annotator confusions, which is usually the case in real-world applications. To get around the issue induced from the unknownness of the true noise transition probability $f_0^{(k,r)}$, we consider the model $f_{\boldsymbol{\theta}}^{(k,r)}$ given in Theorem 1 and write the corresponding transition matrices as $\overline{\tau}_{i,kl}^{(r)} \triangleq \overline{\tau}_{kl}^{(r)}(\mathbf{x}_i) \triangleq f_{\boldsymbol{\theta}}^{(k,r)}(\tilde{y} | \mathbf{x}_i)|_{\tilde{y}=l}$ for $k, l \in [K]$ and $r \in [R]$. We then replace the $\tau_{i,gl}^{(r)}$ in (10) with $\overline{\tau}_{i,gl}^{(r)}$ to determine corrected labels. With a slight abuse of notation, we still use $\overline{\mathcal{D}} = \{\mathbf{x}_i, \overline{y}_i\}_{i=1}^{\bar{n}}$ to denote the set of instances with estimated labels but let $\overline{\Re}_{\text{Bayes}}$ denote the resulting Bayes error. We let $\overline{\mathcal{D}}_{\mathbf{x}} = \{\mathbf{x}_i\}_{i=1}^{\bar{n}}$ represent the set of the considered instances. Combining Theorems 1 and 2 yields the following corollary.

**Corollary 3.** *Suppose that the conditions in Theorem 1 are satisfied, and further assume that $f(\mathbf{x}_i) > \varsigma_2$ for $\mathbf{x}_i \in \overline{\mathcal{D}}_{\mathbf{x}}$ and $\min_{r \in [R], i \in [\bar{n}], k, l \in [K]} \overline{\tau}_{i,kl}^{(r)} \ge \varsigma_3$, where $f(\cdot)$ is the probability density*

*function of* **x**, *and $\varsigma_2$ and $\varsigma_3$ are some positive constants. Then, for any $\epsilon > 0$,*

$$\Pi\left[\boldsymbol{\theta} : \overline{\Re}_{Bayes} \leq \frac{K-1}{\bar{n}}\sum_{i=1}^{\bar{n}}\sum_{g=1}^{K}\hbar_{i,g}\exp\left\{-RI_{\Omega}^{(g)}(\hbar_i, \boldsymbol{\tau}_i) + \epsilon\right\}\Bigg|\overline{\mathcal{D}}_0\right] \longrightarrow 1$$

*in $\mathbb{P}_0^n$ probability as $n \to \infty$, where $I_{\Omega}^{(g)}(\hbar_i, \boldsymbol{\tau}_i)$ is given in Theorem 2 for $i \in [\bar{n}]$ and $g \in [K]$.*

## 4 Algorithm

**Learning the noise transition model.**   In the warm-up stage, we train the base models on noisy training data and obtain the set of anchor points $\overline{\mathcal{D}}_0$ [26]. With $\overline{\mathcal{D}}_0$, we first obtain the maximum a posteriori (MAP) estimate of network parameters of the transition model $\boldsymbol{\theta}$ by maximizing the log posterior distribution of $\boldsymbol{\theta}$, with the constant term omitted,

$$\widehat{\boldsymbol{\theta}} = \arg\max_{\boldsymbol{\theta}}\left\{\sum_{i=1}^{n}\log p_{\boldsymbol{\theta},i} + \log\pi(\boldsymbol{\theta})\right\}, \tag{12}$$

where $p_{\boldsymbol{\theta},i}$ is the probability mass function of the $i$th component in $\overline{\mathcal{D}}_0$ given before Theorem 1, and $\pi(\boldsymbol{\theta})$ is the probability density function of $\boldsymbol{\theta}$ relative to the prior probability measure $\Pi(\cdot)$. Given the MAP estimate $\widehat{\boldsymbol{\theta}}$, according to the prior hierarchy (4), the posterior inclusion probability of the $k$th parameter in the network $\psi_j(\cdot; \boldsymbol{\theta}^{(j)})$ is calculated as

$$\mathbb{P}(\gamma_k^{(j)} = 1|\widehat{\boldsymbol{\theta}}) = \frac{\lambda_{nj}\pi_1(\widehat{\theta}_k^{(j)}; \sigma_{nj}^2)}{\lambda_{nj}\pi_1(\widehat{\theta}_k^{(j)}; \sigma_{nj}^2) + (1 - \lambda_{nj})\pi_0(\widehat{\theta}_k^{(j)}; c_{nj}\sigma_{nj}^2)} \tag{13}$$

for $k \in [J_j]$ with $j = 1, 2$. If the posterior inclusion probability is smaller than a pre-specified threshold, chosen to be 0.5 in our experiments, the associated parameter is zet to be zero. We then fine tune the sparse network and obtain the noise transition model.

**Training the classifiers with corrected labels.**   With the noise transition model trained, we then train the base models with the label correction algorithm proposed in Section 3.3. Specifically, we train two base classifiers to reciprocally provide class priors for each instance in the label correction process. In the $t$th epoch, for instance $\mathbf{x}_i$, if $\overline{y}_i$ satisfies the condition (10) for the pre-specified threshold $\Omega_t$, we put $\{\mathbf{x}_i, \overline{y}_i\}$ in $\overline{D}_t$. The base models are then updated on the collected dataset $\overline{D}_t$. The complete pseudocode of our algorithm is included in Appendix C.

## 5 Experiments

**Datasets.**   We assess the effectiveness of our method on three image datasets with synthetic annotations, MNIST [47], CIFAR-10 [48], and CIFAR-100 [48], and two datasets with human annotations, CIFAR-10N [49] and LabelMe [50, 51]. Detailed information can be found in Appendix C. For all the datasets except LabelMe, we leave out 10% of the training data as a noisy validation set.

**Noise generation.**   For the three datasets, MNIST, CIFAR10, and CIFAR100, we consider three groups of annotators with varying expertise, with an average labeling error rate of about 20%, 35% and 50%, respectively. We abbreviate these three groups as IDN-LOW, IDN-MID, and IDN-HIGH, which represent instance-dependent annotators with low, middle ("mid" for short), and high labeling error rates, respectively. To generate noisy annotations, we independently simulate $R = 5$ annotators for each group according to Algorihtm 2 in [12], with IDN-$\tau$ denoting that the noise rate is upper bounded by $\tau$ for each annotator. For each instance, we then randomly choose one of the annotations given by the $R$ annotators, which is designed to evaluate the methods under incomplete annotator labeling. We manually corrupt the three datasets according to the following three groups of annotators:

        **(I) IDN-LOW.** *IDN-10%, IDN-10%, IDN-20%, IDN-20%, IDN-30%;*

        **(II) IDN-MID.** *IDN-30%, IDN-30%, IDN-40%, IDN-40%, IDN-50%;*

        **(III) IDN-HIGH.** *IDN-50%, IDN-50%, IDN-60%, IDN-60%, IDN-70%.*

**Experiment setup.** The network structure for the MNIST dataset is chosen to be Lenet-5 [52]. We choose ResNet-18 [2] for CIFAR-10 and CIFAR-10N, and ResNet-34 architecture [2] for CIFAR-100 dataset. As in [53], we employ the pretrained VGG-16 network followed by a fully connected layer and a softmax output layer for the LabelMe dataset, using 50% dropout. More implementation details can be found in Appendix C.

**Competing methods.** We compare the proposed method with the following state-of-art methods: (1) CE (Clean), which trains the network with the standard cross entropy loss on the clean datasets; (2) CE (MV), which trains the network using the labels from majority voting; (3) CE (EM) [9]; (4) DoctorNet [54]; (5) GCE [55]; (6) Co-teaching [56]; (6) Co-teaching+ [57]; (7) BLTM [17]; (8) MBEM [11]; (9) CrowdLayer [53]; (10) TraceReg [8]; (11) Max-MIG [7]; (12) CoNAL [58]; (13) GeoCrowdNet (F) [13]; and (14) GeoCrowdNet (W) [13]. Among these methods, GCE, Co-teaching, Co-teaching+, and BLTM are strong baselines dealing with single noisy label issues, and we adapt them to the multiple annotations setting by utilizing the majority vote labels for loss computation.

**Ablation study.** In Figure 1, We plot the average estimation error for the noise transition matrices to demonstrate the effectiveness of the proposed method in modeling the instance-dependent annotator confusions. For instance $\mathbf{x}_i$ with clean class label $\mathbf{y}_i$ in the validation set, we analyse the $\mathbf{y}_i$th row rather than the whole noise transition matrix as in previous studies [16, 12]. Specifically, let $\widehat{\tau}^{(r)}(\mathbf{x}_i)$ and $\tau^{(r)}(\mathbf{x}_i)$ represent the estimated and the true noise transition matrix for annotator $r$, respectively. The estimation error on instance $\mathbf{x}_i$ is defined as $err_i^{(r)} = \max_{l \in K} |\widehat{\tau}^{(r)}(\mathbf{x}_i)_{\mathbf{y}_i,l} - \tau^{(r)}(\mathbf{x}_i)_{\mathbf{y}_i,l}|$, where $\widehat{\tau}^{(r)}(\mathbf{x}_i)_{\mathbf{y}_i,l}$ and $\tau^{(r)}(\mathbf{x}_i)_{\mathbf{y}_i,l}$ stand for the $(\mathbf{y}_i, l)$ element in the corresponding matrices. The average estimation error for annotator $r$ is then calculated as $\frac{1}{n_v}\sum_{i=1}^{n_v} err_i^{(r)}$ with $n_v$ denoting the size of the validation set. For each annotator, we compare the average estimation error of the proposed method with six baselines, CrowdLayer [53], TraceReg [8], GeoCrowdNet (F) [13], GeoCrowdNet (W) [13], MBEM [11], and BLTM [17], on the CIFAR10 dataset. In most of the cases, the proposed method leads to smaller estimation error especially when the noise rate is high, which shows the efficacy of the proposed sparse Bayesian model.

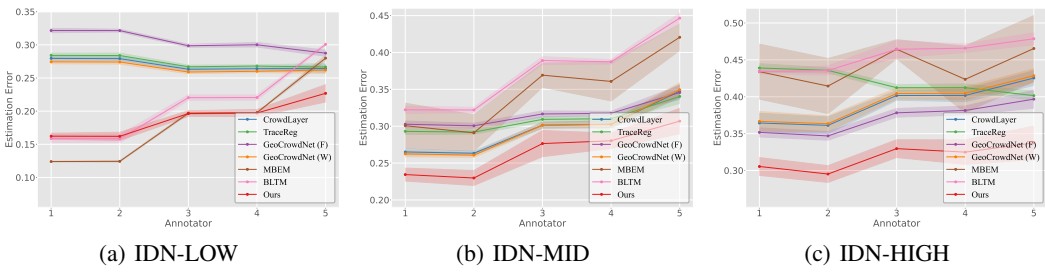

|            |            |            |
|:----------:|:----------:|:----------:|
| (a) IDN-LOW | (b) IDN-MID | (c) IDN-HIGH |

Figure 1: Average estimation error of annotator-specific instance-dependent noise transition matrices on CIFAR10. The error bar for standard deviation has been shaded.

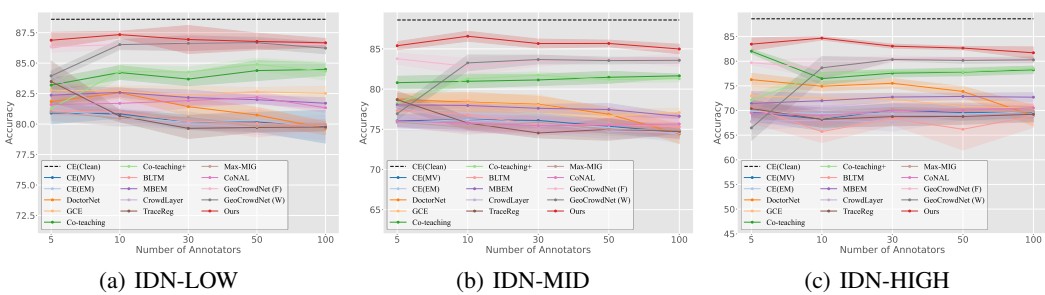

|            |            |            |
|:----------:|:----------:|:----------:|
| (a) IDN-LOW | (b) IDN-MID | (c) IDN-HIGH |

Figure 2: Average accuracy of learning CIFAR-10 dataset with varying number of annotators. The error bar for standard deviation has been shaded.

Table 1: Average accuracy of learning CIFAR-10, CIFAR-100, CIFAR-10N and LabelMe datasets

| | CIFAR-10 | | | CIFAR-100 | | | CIFAR-10N | LabelMe |
|---|---|---|---|---|---|---|---|---|
| | IDN-LOW | IDN-MID | IDN-HIGH | IDN-LOW | IDN-MID | IDN-HIGH | | |
| CE (Clean) | $88.60_{\pm 0.79}$ | | | $58.75_{\pm 0.55}$ | | | $88.60_{\pm 0.79}$ | $91.45_{\pm 0.23}$ |
| CE (MV) | $80.90_{\pm 0.88}$ | $76.05_{\pm 0.70}$ | $69.65_{\pm 1.73}$ | $50.96_{\pm 0.49}$ | $44.80_{\pm 0.99}$ | $38.51_{\pm 0.66}$ | $82.82_{\pm 0.05}$ | $79.49_{\pm 0.48}$ |
| CE (EM) [9] | $81.15_{\pm 0.74}$ | $75.84_{\pm 0.97}$ | $69.85_{\pm 1.43}$ | $51.29_{\pm 1.00}$ | $45.24_{\pm 0.41}$ | $38.01_{\pm 0.90}$ | $83.14_{\pm 0.80}$ | $80.64_{\pm 0.55}$ |
| DoctorNet [54] | $81.85_{\pm 0.41}$ | $78.69_{\pm 0.75}$ | $76.26_{\pm 1.28}$ | $52.61_{\pm 0.70}$ | $47.80_{\pm 0.86}$ | $43.50_{\pm 0.53}$ | $84.52_{\pm 0.69}$ | $79.09_{\pm 0.40}$ |
| GCE [55] | $82.78_{\pm 0.51}$ | $78.08_{\pm 1.18}$ | $72.99_{\pm 1.92}$ | $\mathbf{55.88}_{\pm 1.32}$ | $\mathbf{48.46}_{\pm 0.86}$ | $40.53_{\pm 0.83}$ | $85.25_{\pm 0.46}$ | $80.27_{\pm 0.27}$ |
| Co-teaching [56] | $83.20_{\pm 0.53}$ | $80.80_{\pm 0.79}$ | $\mathbf{82.02}_{\pm 0.42}$ | $53.27_{\pm 0.42}$ | $47.58_{\pm 0.43}$ | $\mathbf{45.49}_{\pm 0.72}$ | $85.90_{\pm 0.50}$ | $80.24_{\pm 0.71}$ |
| Co-teaching+ [57] | $81.27_{\pm 0.44}$ | $78.26_{\pm 0.27}$ | $72.10_{\pm 0.98}$ | $53.31_{\pm 0.81}$ | $48.15_{\pm 0.36}$ | $42.07_{\pm 0.66}$ | $82.31_{\pm 0.89}$ | $81.67_{\pm 0.56}$ |
| BLTM [17] | $81.06_{\pm 0.23}$ | $77.34_{\pm 0.51}$ | $70.64_{\pm 3.19}$ | $52.21_{\pm 0.70}$ | $46.90_{\pm 0.85}$ | $41.26_{\pm 1.59}$ | $82.62_{\pm 0.17}$ | $80.44_{\pm 1.05}$ |
| MBEM [11] | $82.37_{\pm 0.77}$ | $78.05_{\pm 0.83}$ | $71.43_{\pm 2.43}$ | $52.20_{\pm 0.07}$ | $45.26_{\pm 0.50}$ | $38.92_{\pm 0.69}$ | $85.49_{\pm 0.43}$ | $80.10_{\pm 1.09}$ |
| CrowdLayer [53] | $83.98_{\pm 0.35}$ | $77.76_{\pm 1.06}$ | $67.77_{\pm 1.69}$ | $51.28_{\pm 0.64}$ | $45.28_{\pm 0.64}$ | $38.93_{\pm 0.76}$ | $82.84_{\pm 0.24}$ | $82.95_{\pm 0.21}$ |
| TraceReg [8] | $83.49_{\pm 1.68}$ | $78.69_{\pm 1.04}$ | $70.39_{\pm 1.68}$ | $51.60_{\pm 0.99}$ | $45.16_{\pm 0.45}$ | $39.01_{\pm 0.83}$ | $83.16_{\pm 0.24}$ | $82.93_{\pm 0.15}$ |
| Max-MIG [7] | $81.00_{\pm 0.72}$ | $75.90_{\pm 0.52}$ | $70.96_{\pm 0.96}$ | $51.76_{\pm 1.11}$ | $44.93_{\pm 0.71}$ | $38.70_{\pm 0.49}$ | $85.12_{\pm 0.36}$ | $83.25_{\pm 0.26}$ |
| CoNAL [58] | $81.60_{\pm 0.82}$ | $76.02_{\pm 0.79}$ | $69.50_{\pm 1.89}$ | $51.61_{\pm 1.14}$ | $44.19_{\pm 0.62}$ | $38.24_{\pm 0.29}$ | $83.01_{\pm 0.21}$ | $82.96_{\pm 0.30}$ |
| GeoCrowdNet (F) [13] | $\mathbf{86.36}_{\pm 0.46}$ | $\mathbf{83.78}_{\pm 0.68}$ | $79.70_{\pm 0.42}$ | $51.37_{\pm 0.88}$ | $45.04_{\pm 0.56}$ | $38.94_{\pm 0.91}$ | $87.70_{\pm 0.51}$ | $\mathbf{85.74}_{\pm 0.17}$ |
| GeoCrowdNet (W) [13] | $83.95_{\pm 0.41}$ | $76.94_{\pm 0.72}$ | $66.48_{\pm 2.53}$ | $51.58_{\pm 0.72}$ | $45.24_{\pm 1.15}$ | $39.24_{\pm 0.76}$ | $\mathbf{87.84}_{\pm 0.21}$ | $83.28_{\pm 0.45}$ |
| Ours | $\mathbf{86.88}_{\pm 0.65}$ | $\mathbf{85.40}_{\pm 0.50}$ | $\mathbf{83.46}_{\pm 1.24}$ | $\mathbf{59.81}_{\pm 0.55}$ | $\mathbf{54.88}_{\pm 0.60}$ | $\mathbf{49.44}_{\pm 1.30}$ | $\mathbf{88.19}_{\pm 0.47}$ | $\mathbf{84.85}_{\pm 0.27}$ |

**Classification accuracy.** Table 1 presents the average test accuracy of 5 random trials on the datasets of CIFAR-10, CIFAR-100, CIFAR-10N and LabelMe, together with the standard errors of the test accuracies of the random trials, expressed after the plus/minus sign $\pm$, where the two highest accuraries are bold faced; standard errors of the accuracies are calculated based on repeating those experiments 5 times, each with a different random seed. All the results demonstrate the superior performance of the proposed method on both synthetic and real-world noisy datasets. Moreover, to investigate the influence of the sparsity of annotations, we conduct more experiments with the number of annotators varying from 5 to 100, and each instance only has one label. Figure 2 shows the average accuracy with various numbers of annotators, which further exhibit the advantages of the proposed method under different settings. Additional experimental results, including the test accuracy on MNIST, the average estimation error on MNIST and CIFAR100, and the accuracy of the corrected labels using algorithm (10), are deferred to Appendix C to save space.

## 6 Conclusion

In this paper, we address the challenge of training classifiers using noisy crowdsourced labels, a common issue in various applications. We formulate the annotator-specific instance-dependent noise transition matrix within the Bayesian framework, and theoretically characterize the closeness of the proposed model and the true annotator confusions with respect to the Hellinger distance. Our result is established for the setup of i.n.i.d. observations, which substantially broadens the application scope of our method. Building on the convergence rate of the posterior measure, we propose a novel algorithm to aggregate noisy annotations and infer the ground truth label based using pairwise LRT. Additionally, we provide information-theoretic bounds on the Bayes error of the proposed algorithm. Empirical evidence demonstrates the effectiveness of our algorithm on both synthetic and real-world noisy datasets.

## Limitations and Extensions

Our work can be further extended in different directions. It is interesting to generalize the setup here to the hierarchical classification setup. Instance-dependent transition matrices can be further refined with varying structures imposed and are learned with manifold regularization.

## Acknowledgements

Yi is the Canada Research Chair in Data Science (Tier 1). Her research was supported by the Canada Research Chairs Program and the Natural Sciences and Engineering Research Council of Canada (NSERC).

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

# Supplemental Materials

In the supplementary materials, we first summarize the regularity conditions on the underlying true model and prior distributions of the network parameters in Section A. The proofs of Theorem 1, Theorem 2, and Corollary 3 are provided in Sections B.2-B.4 with all the needed preliminaries presented in Section B.1. Implementation details of the proposed method and additional experiment results are exhibited in Section C, including the accuracy and number of selected labels using the proposed label correction algorithm (10), the test accuracy on MNIST, the hyperparameter analysis on CIFAR100, the classification accuracy on CIFAR100 with varying number of annotators, and the average estimation error on CIFAR10 and CIFAR100 with varying number of annotators.

## A    Regularity Conditions

### A.1    Network structure

To incorporate the sparse high dimensional setting [59], we utilize sparse Bayesian DNNs to reconstruct $\psi_1(\mathbf{x})$ and $\psi_2(\mathbf{x})$ in (3) [37, 60]. Specifically, to approximate $\psi_j(\mathbf{x})$ with $j = 1, 2$, we consider a network with $H_{nj} - 1$ hidden layers and a width vector $\boldsymbol{L}^{(j)} = (L_0^{(j)}, L_1^{(j)}, ..., L_{H_{nj}}^{(j)})^\top$, where the width of the $h$th layer is denoted $L_h^{(j)}$ for $h = 0, ..., H_{nj}$ with $L_0^{(j)} = p_n$ for the input layer and $L_{H_{nj}}^{(j)} \triangleq M^{(j)}$ for the output layer. Then the DNN with network architecture $\{H_{nj}, \boldsymbol{L}^{(j)}\}$ is the nonlinear function of the form:

$$\psi_j(\mathbf{x}; \boldsymbol{\theta}^{(j)}) = \mathbf{W}^{(j, H_{nj})} \sigma \left( ...\sigma \left[ \mathbf{W}^{(j,h)} \sigma \left\{ ...\sigma(\mathbf{W}^{(j,1)}\mathbf{x} + \mathbf{b}^{(j,1)})... \right\} + \mathbf{b}^{(j,h)} \right] ... \right) + \mathbf{b}^{(j, H_{nj})},$$

where for $h = 1, ..., H_{nj}$, $\mathbf{W}^{(j,h)}$ is a $L_h^{(j)} \times L_{h-1}^{(j)}$ weight matrix, $\mathbf{b}^{(j,h)} \in \mathbb{R}^{L_h^{(j)}}$ is the bias of layer $h$, $\sigma(\cdot)$ is a nonlinear activation function, and $\boldsymbol{\theta}^{(j)}$ represents the $J_{nj} \times 1$ vector formed from stacking $\{\mathbf{W}^{(j,h)}, \mathbf{b}^{(j,h)}\}_{h=1}^{H_{nj}}$ from bottom to the top, with $J_{nj} \triangleq \sum_{h=1}^{H_{nj}} (L_{h-1}^{(j)} \times L_h^{(j)} + L_h^{(j)})$. We treat weights and biases equally without distinguishing them in $\boldsymbol{\theta}^{(j)}$, and write $\boldsymbol{\theta}^{(j)}$ as $\boldsymbol{\theta}^{(j)} = (\theta_1^{(j)}, ..., \theta_{J_{nj}}^{(j)})^\top$. Let $\boldsymbol{\gamma}^{(j)} = (\gamma_1^{(j)}, ..., \gamma_{J_{nj}}^{(j)})^\top$ denote the indicator vector, with $\gamma_k^{(j)} = \mathbf{1}(\theta_k^{(j)} \neq 0)$ for $k = 1, ..., J_{nj}$. For ease of presentation, we use $\psi_j(\mathbf{x}; \boldsymbol{\theta}^{(j)})$ and $\psi_j(\mathbf{x}; \boldsymbol{\theta}^{(j)}, \boldsymbol{\gamma}^{(j)})$ exchangeably to represent the model for $\psi_j(\mathbf{x})$, and let $\mathcal{F}_n = \mathcal{F}(H_{n1}, H_{n2}, \boldsymbol{L}^{(1)}, \boldsymbol{L}^{(2)}, C_1, C_2, \epsilon_1)$ denote the space of all sparse networks that satisfy Condition A.3 in Appendix A.2 and are constrained by positive constants $C_1$, $C_2$ and $\epsilon_1$.

To determine the parameters of the sparse DNNs $\psi_j(\mathbf{x}; \boldsymbol{\theta}^{(j)}, \boldsymbol{\gamma}^{(j)})$ that best approximate $\psi_j(\mathbf{x})$ for $j = 1, 2$, we define

$$(\boldsymbol{\theta}^{(1)*}, \boldsymbol{\gamma}^{(1)*}, \boldsymbol{\theta}^{(2)*}, \boldsymbol{\gamma}^{(2)*}) = \underset{\substack{(\boldsymbol{\theta}^{(1)}, \boldsymbol{\gamma}^{(1)}, \boldsymbol{\theta}^{(2)}, \boldsymbol{\gamma}^{(2)}) \in \mathcal{F}_n \\ \|\psi_1(\mathbf{x}; \boldsymbol{\theta}^{(1)}, \boldsymbol{\gamma}^{(1)}) - \psi_1(\mathbf{x})\|_{L^2(\Omega)} \leq \varpi_{n1} \\ \|\psi_2(\mathbf{x}; \boldsymbol{\theta}^{(2)}, \boldsymbol{\gamma}^{(2)}) - \psi_2(\mathbf{x})\|_{L^2(\Omega)} \leq \varpi_{n2}}}{\arg \min} \left\{ |\boldsymbol{\gamma}^{(1)}| + |\boldsymbol{\gamma}^{(2)}| \right\}, \qquad (14)$$

where for $j = 1, 2$, $\varpi_{nj}$ is an $n$-dependent positive constant satisfying $\varpi_{nj} \to 0$ as $n \to \infty$[3] We call $\boldsymbol{\theta}^{(1)*}$, $\boldsymbol{\gamma}^{(1)*}$, $\boldsymbol{\theta}^{(2)*}$, and $\boldsymbol{\gamma}^{(2)*}$ the *true parameters* of $\boldsymbol{\theta}^{(1)}$, $\boldsymbol{\gamma}^{(1)}$, $\boldsymbol{\theta}^{(2)}$, and $\boldsymbol{\gamma}^{(2)}$, respectively.

### A.2    Conditions for the sparse GLM

   A.1 The input vector is standardized so that $\mathbf{x} \in \Omega \subset [-1, 1]^{p_n}$, where $\Omega$ is the support of $\mathbf{x}$, and the probability density $f(\cdot)$ of $\mathbf{x}$ satisfies that $\sup_{\mathbf{x} \in \Omega} |f(\mathbf{x})| \leq C_0$ for some positive constant $C_0$.

   A.2 The activation function $\sigma(\cdot)$ is 1-Lipschitz.

---

[3]In this definition, the $L_2$ norm of the network approximation error is bounded. Consider a measure space $(\Omega, \mathcal{G}, \mu)$ and $0 < p < q \leq \infty$, we have that $\|f\|_p \leq \mu(\Omega)^{1/p-1/q} \|f\|_q$ using Hölder's inequality and therefore, $\|f\|_1 \leq \|f\|_2$ and $L^1(\Omega, \mu) \subset L^2(\Omega, \mu)$ for probability measure $\mu$. Thus, the $L_1$ norm of the network approximation error is also bounded, which will be used in the following proofs.

A.3 The true sparse DNN model satisfies the following conditions.

A.3.1 For $j = 1, 2$, let $r_{nj} = \|\boldsymbol{\gamma}^{(j)}\|_1$ denote the connectivity of $\boldsymbol{\gamma}^{(j)}$, let $\overline{L}_{nj} = \max_{1 \leq h \leq H_{nj}} L_h^{(j)}$ denote the maximum layer width, and let $s_{nj}$ represent the input dimension of $\boldsymbol{\gamma}^{(j)}$. Let $\zeta_n = \{(r_{n1} + r_{n2})(H_{n1} + H_{n2})\log n + (r_{n1} + r_{n2})(\log \overline{L}_{n1} + \log \overline{L}_{n2}) + (s_{n1} + s_{n2})\log p_n\}/n$. The true sparse DNN model satisfies that $\zeta_n \leq C_1 n^{-\epsilon_1}$ for some constants $C_1 > 0$ and $0 < \epsilon_1 < 1$.

A.3.2 For $j = 1, 2$, $\|\boldsymbol{\theta}^{(j)}\|_\infty \leq E_{nj}$, where positive constant $E_{nj} \leq n^{C_2}$ for some constant $C_2 > 0$.

A.4 Write the function $G(\cdot)$ in (3) as $(G_1(\cdot), ..., G_K(\cdot))^\top$. For $\mathbf{x} \in \Omega$, $r \in [R]$ and $k \in [K]$, write $\boldsymbol{\omega}^{(k,r)} = \mathbf{A}^{(r)\mathrm{T}}\psi_1(\mathbf{x}; \boldsymbol{\theta}^{(1)}) + \mathbf{B}^{(k)\mathrm{T}}\psi_2(\mathbf{x}; \boldsymbol{\theta}^{(2)})$.

A.4.1 For $k \in [K]$ and $r \in [R]$, in the neighbourhood of $\boldsymbol{\theta}_0$,

$$\sup_{j,l \in [K]} \left|\frac{\partial G_j(\boldsymbol{\omega})}{\partial \omega_l}\Big|_{\boldsymbol{\omega}=\boldsymbol{\omega}^{(k,r)}}\right| = C_3 \text{ and } \sup_{j \in [K]} \left|\frac{G_j(\boldsymbol{\omega}^{(k,r)})}{G_j(\boldsymbol{\omega}_0^{(k,r)})} - 1\right| = \epsilon_2$$

for some constants $C_3 > 0$ and $\epsilon_2 \in (0, 1)$, where the latter requirement can be achieved if $G_j(\boldsymbol{\omega}_0^{k,r}) > \varsigma$ for some positive constant $\varsigma > 0$.

A.4.2 For $r \in [R]$ and $j, k \in [K]$, $\|\boldsymbol{\alpha}_{j0}^{(r)}\|_\infty \leq F_1$ and $\|\boldsymbol{\beta}_{j0}^{(k)}\|_\infty \leq F_2$, where $F_1$ and $F_2$ are positive constants.

## A.3 Conditions for the prior

B.1 For $j = 1, 2$, assume each element of $\boldsymbol{\theta}^{(j)}$ has independent continuous prior distribution, denoted $\pi_\theta^{(j)}(\cdot)$. Thus, its minimum value on the interval $[-E_{nj}-1, E_{nj}+1]$ exits, and let $\underline{\pi}_\theta^{(j)}$ denote it. For a sequence of positive constants $I_n^{(j)}$ with $\log I_n^{(j)} = O(\log n)$, let $\delta_{nj}$ and $\delta'_{nj}$ be two sequences of constants satisfying that $\delta_{nj} < 1/n J_{nj}(c_0 I_n^{(j)})^{H_{nj}}(n/H_{nj})^{H_{nj}}$ and $\delta'_{nj} < 1/n J_{nj}(c_0 E_{nj})^{H_{nj}}(r_{nj}/H_{nj})^{H_{nj}}$ for some constant $c_0 > 1$, respectively. Assume that:

B.1.1 $\log(1/\underline{\pi}_\theta^{(j)}) = O(H_{nj}\log n + \log \overline{L}_{nj})$;

B.1.2 $\pi_\theta^{(j)}([-\delta_{nj}, \delta_{nj}]) \geq 1 - \frac{1}{J_{nj}}\exp\Big[-S_0^{(j)}\{(H_{n1} + H_{n2})\log n$
$+ \log \overline{L}_{n1} + \log \overline{L}_{n2} + \log p_n\}\Big]$

for some constant $S_0^{(j)} > 2$;

$$\pi_\theta^{(j)}([-\delta'_{nj}, \delta'_{nj}]) \geq 1 - \frac{1}{J_{nj}};$$

B.1.3 $-\log\left\{J_{nj}\pi_\theta^{(j)}(|\theta_1^{(j)}| > I_n^{(j)})\right\} \succ (2 + \epsilon_3^{(j)})n\epsilon_n^2$ for some positive constant $\epsilon_3^{(j)}$.

B.2 For $k \in [K]$, let $\overline{B}_n^{(k)}$ and $\underline{B}_n^{(k)}$ denote the largest and the smallest eigenvalues of $\boldsymbol{\Sigma}_\beta^{(k)}$, respectively, and for $r \in [R]$, let $\overline{A}_n^{(r)}$ and $\underline{A}_n^{(r)}$ denote the largest and the smallest eigenvalues of $\boldsymbol{\Sigma}_\alpha^{(r)}$, respectively. Assume that for large enough $n$,

B.2.1 $\overline{A}_n^{(r)} \leq S_1^{(1)}M^{(1)q_1}$ and $\underline{A}_n^{(r)} \geq S_1^{(1)}\{\log M^{(1)}\}^{-1}$ for some positive constants $S_1^{(1)}$, $S_2^{(1)}$ and $q_1$.

B.2.2 $\overline{B}_n^{(k)} \leq S_1^{(2)}M^{(2)q_1}$ and $\underline{B}_n^{(k)} \geq S_2^{(2)}\{\log M^{(2)}\}^{-1}$ for some positive constants $S_1^{(2)}$, $S_2^{(2)}$ and $q_2$;

## A.4 Remark

1. Assumption A.1 specifies that the hypothesis set we consider is a class of DNNs, which is a common setting in the literature [61, 35].

2. Assumption A.2 ensures that the underlying noise transition probability can be approximated by a sparse model. Existing works [62–64] empirically show that large DNNs often contain a large number of redundant parameters and propose methods for compressing neural networks without affecting performance. Moreover, theoretical works [65, 66] in approximation theory provide theories that guarantee uniform approximation rates for a broad family of function classes. Similar assumptions can be found in [35, 67].

3. Assumption A.3 specifies the constraints on the prior distribution we use. As in our experiments in Section 5, we may employ the spike-and-slab prior $\lambda_n \mathcal{N}(0, \sigma_{1n}^2) + (1 - \lambda_n)\mathcal{N}(0, \sigma_{0n}^2)$ for each element of the parameter vector $\boldsymbol{\theta}^{(j)}$ of the sparse Bayesian DNN $\psi_j(\mathrm{x}; \boldsymbol{\theta}^{(j)})$, with $j = 1, 2$; we take the normal prior $\mathcal{N}(0, \sigma_n^2)$ for each element of the regression weights. It can be verified that Condition B.1 in A.3 is satisfied if the values of $\lambda_n$, $\sigma_{1n}$, and $\sigma_{0n}$ are properly chosen [35]. In particular, the value of $\lambda_n$ is related to the sparsity of the model and we require it to satisfy that $\lambda_n = O(1/J_{nj}[n^{H_{n1}+H_{n2}}(\overline{L}_{n1} + \overline{L}_{n2})p_n)]^c)$ for some positive constant $c$ and $j = 1, 2$, which should be chosen by considering the network structure and the number of data points, $n$. Moreover, by using techniques such as Mill's ratio [68], Condition B.2 in A.3 is satisfied if $c_1 < \sigma_n < c_2$ for some positive constants $c_1$ and $c_2$, which are related to $S_1^{(j)}$, $S_2^{(j)}$, $q_j$ and $M^{(j)}$ for $j = 1, 2$ in Condition B.2 [60, 67]. Similar assumptions can be found in [60, 35, 67].

## B Proofs

In this section, we present the proofs of Theorem 1, Theorem 2, and Corollary 3. Specifically, we provide all the need preliminaries in Section B.1, where the definitions and results in Sections B.1.1-B.1.4 will be used in the proof of Theorem 1, and the information-theoretical definitions and lemmas provided in Section B.1.5 will be utilized in the proof of Theorem 2 and Corollary 3.

### B.1 Preliminaries

#### B.1.1 Definitions of some discrepancy measures

**Definition 1** ([69]). Let $f$ and $f_0$ denote two conditional probability density/mass functions of $\tilde{\mathrm{y}}$, given $\mathrm{x}$. Let $\nu_1(d\mathrm{x})$ denote the probability measure for $\mathrm{x}$ associated with the density $f(\mathrm{x})$ and let $\nu_2(d\tilde{\mathrm{y}}) = \bigotimes_{r=1}^R \nu_{2,r}(d\tilde{\mathrm{y}}^{(r)})$ be a dominating measure for $f$ and $f_0$, and hence, a dominating measure of $(\mathrm{x}, \tilde{\mathrm{y}})$ is taken as the product $\nu_1(d\mathrm{x})\nu_2(d\tilde{\mathrm{y}})$.

(i) The Hellinger distance between $f$ and $f_0$ is defined as

$$d(f, f_0) = \sqrt{\int \int (\sqrt{f} - \sqrt{f_0})^2 \nu_2(d\tilde{\mathrm{y}})\nu_1(d\mathrm{x})}.$$

(ii) For any $t > 0$, define

$$d_t(f, f_0) = \frac{1}{t}\left\{\int \int f_0\left(\frac{f_0}{f}\right)^t \nu_2(d\tilde{\mathrm{y}})\nu_1(d\mathrm{x}) - 1\right\}.$$

(iii) The Kullback-Leibler divergence between $f$ and $f_0$ is defined as

$$d_0(f, f_0) \triangleq K(f, f_0) = \int \int f_0 \log\left(\frac{f_0}{f}\right)\nu_2(d\tilde{\mathrm{y}})\nu_1(d\mathrm{x}).$$

(iv) For $q > 1$, define

$$V_q(f, f_0) = \int \int f_0\left|\log\left(\frac{f_0}{f}\right)\right|^q \nu_2(d\tilde{\mathrm{y}})\nu_1(d\mathrm{x}).$$

For $q = 2$, the index of $V_2(\cdot, \cdot)$ is omitted and the discrepancy measure is denoted $V(\cdot, \cdot)$.

[60] shows that (1) $d(f, f_0) \leq \sqrt{d_0(f, f_0)}$; (2) $d_t(f, f_0)$ decreases to $d_0(f, f_0)$ as $t$ decreases to 0.

### B.1.2 Mathematical details about the regression weights

Adapting the proof in [69], we prove the following proposition.

**Proposition 1.** *Assume that $\boldsymbol{\beta} \sim \mathcal{N}(\mathbf{0}, \boldsymbol{D}_{\boldsymbol{\beta}})$, where $\boldsymbol{D}_{\boldsymbol{\beta}}$ is a positive definite matrix. Then, for any given $\dim(\boldsymbol{\beta}) \times 1$ vector of functions $\phi(\mathbf{x}; \boldsymbol{\theta})$ of $\mathbf{x}$ and $\boldsymbol{\theta}$, and for any constant $\Delta > 0$,*

$$P\left\{|(\phi(\mathbf{x}; \boldsymbol{\theta})^{\mathrm{T}}\boldsymbol{\beta} - \phi(\mathbf{x}; \boldsymbol{\theta})^{\mathrm{T}}\boldsymbol{\beta}_0| < \Delta\right\}$$

$$> 8\exp\left\{-\frac{\{\phi(\mathbf{x}; \boldsymbol{\theta})^{\mathrm{T}}\boldsymbol{\beta}_0\}^2 + \Delta^2}{2\underline{B}\|\phi(\mathbf{x}; \boldsymbol{\theta})\|^2}\right\}\frac{\Delta^4}{\overline{B}^2\|\phi(\mathbf{x}; \boldsymbol{\theta})\|^4},$$

*where $\overline{B}$ and $\underline{B}$ are the largest and the smallest eigenvalues of $\boldsymbol{D}_{\boldsymbol{\beta}}$, respectively.*

*Proof.* The proof is established in two steps.

**Step 1.** We first prove that

$$P\left\{|(\phi(\mathbf{x}; \boldsymbol{\theta})^{\mathrm{T}}\boldsymbol{\beta} - \phi(\mathbf{x}; \boldsymbol{\theta})^{\mathrm{T}}\boldsymbol{\beta}_0| < \Delta\right\} > P(X - Y \geq 2), \tag{15}$$

where $X \sim Pois(\frac{\Delta_1}{2})$ and $Y \sim Pois(\frac{\lambda}{2})$, with $\Delta_1 = \frac{\Delta^2}{\phi(\mathbf{x};\boldsymbol{\theta})^{\mathrm{T}}\boldsymbol{D}_{\boldsymbol{\beta}}\phi(\mathbf{x};\boldsymbol{\theta})}$ and $\lambda = \frac{\{\phi(\mathbf{x};\boldsymbol{\theta})^{\mathrm{T}}\boldsymbol{\beta}_0\}^2}{\phi(\mathbf{x};\boldsymbol{\theta})^{\mathrm{T}}\boldsymbol{D}_{\boldsymbol{\beta}}\phi(\mathbf{x};\boldsymbol{\theta})}$, and $X$ and $Y$ are independent.

By the definition of the noncentral chi-squared distribution, it can be easily seen that $T \triangleq \frac{\{\phi(\mathbf{x};\boldsymbol{\theta})^{\mathrm{T}}\boldsymbol{\beta} - \phi(\mathbf{x};\boldsymbol{\theta})^{\mathrm{T}}\boldsymbol{\beta}_0\}^2}{\phi(\mathbf{x};\boldsymbol{\theta})^{\mathrm{T}}\boldsymbol{D}_{\boldsymbol{\beta}}\phi(\mathbf{x};\boldsymbol{\theta})}$ is distributed according to the noncentral chi-squared distribution $\chi_1^2(\lambda)$. Thus, by utilizing the cumulative distribution function (CDF) of $\chi_1^2(\lambda)$, we obtain that

$$P\left\{|(\phi(\mathbf{x}; \boldsymbol{\theta})^{\mathrm{T}}\boldsymbol{\beta} - \phi(\mathbf{x}; \boldsymbol{\theta})^{\mathrm{T}}\boldsymbol{\beta}_0| < \Delta\right\}$$
$$= P\left\{T < \Delta_1\right\}$$
$$= \sum_{j=0}^{\infty}\frac{\exp\left(-\frac{\lambda}{2}\right)\left(\frac{\lambda}{2}\right)^j}{j!}Q(\Delta_1; 1 + 2j),$$

where $Q(\cdot; 1 + 2j)$ is the CDF of $\chi_{1+2j}^2$, the central chi-squared distribution with $1 + 2j$ degrees of freedom.

Noting that $Q(\Delta_1; 1 + 2j) > Q(\Delta_1; 2 + 2j)$, by the result that $Z_{j1} + ... + Z_{j,j+1} \sim \chi_{2+2j}^2$ if $Z_{j1}, ..., Z_{j,j+1} \overset{\text{i.i.d.}}{\sim} \chi_2^2 \overset{\text{d}}{=} \exp(\frac{1}{2})$, we obtain that $Q(\Delta_1; 1 + 2j) > P\{Z_{j1} + ... + Z_{j,j+1} < \Delta_1\}$. According to the relationship between the Poisson counting process and exponential variables, we have that the counting process with $Z_{j1}, ..., Z_{j,j+1}$ as inter-arrival times is the Poisson process with rate $\frac{1}{2}$. Let $N(\Delta_1)$ denote the total number of occurrences or events that have happened up to time $\Delta_1$. Then, $N(\Delta_1)$ follows the Poisson distribution with parameter $\frac{\Delta_1}{2}$, i.e., $N(\Delta_1) \overset{\text{d}}{=} X$, and $\{Z_{j1} + ... + Z_{j,j+1} < \Delta_1\} = \{N(\Delta_1) > j + 1\}$. Consequently, we have that

$$\sum_{j=0}^{\infty}\frac{\exp\left(-\frac{\lambda}{2}\right)\left(\frac{\lambda}{2}\right)^j}{j!}Q(\Delta_1; 1 + 2j) > \sum_{j=0}^{\infty}\frac{\exp\left(-\frac{\lambda}{2}\right)\left(\frac{\lambda}{2}\right)^j}{j!}P\{Z_{j1} + ... + Z_{j,j+1} < \Delta_1\}$$

$$= \sum_{j=0}^{\infty}\frac{\exp\left(-\frac{\lambda}{2}\right)\left(\frac{\lambda}{2}\right)^j}{j!}P\{N(\Delta_1) > j + 1\}$$

$$= \sum_{j=0}^{\infty}\frac{\exp\left(-\frac{\lambda}{2}\right)\left(\frac{\lambda}{2}\right)^j}{j!}P\{X \geq j + 2\}$$

$$= \sum_{j=0}^{\infty}\mathbb{P}(Y = j)\mathbb{P}(X \geq j + 2)$$

$$= \sum_{j=0}^{\infty}\mathbb{P}(X \geq j + 2, Y = j)$$

$$= P\{X - Y \geq 2\},$$

where we use the distribution assumptions and the independence assumption for $X$ and $Y$. Thus, (15) is proved.

**Step 2.** Since $X \sim Pois(\frac{\Delta_1}{2})$ and $Y \sim Pois(\frac{\lambda}{2})$, and they are independent, we have that $X - Y$ follows the Poisson difference distribution (aka the Skellam distribution): for $k = 0, \pm 1, \pm 2, ...$,

$$P\{X - Y = k\} = \exp\left(-\frac{\Delta_1 + \lambda}{2}\right)\left(\frac{\Delta_1}{\lambda}\right)^{\frac{k}{2}} I_k(\sqrt{\Delta_1 \lambda}),$$

where $I_\nu(z) = \sum_{m=0}^{\infty} \frac{(z/2)^{\nu+2m}}{m!\Gamma(\nu+m+1)}$ is the modified Bessel function of the first kind [70]. Using the fact that $I_\nu(z) > z^\nu 2^\nu \Gamma(\nu + 1)$ for $z > 0$ [71] and plugging in $\Delta_1$ and $\lambda$, we obtain that

$$P\{X - Y \geq 2\} > P\{X - Y = 2\} > 8\exp\left(-\frac{\Delta_1 + \lambda}{2}\right)\Delta_1^2$$

$$=8\exp\left\{-\frac{\{\phi(\mathbf{x};\boldsymbol{\theta})^\mathrm{T}\boldsymbol{\beta}_0\}^2 + \Delta^2}{2\phi(\mathbf{x};\boldsymbol{\theta})^\mathrm{T}\boldsymbol{D_\beta}\phi(\mathbf{x};\boldsymbol{\theta})}\right\} \cdot \frac{\Delta^4}{\{\phi(\mathbf{x};\boldsymbol{\theta})^\mathrm{T}\boldsymbol{D_\beta}\phi(\mathbf{x};\boldsymbol{\theta})\}^2}$$

$$\geq 8\exp\left\{-\frac{\{\phi(\mathbf{x};\boldsymbol{\theta})^\mathrm{T}\boldsymbol{\beta}_0\}^2 + \Delta^2}{2\underline{B}\|\phi(\mathbf{x};\boldsymbol{\theta})\|^2}\right\} \cdot \frac{\Delta^4}{\overline{B}^2\|\phi(\mathbf{x};\boldsymbol{\theta})\|^4},$$

where the last inequality follows from the fact that for a symmetric matrix $D$, $\sup_{\|u\|=1} u^\mathrm{T}Du = \lambda_{max}(D)$ and $\inf_{\|u\|=1} u^\mathrm{T}Du = \lambda_{min}(D)$, where $\lambda_{max}(D)$ and $\lambda_{min}(D)$ represent the largest and the smallest eigenvalues of $D$, respectively. This completes the proof. $\qquad\square$

### B.1.3 Mathematical details about the sparse Bayesian DNNs

Consider a DNN with network architecture $(H_n, \boldsymbol{L})$, where $H_n - 1$ is the number of hidden layers, and $\boldsymbol{L} = (L_0, L_1, ..., L_{H_n})^\mathrm{T}$ is the width vector with $L_0 = p_n$ for the input layer and $L_{H_n} = M$ for the output layer. For the corresponding indicator vector $\boldsymbol{\gamma}$ and for $h = 1, ..., H_n$, let $r_h$ denote the number of nonzero connections to the $h$th hidden layer which includes the bias for the $h$th hidden layer and the weights between the $(h-1)$th and the $h$th layer, such that $\sum r_h = |\boldsymbol{\gamma}|_1 \doteq r_n$. For a parameter vector $\boldsymbol{\theta}$, let $O_{h,j}(\boldsymbol{\theta}, \mathbf{x})$ represent the output value of the $j$th node of the $h$th hidden layer for $j = 1, ..., L_h$.

**Lemma 1** ([37], Lemma S1). *Consider a sparse DNN with parameter vector $\boldsymbol{\theta} = (\theta_1, ..., \theta_q)^\top$ with dimension $q$, the corresponding indicator vector $\boldsymbol{\gamma}$ and the network architecture as mentioned above. Suppose that Conditions A.1-A.2 in Section A.2 are satisfied and $|\boldsymbol{\theta}|_\infty \leq E_n$ for a positive constant $E_n$. Then, for $1 \leq h \leq H_n$, the summation of the outputs of the $h$th layer is upper bounded by*

$$\sum_{j=1}^{L_h} O_{h,j}(\boldsymbol{\theta}, \mathbf{x}) \leq E_n^h \prod_{k=1}^h r_k.$$

**Lemma 2** ([37], Lemma S2). *Consider a sparse DNN $\phi(\boldsymbol{\theta}, \mathbf{x})$ with indicator vector $\boldsymbol{\gamma}$ satisfying the conditions in Lemma 1, and a DNN $\phi(\bar{\boldsymbol{\theta}}, \mathbf{x})$ with $\bar{\boldsymbol{\theta}} \in \mathcal{A}$, where $\mathcal{A}$ is defined as*

$$\mathcal{A} = \{\tilde{\boldsymbol{\theta}} = (\tilde{\theta}_1, ..., \tilde{\theta}_q)^\top : |\theta_j - \tilde{\theta}_j| < \delta_1 \text{ for } j \in \boldsymbol{\gamma} \text{ and } |\theta_j - \tilde{\theta}_j| < \delta_2 \text{ for } j \notin \boldsymbol{\gamma}\}$$

*for given $\delta_1 > 0$ and $\delta_2 > 0$. Then*

$$\max_{|\mathbf{x}|_\infty \leq 1} \|\phi(\boldsymbol{\theta}, \mathbf{x}) - \phi(\bar{\boldsymbol{\theta}}, \mathbf{x})\|_1 \leq \delta_1 H_n (E_n + \delta_1)^{H_n-1} \prod_{h=1}^{H_n} r_h$$

$$+ \delta_2 \left(p_n L_1 + \sum_{h=1}^{H_n} L_h\right) \prod_{h=1}^{H_n} \{(E_n + \delta_1)r_h + \delta_2 L_h\}.$$

### B.1.4 A useful lemma

Assume $\mathcal{P}_n$ is a sequence of sets of probability densities and $\mathcal{P}_n^c$ denotes the complement of $\mathcal{P}_n$ for each $n$. Let $\epsilon_n$ be a sequence of positive numbers. An $\epsilon_n$-*cover* of $\mathcal{P}_n$ with respect to (w.r.t.)

distance $d$ is a set $\{f_1, ..., f_k\} \subset \mathcal{P}_n$ such that for each $f \in \mathcal{P}_n$, there exists $j \in \{1, ..., k\}$ such that $d(f, f_j) \leq \epsilon_n$. The $\epsilon_n$-*covering number* is the cardinality of the smallest $\epsilon_n$-cover [72]. Let $N(\epsilon_n, \mathcal{P}_n, d)$ denote the $\epsilon_n$-covering number of $\mathcal{P}_n$ w.r.t. the distance $d$.

Consider a vector of independently (not necessarily identically) distributed observations $\mathcal{D}^n$, where the $i$th component is generated from distribution $\mathbb{P}_{\boldsymbol{\theta},i}$ with the density $p_{\boldsymbol{\theta},i}$ relative to a $\sigma$-finite measure $\nu_i$ on $(\mathcal{X}_i, \mathcal{A}_i)$ for $i \in [n]$, and $\boldsymbol{\theta}$ is the vector of parameters in the parameter space $\Theta$. We define $\mathbb{P}_{\boldsymbol{\theta}}^n$ to be the product measure $\bigotimes_{i=1}^n \mathbb{P}_{\boldsymbol{\theta},i}$ on the corresponding product measurable space $\bigotimes_{i=1}^n (\mathcal{X}_i, \mathcal{A}_i)$. Assume that $\mathcal{D}^n$ is generated from the true distribution $\mathbb{P}_{\boldsymbol{\theta}_0}^n$. We define the square of the *semimetric* $d_n$ as in [73]:

$$d_n^2(\boldsymbol{\theta}, \boldsymbol{\theta}_0) = \frac{1}{n} \sum_{i=1}^n \int (\sqrt{p_{\boldsymbol{\theta},i}} - \sqrt{p_{\boldsymbol{\theta}_0,i}})^2 d\nu_i, \tag{16}$$

which can be seen as the average of the squares of the Hellinger distances. For $\epsilon > 0$, we define the $\epsilon$-neighborhood around $\boldsymbol{\theta}_0$:

$$G_n^*(\boldsymbol{\theta}_0, \epsilon) = \left\{ \boldsymbol{\theta} : \frac{1}{n} \sum_{i=1}^n K_i(\boldsymbol{\theta}, \boldsymbol{\theta}_0) \leq \epsilon^2; \frac{1}{n} \sum_{i=1}^n V_i(\boldsymbol{\theta}, \boldsymbol{\theta}_0) \leq \epsilon^2 \right\}, \tag{17}$$

where $K_i(\boldsymbol{\theta}, \boldsymbol{\theta}_0) = K(\mathbb{P}_{\boldsymbol{\theta},i}, \mathbb{P}_{\boldsymbol{\theta}_0,i})$ and $V_i(\boldsymbol{\theta}, \boldsymbol{\theta}_0) = V(\mathbb{P}_{\boldsymbol{\theta},i}, \mathbb{P}_{\boldsymbol{\theta}_0,i})$, defined in Definition 1.

Let $\Pi(\cdot)$ denote the prior probability measure on $\boldsymbol{\theta}$, and let $\Pi(\cdot|\mathcal{D}^n)$ represent the associated posterior measure given the data $\mathcal{D}^n$. For ease of exposition, we put $\Pi(\Theta^*)$ for $\Pi(\{\theta \in \Theta^*\})$ for any $\Theta^* \subset \Theta$. The following lemma is modified from Theorem 4 in [73] and will be used in the proof of Theorem 1.

**Lemma 3.** *Suppose that for a sequence of sets $\Theta_n \subset \Theta$ and for a sequence of positive numbers $\{\epsilon_n\}_{n=1}^\infty$ such that $\epsilon_n \to 0$ as $n \to \infty$ and $n\epsilon_n^2$ is bounded away from zero, the following conditions hold for large enough $n$:*

*(a) $\sup_{\epsilon > \epsilon_n} \log N(\epsilon/36, \{\boldsymbol{\theta} \in \Theta_n : d_n(\boldsymbol{\theta}, \boldsymbol{\theta}_0) < \epsilon\}, d_n) \leq n\epsilon_n^2$;*

*(b) $\Pi(\Theta \backslash \Theta_n) = o(\exp\{-(r+2)n\epsilon_n^2\})$;*

*(c) $\Pi(G_n^*(\boldsymbol{\theta}_0, \epsilon_n)) \geq \exp(-rn\epsilon_n^2)$*

*for some constant $r > 0$. Then for any $M_n > 0$ $\mathbb{P}_{\boldsymbol{\theta}_0}^n \Pi(\boldsymbol{\theta} : d_n(\boldsymbol{\theta}, \boldsymbol{\theta}_0) \geq M_n \epsilon_n | \mathcal{D}^n) \to 0$ as $M_n \to \infty$.*

### B.1.5 Useful information-theoretical definitions and lemmas

We give some useful information-theoretical definitions and lemmas in this subsection, which will be used in the proof of Theorem 2 and Corollary 3.

**Lemma 4** (Log sum inequality, [45]). *For positive numbers, $a_1, .., a_n$ and $b_1, ..., b_n$,*

$$\sum_{i=1}^n a_i \log \left( \frac{a_i}{b_i} \right) \geq \left( \sum_{i=1}^n a_i \right) \log \left( \frac{\sum_{i=1}^n a_i}{\sum_{i=1}^n b_i} \right),$$

*with equality if and only if $\frac{a_i}{b_i}$ equals for $i = 1, ..., n$.*

**Definition 2** ([74, 75]). *Let $P$ and $Q$ be probability distributions on the set $\mathcal{X}$, and let $f : \mathbb{R}_+ \longrightarrow \mathbb{R}$ be a convex function satisfying $f(1) = 0$. Without loss of generality, assume that $P$ and $Q$ are absolutely continuous with respect to the base measure $\mu$. The $f$-divergence between $P$ and $Q$ is then defined as*

$$D_f(P\|Q) := \int_{\mathcal{X}} q(x) f \left( \frac{p(x)}{q(x)} \right) d\mu(x) + f'(\infty) P\{q = 0\},$$

*where $p$ and $q$ are the densities of $P$ and $Q$ with respect to the measure $\mu$, respectively, and $f'(\infty)$ represents $\lim_{x \to \infty} f(x)/x$.*

**Example 1.** By taking different $f$ functions, we provide some popular examples of $f$-divergences.

- Kullback-Leibler (KL) divergence: taking $f(t) = t \log t$ gives $D_f(P\|Q) \triangleq D_{\text{KL}}(P\|Q) = \int p \log(p/q) d\mu$, which is also denoted $d_0(p, q)$ in Definition 1.

- The total variation distance: taking $f(t) = \frac{1}{2}|t - 1|$ yields $D_f(P\|Q) \triangleq \|P - Q\|_{\mathrm{TV}} = \frac{1}{2}\int \left|\frac{p}{q} - 1\right| q d\mu = \sup_{A \subset \mathcal{X}} |P(A) - Q(A)|$, which is also denoted $d_0(p, q)$ in Definition 1.

- The Hellinger distance: taking $f(t) = (\sqrt{t} - 1)^2 = t - 2\sqrt{t} + 1$ leads to the squared Hellinger distance $D_f(P\|Q) \triangleq H^2(P\|Q) = \int(\sqrt{p} - \sqrt{q})^2 d\mu$, which is also denoted $d^2(p, q)$ in Definition 1.

- The $\chi^2$-divergence: taking $f(t) = \frac{1}{2}(t - 1)^2$ produces the $\chi^2$-divergence $D_f(P\|Q) \triangleq \chi^2(P\|Q) = \frac{1}{2}\int(\frac{p}{q} - 1)^2 d\mu$.

**Lemma 5** ([75])**.** *For the quantities defined in Example 1, the following relationships hold:*

*(i) For the Hellinger distance,*

$$\frac{1}{2}H^2(P, Q) \le \|P - Q\|_{TV} \le H(P, Q)\sqrt{1 - H^2(P, Q)/4}.$$

*(ii) Pinsker's inequality: for any distributions P and Q,*

$$\|P - Q\|_{TV}^2 \le \frac{1}{2}D_{KL}(P\|Q).$$

**Definition 3** ([41, 40])**.** Let $\mathcal{P} = \{P_\theta : \theta \in \Theta\}$ be a family of probability measures on a space $\mathcal{X}$, indexed by $\theta \in \Theta$, and let $\omega$ be a probability measure on $\Theta$. For each $f$ satisfying the conditions in Definition 2, the $f$-informativity, $I_f(\omega, \mathcal{P})$, is defined as

$$I_f(\omega, \mathcal{P}) := \inf_Q \int D_f(P_\theta\|Q)\omega(d\theta),$$

where the infimum is taken over all possible probability measures $Q$ on $\mathcal{X}$. In particular, when $f(t) = t \log t$, the $f$-informativity is equal to the mutual information and is denoted by $I(\omega, \mathcal{P})$.

For each $f$ satisfying the conditions in Definition 2, let $\phi_f : [0, 1]^2 \to \mathbb{R}$ be the function defined as follows: for $a, b \in [0, 1]^2$, $\phi_f(a, b)$ is the $f$-divergence between the two probability measures $P$ and $Q$ on $\{0, 1\}$ given by $P\{1\} = a$ and $Q\{1\} = b$. Then, $\phi_f(a, b)$ has the following expression:

$$\phi_f(a, b) = \begin{cases} bf\left(\frac{a}{b}\right) + (1 - b)f\left(\frac{1 - a}{1 - b}\right) & \text{for } 0 < b < 1; \\ f(1 - a) + af'(\infty) & \text{for } b = 0; \\ f(a) + (1 - a)f'(\infty) & \text{for } b = 1. \end{cases} \tag{18}$$

The following lemma implies monotonicity and convexity properties of $\phi_f$ by taking it as a univariate function with one argument of $\phi_f$ fixed at a given value.

**Lemma 6** ([40])**.** *For each $f$ satisfying the conditions in Definition 2, consider $\phi_f$ defined in (18). Then*

*(a). for every fixed $b > 0$, the map $g(a) : a \mapsto \phi_f(a, b)$ is non-increasing for $a \in [0, b]$ and $g(a)$ is convex and continuous in $a$;*

*(b). for every fixed $a < 1$, the map $h(b) : b \mapsto \phi_f(a, b)$ is non-decreasing for $b \in [a, 1]$.*

**Lemma 7** ([40])**.** *Let $\mathcal{P} = \{P_\theta : \theta \in \Theta\}$ be a family of probability measures on a space $\mathcal{X}$ and let $\omega$ be a probability measure on $\Theta$. Suppose that the loss function $\mathcal{L}$ is zero-one valued. Define the Bayes error as $\Re_{Bayes}(\omega) = \inf_{\mathfrak{d}} \int_\Theta \mathbb{E}\mathcal{L}(\theta, \mathfrak{d}(X))\omega(d\theta)$ with $\mathfrak{d} : \mathcal{X} \to \Theta$ denoting a mapping from the sample space to the parameter space. For any $f$ satisfying the conditions in Definition 2, we have that*

$$I_f(\omega, \mathcal{P}) \ge \phi_f(\Re_{Bayes}(\omega), \Re_0),$$

*where $\phi_f$ is given by (18), and $\Re_0$ is defined as $\Re_0 \triangleq \inf_{a \in \Theta} \int_\Theta \mathcal{L}(\theta, a)\omega(d\theta)$.*

## B.2 Proof of Theorem 1

We now establish the proof of Theorem 1 by checking the three conditions in Lemma 3 established for the sparse Bayesian deep learning framework. We first provide a useful result on the discrepancy measures for generalized linear models in Section B.2.1, and then verify the three conditions in Sections B.2.2-B.2.4 following the proof techniques in [37, 67].

### B.2.1 Discrepancy measures for generalized linear models

Paired variables in the set of anchor points $\overline{\mathcal{D}}_0 = \overline{\mathcal{D}}_{0,1} \cup \ldots \cup \overline{\mathcal{D}}_{0,K}$ in Section 3.1 can be seen as independently (not necessarily identically) distributed, and we let $\mathcal{D}^n$ denote $\overline{\mathcal{D}}_0$ in the following derivations to emphasize its dependence on the sample size $n$. For $\{\mathbf{x}, \tilde{\mathbf{y}}\} \in \overline{\mathcal{D}}_{0,k}$ with $k \in [K]$, we write the $r$th element of $\tilde{\mathbf{y}}$, $\tilde{y}^{(r)}$, in the 1-of-$K$ fashion, i.e., only the $j$th element is equal to 1 while others are all 0 if $\tilde{y}^{(r)}$ is the $j$th class, and then, the conditional probability density/mass function of $\tilde{y}^{(r)}$ induced by $\boldsymbol{\theta}$ is given by

$$f_{\boldsymbol{\theta}}^{(k,r)}(\tilde{y}^{(r)}) = G^{\top}(\boldsymbol{\omega}^{(k,r)})\tilde{y}^{(r)} = \left( G_1(\boldsymbol{\omega}^{(k,r)}), \ldots, (G_K(\boldsymbol{\omega}^{(k,r)}) \right)\tilde{y}^{(r)} \tag{19}$$

for $r \in [R]$. We denote the joint conditional probability density/mass function of $\tilde{\mathbf{y}}$ as $f_{\boldsymbol{\theta}}^{(k)}(\tilde{\mathbf{y}})$, given by

$$f_{\boldsymbol{\theta}}^{(k)}(\tilde{\mathbf{y}}) = \prod_{r=1}^{R} f_{\boldsymbol{\theta}}^{(k,r)}(\tilde{y}^{(r)}), \tag{20}$$

and in contrast, we denote the underlying true conditional probability density/mass functions for $\tilde{y}^{(r)}$ and $\tilde{\mathbf{y}}$ as $f_0^{(k,r)}$ and $f_0^{(k)}$, respectively.

**Result 1.** *If Conditions A.4 in Section A.2 are satisfied, then, for any $k \in [K]$,*

$$K(f_{\boldsymbol{\theta}}^{(k)}, f_0^{(k)}) \leq C_K \sum_{r=1}^{R} \mathbb{E}_{\mathbf{x}}\|\boldsymbol{\omega}^{(k,r)} - \boldsymbol{\omega}_0^{(k,r)}\|_1; \tag{21}$$

$$V(f_{\boldsymbol{\theta}}^{(k)}, f_0^{(k)}) \leq C_V \sum_{r=1}^{R} \mathbb{E}_{\mathbf{x}}\|\boldsymbol{\omega}^{(k,r)} - \boldsymbol{\omega}_0^{(k,r)}\|_1, \tag{22}$$

*for some positive constants $C_K$ and $C_V$ in the neighbourhood of $\boldsymbol{\theta}_0$[4].*

*Proof.* By Definition (1) (iii) and (20), we have that

$$K(f_{\boldsymbol{\theta}}^{(k)}, f_0^{(k)}) = \int \int f_0^{(k)} \log \left( \frac{f_0^{(k)}}{f_{\boldsymbol{\theta}}^{(k)}} \right) \nu_2(d\tilde{\mathbf{y}})\nu_1(d\mathbf{x})$$

$$= \sum_{r=1}^{R} \int \int f_0^{(k)} \log \left( \frac{f_0^{(k,r)}}{f_{\boldsymbol{\theta}}^{(k,r)}} \right) \nu_2(d\tilde{\mathbf{y}})\nu_1(d\mathbf{x})$$

$$= \sum_{r=1}^{R} \int \int f_0^{(k,r)} \log \left( \frac{f_0^{(k,r)}}{f_{\boldsymbol{\theta}}^{(k,r)}} \right) \nu_{2,r}(d\tilde{y}^{(r)})\nu_1(d\mathbf{x}).$$

---

[4]These two distances can also be upper bounded by the $L_2$ norm of $\boldsymbol{\omega}_2 - \boldsymbol{\omega}_1$ since the $L_1$ norm and $L_2$ norm are equivalent on $\mathbb{R}^p$ in the sense that $|\boldsymbol{\omega}|_2 \leq |\boldsymbol{\omega}|_1 \leq \sqrt{p}|\boldsymbol{\omega}|_2$.

According to (19) and by taking the Taylor's expansion of $\log\left(f_{\boldsymbol{\theta}}^{k,r}\right)$ at $\boldsymbol{\omega}_0^{(k,r)}$, we obtain that

$$
\int f_0^{(k,r)} \log\left(\frac{f_0^{(k,r)}}{f_{\boldsymbol{\theta}}^{(k,r)}}\right) \nu_{2,r}(d\tilde{\mathbf{y}}^{(r)})
$$

$$
= -\int f_0^{(k,r)} \left[\left(\frac{\partial \log f_{\boldsymbol{\theta}}^{(k,r)}}{\partial \omega}\bigg|_{\boldsymbol{\omega}=\overline{\boldsymbol{\omega}}}\right)^{\top} (\boldsymbol{\omega}^{(k,r)} - \boldsymbol{\omega}_0^{(k,r)})\right] \nu_{2,r}(d\tilde{\mathbf{y}}^{(r)})
$$

$$
= -\int f_0^{(k,r)} \left[\left(\frac{\partial \log(G(\boldsymbol{\omega})^{\top}\tilde{\mathbf{y}}^{(r)})}{\partial \omega}\bigg|_{\boldsymbol{\omega}=\overline{\boldsymbol{\omega}}}\right)^{\top} (\boldsymbol{\omega}^{(k,r)} - \boldsymbol{\omega}_0^{(k,r)})\right] \nu_{2,r}(d\tilde{\mathbf{y}}^{(r)})
$$

$$
= -\int f_0^{(k,r)} \left[\left(\frac{1}{G(\overline{\boldsymbol{\omega}})^{\top}} \frac{\partial G(\overline{\boldsymbol{\omega}})^{\top}}{\partial \boldsymbol{\omega}}\right)^{\top} (\boldsymbol{\omega}^{(k,r)} - \boldsymbol{\omega}_0^{(k,r)})\right] \nu_{2,r}(d\tilde{\mathbf{y}}^{(r)}),
$$

where $\overline{\boldsymbol{\omega}} = (\overline{\omega}_1, ..., \overline{\omega}_K)^{\top}$ is between $\boldsymbol{\omega}^{(k,r)}$ and $\boldsymbol{\omega}_0^{(k,r)}$ and

$$
\frac{\partial G(\overline{\boldsymbol{\omega}})^{\top}}{\partial \boldsymbol{\omega}} = \left(\frac{\partial G_1(\overline{\boldsymbol{\omega}})}{\partial \overline{\omega}}, ..., \frac{\partial G_K(\overline{\boldsymbol{\omega}})}{\partial \overline{\omega}}\right) = \begin{pmatrix} \frac{\partial G_1(\overline{\boldsymbol{\omega}})}{\partial \overline{\omega}_1} & \cdots & \frac{\partial G_K(\overline{\boldsymbol{\omega}})}{\partial \overline{\omega}_1} \\ \vdots & & \vdots \\ \frac{\partial G_1(\overline{\boldsymbol{\omega}})}{\partial \overline{\omega}_K} & \cdots & \frac{\partial G_K(\overline{\boldsymbol{\omega}})}{\partial \overline{\omega}_K} \end{pmatrix}.
$$

Since $\tilde{\mathbf{y}}^{(r)}$ is discretely distributed, according to (19) and Condition A.4.1, we further obtain that

$$
\int f_0^{(k,r)} \log\left(\frac{f_0^{(k,r)}}{f_{\boldsymbol{\theta}}^{(k,r)}}\right) \nu_{2,r}(d\tilde{\mathbf{y}}^{(r)})
$$

$$
= -\sum_{j=1}^{K} \frac{G_j(\boldsymbol{\omega}_0^{(k,r)})}{G_j(\overline{\boldsymbol{\omega}})} \left\{\frac{\partial G_j(\overline{\boldsymbol{\omega}})}{\partial \boldsymbol{\omega}}\right\}^{\top} (\boldsymbol{\omega}^{(k,r)} - \boldsymbol{\omega}_0^{(k,r)})
$$

$$
\leq \frac{C_3 K}{1 - \epsilon_2} \|\boldsymbol{\omega}^{(k,r)} - \boldsymbol{\omega}_0^{(k,r)}\|_1.
$$

Thus, the proof of inequality (21) is completed.

Similarly, we write $V(f_{\boldsymbol{\theta}}^{(k)}, f_0^{(k)})$ as:

$$
V(f_{\boldsymbol{\theta}}^{(k)}, f_0^{(k)}) = \int\int f_0^{(k)} \left\{\log\left(\frac{f_0^{(k)}}{f_{\boldsymbol{\theta}}^{(k)}}\right)\right\}^2 \nu_2(d\tilde{\mathbf{y}})\nu_1(d\mathbf{x})
$$

$$
= \int\int f_0^{(k)} \left\{\sum_{r=1}^{R} \log\left(\frac{f_0^{(k,r)}}{f_{\boldsymbol{\theta}}^{(k,r)}}\right)\right\}^2 \nu_2(d\tilde{\mathbf{y}})\nu_1(d\mathbf{x})
$$

$$
= \sum_{r=1}^{R} \int\int f_0^{(k,r)} \left\{\log\left(\frac{f_0^{(k,r)}}{f_{\boldsymbol{\theta}}^{(k,r)}}\right)\right\}^2 \nu_{2,r}(d\tilde{\mathbf{y}}^{(r)})\nu_1(d\mathbf{x})
$$

$$
+ \sum_{r_1 \neq r_2} \int \left[\int f_0^{(k,r_1)} \left\{\log\left(\frac{f_0^{(k,r_1)}}{f_{\boldsymbol{\theta}}^{(k,r_1)}}\right)\right\} \nu_{2,r_1}(d\tilde{\mathbf{y}}^{(r_1)})\right]
$$

$$
\cdot \left[\int f_0^{(k,r_2)} \left\{\log\left(\frac{f_0^{(k,r_2)}}{f_{\boldsymbol{\theta}}^{(k,r_2)}}\right)\right\} \nu_{2,r_2}(d\tilde{\mathbf{y}}^{(r_2)})\right] \nu_1(d\mathbf{x}),
$$

where the second term can be upper bounded by

$$
\left\{\frac{C_3 K}{1 - \epsilon_2}\right\}^2 \sum_{r_1 \neq r_2} \mathbb{E}_{\mathbf{x}} \|\boldsymbol{\omega}^{(k,r_1)} - \boldsymbol{\omega}_0^{(k,r_1)}\|_1 \|\boldsymbol{\omega}^{(k,r_2)} - \boldsymbol{\omega}_0^{(k,r_2)}\|_1,
$$

and for the first term, we have

$$
\int f_0^{(k,r)} \left\{\log\left(\frac{f_0^{(k,r)}}{f_{\boldsymbol{\theta}}^{(k,r)}}\right)\right\}^2 \nu_{2,r}(d\tilde{\mathbf{y}}^{(r)}) = \sum_{j=1}^{K} G_j(\boldsymbol{\omega}_0^{(k,r)}) \left\{\log\left(\frac{G_j(\boldsymbol{\omega}_0^{(k,r)})}{G_j(\boldsymbol{\omega}^{(k,r)})}\right)\right\}^2.
$$

If $G_j(\boldsymbol{\omega}_0^{(k,r)}) \leq \|\boldsymbol{\omega}^{(k,r)}) - \boldsymbol{\omega}_0^{(k,r)})\|_1$, we have that

$$G_j(\boldsymbol{\omega}_0^{(k,r)}) \left\{ \log\left(\frac{G_j(\boldsymbol{\omega}_0^{(k,r)})}{G_j(\boldsymbol{\omega}^{(k,r)})}\right) \right\}^2 \leq \max\{|\log(1-\epsilon_2)|, |\log(1+\epsilon_2)|\} \|\boldsymbol{\omega}^{(k,r)}) - \boldsymbol{\omega}_0^{(k,r)})\|_1$$

according to Condition A.4.1. If $G_j(\boldsymbol{\omega}_0^{(k,r)}) > \|\boldsymbol{\omega}^{(k,r)}) - \boldsymbol{\omega}_0^{(k,r)})\|_1$, similar to the proof for (21), we obtain that

$$
\begin{aligned}
G_j(\boldsymbol{\omega}_0^{(k,r)}) \left\{ \log\left(\frac{G_j(\boldsymbol{\omega}_0^{(k,r)})}{G_j(\boldsymbol{\omega}^{(k,r)})}\right) \right\}^2 &= G_j(\boldsymbol{\omega}_0^{(k,r)}) \left[ \frac{1}{G_j(\overline{\boldsymbol{\omega}})} \left\{ \frac{\partial G_j(\overline{\boldsymbol{\omega}})}{\partial \boldsymbol{\omega}} \right\}^\top (\boldsymbol{\omega}^{(k,r)} - \boldsymbol{\omega}_0^{(k,r)}) \right]^2 \\
&\leq \frac{G_j(\boldsymbol{\omega}_0^{(k,r)})}{\left\{ (1-\epsilon_2) G_j(\boldsymbol{\omega}_0^{(k,r)}) \right\}^2} \cdot \left( C_3 \|\boldsymbol{\omega}^{(k,r)} - \boldsymbol{\omega}_0^{(k,r)}\|_1 \right)^2 \\
&\leq \left( \frac{C_3}{1-\epsilon_2} \right)^2 \|\boldsymbol{\omega}^{(k,r)} - \boldsymbol{\omega}_0^{(k,r)}\|_1,
\end{aligned}
$$

where $\bar{\omega}$ is between $\boldsymbol{\omega}_0^{(k,r)}$ and $\boldsymbol{\omega}^{(k,r)}$. Thus, the inequality (22) holds in the neighbourhood of $\boldsymbol{\theta}_0$. $\qquad\square$

### B.2.2 Verification of Condition (c) in Lemma 3

*Proof.* For the sequence $\{\epsilon_n\}_{n=1}^\infty$ in Lemma 3, by (21) and (22) in Result 1 and (17), we obtain that

$$\Pi(G_n^*(\boldsymbol{\theta}_0, \epsilon_n)) \geq \Pi\left\{ \boldsymbol{\theta} : \mathbb{E}_{\mathbf{x}}\|\boldsymbol{\omega}^{(k,r)} - \boldsymbol{\omega}_0^{(k,r)}\|_1 \leq \Delta_n \text{ for } k \in [K], r \in [R] \right\},$$

where $\Delta_n = U_1 \epsilon_n^2$ for some positive constant $U_1$.

For simplicity of presentation, we now omit $k$ and $r$ in $f_0^{(k,r)}$, $f_{\boldsymbol{\theta}}^{(k,r)}$, $\boldsymbol{\omega}_0^{(k,r)}$, $\boldsymbol{\omega}^{(k,r)}$, $\mathbf{B}^{(k)}$ and $\mathbf{A}^{(r)}$ in this proof here. Note that

$$
\begin{aligned}
\|\boldsymbol{\omega} - \boldsymbol{\omega}_0\|_1 &= \|\mathbf{B}^\top \psi_2(\mathbf{x}; \boldsymbol{\theta}^{(2)}) + \mathbf{A}^\top \psi_1(\mathbf{x}; \boldsymbol{\theta}^{(1)}) - \mathbf{B}_0^\top \psi_2(\mathbf{x}) - \mathbf{A}_0^\top \psi_1(\mathbf{x})\|_1 \\
&\leq \|\mathbf{B}^\top \psi_2(\mathbf{x}; \boldsymbol{\theta}^{(2)}) - \mathbf{B}_0^\top \psi_2(\mathbf{x})\|_1 + \|\mathbf{A}^\top \psi_1(\mathbf{x}; \boldsymbol{\theta}^{(1)}) - \mathbf{A}_0^\top \psi_1(\mathbf{x})\|_1 \\
&\leq \|\mathbf{B}^\top \psi_2(\mathbf{x}; \boldsymbol{\theta}^{(2)}) - \mathbf{B}_0^\top \psi_2(\mathbf{x}; \boldsymbol{\theta}^{(2)})\|_1 + \|\mathbf{B}_0^\top \psi_2(\mathbf{x}; \boldsymbol{\theta}^{(2)}) - \mathbf{B}_0^\top \psi_2(\mathbf{x}; \boldsymbol{\theta}^{(2)*})\|_1 \\
&\quad + \|\mathbf{B}_0^\top \psi_2(\mathbf{x}; \boldsymbol{\theta}^{(2)*}) - \mathbf{B}_0^\top \psi_2(\mathbf{x})\|_1 + \|\mathbf{A}^\top \psi_1(\mathbf{x}; \boldsymbol{\theta}^{(1)}) - \mathbf{A}_0^\top \psi_1(\mathbf{x}; \boldsymbol{\theta}^{(1)})\|_1 \\
&\quad + \|\mathbf{A}_0^\top \psi_1(\mathbf{x}; \boldsymbol{\theta}^{(1)}) - \mathbf{A}_0^\top \psi_1(\mathbf{x}; \boldsymbol{\theta}^{(1)*})\|_1 + \|\mathbf{A}_0^\top \psi_1(\mathbf{x}; \boldsymbol{\theta}^{(1)*}) - \mathbf{A}_0^\top \psi_1(\mathbf{x})\|_1.
\end{aligned}
$$

Corresponding to each term above, we consider the following six terms:

$$
\begin{aligned}
&\text{(I)} : \Pi\{\boldsymbol{\theta} : \mathbb{E}_{\mathbf{x}}\|\mathbf{B}^\top \psi_2(\mathbf{x}; \boldsymbol{\theta}^{(2)}) - \mathbf{B}_0^\top \psi_2(\mathbf{x}; \boldsymbol{\theta}^{(2)})\|_1 \leq \Delta_n/6\}; \\
&\text{(II)} : \Pi\{\boldsymbol{\theta} : \mathbb{E}_{\mathbf{x}}\|\mathbf{B}_0^\top \psi_2(\mathbf{x}; \boldsymbol{\theta}^{(2)}) - \mathbf{B}_0^\top \psi_2(\mathbf{x}; \boldsymbol{\theta}^{(2)*})\|_1 \leq \Delta_n/6\}; \\
&\text{(III)} : \Pi\{\boldsymbol{\theta} : \mathbb{E}_{\mathbf{x}}\|\mathbf{B}_0^\top \psi_2(\mathbf{x}; \boldsymbol{\theta}^{(2)*}) - \mathbf{B}_0^\top \psi_2(\mathbf{x})\|_1 \leq \Delta_n/6\}; \\
&\text{(I)}' : \Pi\{\boldsymbol{\theta} : \mathbb{E}_{\mathbf{x}}\|\mathbf{A}^\top \psi_1(\mathbf{x}; \boldsymbol{\theta}^{(1)}) - \mathbf{A}_0^\top \psi_1(\mathbf{x}; \boldsymbol{\theta}^{(1)})\|_1 \leq \Delta_n/6\}; \\
&\text{(II)}' : \Pi\{\boldsymbol{\theta} : \mathbb{E}_{\mathbf{x}}\|\mathbf{A}_0^\top \psi_1(\mathbf{x}; \boldsymbol{\theta}^{(1)}) - \mathbf{A}_0^\top \psi_1(\mathbf{x}; \boldsymbol{\theta}^{(1)*})\|_1 \leq \Delta_n/6\}; \\
&\text{(III)}' : \Pi\{\boldsymbol{\theta} : \mathbb{E}_{\mathbf{x}}\|\mathbf{A}_0^\top \psi_1(\mathbf{x}; \boldsymbol{\theta}^{(1)*}) - \mathbf{A}_0^\top \psi_1(\mathbf{x})\|_1 \leq \Delta_n/6\}.
\end{aligned}
\tag{23}
$$

Corresponding to $\mathbf{B}^{(k)}$ for (6), we write $\mathbf{B} = (\boldsymbol{\beta}_1, ..., \boldsymbol{\beta}_K)$ and $\mathbf{B}_0 = (\boldsymbol{\beta}_{10}, ..., \boldsymbol{\beta}_{K0})$. Then for (I), we have that

$$
\begin{aligned}
&\Pi\left\{ \boldsymbol{\theta} : \mathbb{E}_{\mathbf{x}}\|\mathbf{B}^\top \psi_2(\mathbf{x}; \boldsymbol{\theta}^{(2)}) - \mathbf{B}_0^\top \psi_2(\mathbf{x}; \boldsymbol{\theta}^{(2)})\|_1 \leq \Delta_n/6 \right\} \\
=&\Pi\left\{ (\boldsymbol{\theta}^{(1)\top}, \mathbf{B}^\top)^\top : \sum_{j=1}^K \mathbb{E}_{\mathbf{x}}|\boldsymbol{\beta}_j^\top \psi_2(\mathbf{x}; \boldsymbol{\theta}^{(2)}) - \boldsymbol{\beta}_{j0}^\top \psi_2(\mathbf{x}; \boldsymbol{\theta}^{(2)})| \leq \Delta_n/6 \right\} \\
\geq&\Pi\left\{ (\boldsymbol{\theta}^{(1)\top}, \mathbf{B}^\top)^\top : \mathbb{E}_{\mathbf{x}}|\boldsymbol{\beta}_j^\top \psi_2(\mathbf{x}; \boldsymbol{\theta}^{(2)}) - \boldsymbol{\beta}_{j0}^\top \psi_2(\mathbf{x}; \boldsymbol{\theta}^{(2)})| \leq \frac{\Delta_n}{6K} \text{ for } j = 1, ..., K \right\} \\
=&\left[ \Pi\left\{ (\boldsymbol{\theta}^{(1)\top}, \boldsymbol{\beta}_1^\top)^\top : \mathbb{E}_{\mathbf{x}}|\boldsymbol{\beta}_1^\top \psi_2(\mathbf{x}; \boldsymbol{\theta}^{(2)}) - \boldsymbol{\beta}_{10}^\top \psi_2(\mathbf{x}; \boldsymbol{\theta}^{(2)})| \leq \frac{\Delta_n}{6K} \right\} \right]^K,
\end{aligned}
$$

where the last equality holds since all columns of $\mathbf{B}$ have independent and identical prior.

Taking $\Delta = \frac{\Delta_n}{6K} = \frac{U_1 \epsilon_n^2}{6K}$, by Proposition 1, we have that for any given nonzero $\psi_2(\mathbf{x}; \boldsymbol{\theta}^{(2)})$, $\mathcal{M}_1 > 0$, and $\mathcal{M}_2 > 0$,

$$
\Pi \left\{ \boldsymbol{\beta}_1 : |\boldsymbol{\beta}_1^\top \psi_2(\mathbf{x}; \boldsymbol{\theta}^{(2)}) - \boldsymbol{\beta}_{10}^\top \psi_2(\mathbf{x}; \boldsymbol{\theta}^{(2)})| \leq \Delta \Big| \psi_2(\mathbf{x}; \boldsymbol{\theta}^{(2)}) \right\}
$$

$$
> \exp \left\{ -\frac{\left\{ \psi_2^\top(\mathbf{x}; \boldsymbol{\theta}^{(2)}) \boldsymbol{\beta}_0 \right\}^2 + \Delta^2}{2\underline{B} \|\psi_2(\mathbf{x}; \boldsymbol{\theta}^{(2)})\|^2} \right\} \frac{8\Delta^4}{\overline{B}^2 \|\psi_2(\mathbf{x}; \boldsymbol{\theta}^{(2)})\|^4}
$$

$$
> \exp \left\{ -\mathcal{M}_1 n \epsilon_n^2 \cdot \frac{F_2^2 \|\psi_2(\mathbf{x}; \boldsymbol{\theta}^{(2)})\|^2 + \Delta^2}{2\mathcal{M}_1 n \epsilon_n^2 \cdot S_2^{(2)} (\log M^{(2)})^{-1} \|\psi_2(\mathbf{x}; \boldsymbol{\theta}^{(2)})\|^2} \right\} \frac{8\Delta^4}{\{S_1^{(2)} M^{(2)q_2}\}^2 \|\psi_2(\mathbf{x}; \boldsymbol{\theta}^{(2)})\|^4}
$$

$$
> \exp \left\{ -\mathcal{M}_1 n \epsilon_n^2 \cdot \frac{(F_2^2 + 1) \log M^{(2)}}{2\mathcal{M}_1 n \epsilon_n^2 \cdot S_2^{(2)}} \right\}
$$

$$
\cdot \exp \left[ -\left\{ 2 \log S_1^{(2)} + 2q_2 \log M^{(2)} + 4 \log \|\psi_2(\mathbf{x}; \boldsymbol{\theta}^{(2)})\| - \log(8\Delta^4) \right\} \right]
$$

$$
\geq \exp\{-\mathcal{M}_1 n \epsilon_n^2\} \cdot \exp\{-\mathcal{M}_2 n \epsilon_n^2\},
$$

for large enough $n$, where the second inequality follows from Condition A.4.2 in Section A.2 and Condition B.2.2 in Section A.3, and the third inequality holds since $\Delta = \frac{\Delta_n}{6K} = \frac{U_1 \epsilon_n^2}{6K}$ by definition and $\epsilon_n \to 0$ as $n \to \infty$. The last inequality holds if $4 \log(1/\epsilon_n^2) \leq \mathcal{M}_2 n \epsilon_n^2$ for large enough $n$ and $\log \|\psi_2(\mathbf{x}; \boldsymbol{\theta}^{(2)})\| \prec \mathcal{M}_2 n \epsilon_n^2$, where, according to Lemma 1, the latter holds if $H_{n2} \left( \log E_{n2} + \log \frac{r_{n2}}{H_{n2}} \right) \prec \mathcal{M}_2 n \epsilon_n^2$, which can be guaranteed by Condition A.3.1 in Section A.2. Since the result above holds for any given $\psi_2(\mathbf{x}; \boldsymbol{\theta}^{(2)})$, by summarizing the discussion above, we obtain that

$$
\text{(I)} \geq \left[ \inf_{\boldsymbol{\theta} \in \mathcal{F}_n} \Pi \left\{ \boldsymbol{\beta}_1 : \mathbb{E}_{\mathbf{x}} |\boldsymbol{\beta}_1^\top \psi_2(\mathbf{x}; \boldsymbol{\theta}^{(2)}) - \boldsymbol{\beta}_{10}^\top \psi_2(\mathbf{x}; \boldsymbol{\theta}^{(2)})| \leq \frac{\Delta_n}{6K} \Big| \psi_2(\mathbf{x}; \boldsymbol{\theta}^{(2)}) \right\} \right]^K
$$

$$
\geq \exp\{-K(\mathcal{M}_1 + \mathcal{M}_2) n \epsilon_n^2\}.
$$

Now we consider the second term (II) in (23), which can be written as

$$
\text{(II)} = \Pi \{ \boldsymbol{\theta}^{(2)} : \mathbb{E}_{\mathbf{x}} \|\mathbf{B}_0^\top \psi_2(\mathbf{x}; \boldsymbol{\theta}^{(2)}) - \mathbf{B}_0^\top \psi_2(\mathbf{x}; \boldsymbol{\theta}^{(2)*})\|_1 \leq \Delta_n/6 \}
$$

$$
= \Pi \left\{ \boldsymbol{\theta}^{(2)} : \sum_{j=1}^K \mathbb{E}_{\mathbf{x}} |\boldsymbol{\beta}_{j0}^\top \psi_2(\mathbf{x}; \boldsymbol{\theta}^{(2)}) - \boldsymbol{\beta}_{j0}^\top \psi_2(\mathbf{x}; \boldsymbol{\theta}^{(2)*})| \leq \Delta_n/6 \right\}
$$

$$
\geq \Pi \left\{ \boldsymbol{\theta}^{(2)} : \mathbb{E}_{\mathbf{x}} \|\psi_2(\mathbf{x}; \boldsymbol{\theta}^{(2)}) - \psi_2(\mathbf{x}; \boldsymbol{\theta}^{(2)*})\|_1 \leq \frac{\Delta_n}{6KF_2} \right\},
$$

where the last inequality follows from Condition A.4.2 in Section A.2.

Consider the set

$$
\mathcal{S}^{(2)} = \{ \boldsymbol{\theta}^{(2)} : |\boldsymbol{\theta}_j^{(2)} - \boldsymbol{\theta}_j^{(2)*}| \leq \eta_n \text{ for } j \in \boldsymbol{\gamma}^{(2)*}; \ |\boldsymbol{\theta}_j^{(2)} - \boldsymbol{\theta}_j^{(2)*}| \leq \eta_n' \text{ for } j \notin \boldsymbol{\gamma}^{(2)*} \}, \qquad (24)
$$

where $\eta_n$ and $\eta_n'$ will be specified later. Let $r_h$ denote the number of nonzero connections to the $h$th hidden layer which includes the bias for the $h$th hidden layer and the weights between the $(h-1)$th and the $h$th layer, such that $\sum r_h = \|\boldsymbol{\gamma}^{(2)*}\|_1 \triangleq r_{n2}$. Then, for any $\mathbf{x}$ satisfying $\|\mathbf{x}\|_\infty \leq 1$,

$$
\|\psi_2(\mathbf{x}; \boldsymbol{\theta}^{(2)}) - \psi_2(\mathbf{x}; \boldsymbol{\theta}^{(2)*})\|_1
$$

$$
\leq \eta_n H_{n2} (E_{n2} + \eta_n)^{H_{n2}-1} \prod_{h=1}^{H_{n2}} r_h + \eta_n' J_{n2} \prod_{h=1}^{H_{n2}} \left( E_{n2} + \eta_n + \frac{\eta_n' L_h^{(2)}}{r_h} \right) r_h
$$

$$
\leq \eta_n H_{n2} (U_2 E_{n2})^{H_{n2}} \left( \frac{\sum r_h}{H_{n2}} \right)^{H_{n2}} + \eta_n' J_{n2} (U_2 E_{n2})^{H_{n2}} \left( \frac{\sum r_h}{H_{n2}} \right)^{H_{n2}}
$$

$$
\leq 2 U_3 \epsilon_n^2,
$$

where the first inequality follows from Lemma 2, and the second and the third inequalities hold if we take $\eta_n = U_3\epsilon_n^2/\{H_{n2}(U_2E_{n2})^{H_{n2}}(r_{n2}/H_{n2})^{H_{n2}}\}$ and $\eta_n' = U_3\epsilon_n^2/\{J_{n2}(U_2E_{n2})^{H_{n2}}(r_{n2}/H_{n2})^{H_{n2}}\}$ for some constants $U_2 > 1$ and $U_3 > 0$. By taking a small enough $U_3$, we can obtain that $\mathbb{E}_{\mathbf{x}}\|\psi_2(\mathbf{x};\boldsymbol{\theta}^{(2)}) - \psi_2(\mathbf{x};\boldsymbol{\theta}^{(2)*})\|_1 \le \frac{\Delta_n}{6KF_1}$ for any $\boldsymbol{\theta}^{(2)} \in \mathcal{S}^{(2)}$ defined in (24). Then, it suffices to prove that $\Pi(\mathcal{S}^{(2)}) = \Pi^{(2)}(\mathcal{S}^{(2)}) \ge \exp(-\mathcal{M}_3 n\epsilon_n^2)$ for some positive constant $\mathcal{M}_3$, where $\Pi^{(2)}(\cdot)$ is the prior measure of $\boldsymbol{\theta}^{(2)}$ as given in Section A.3.

As defined in (5) in Section 3.2, each element of $\boldsymbol{\theta}^{(2)}$ has an independent continuous prior, denoted $\pi_\theta^{(2)}(\cdot)$, and we use $\underline{\pi}_\theta^{(2)}$ to denote the minimal density value of $\pi_\theta^{(2)}(\cdot)$ on the interval $[-E_{n2} - 1, E_{n2} + 1]$. Then, we obtain that

$$
\begin{aligned}
&\Pi^{(2)}(\mathcal{S}^{(2)})\\
&= \prod_{j\in\boldsymbol{\gamma}^{(2)*}} \pi_\theta^{(2)}(\theta_j^{(2)*} - \eta_n \le \theta_j^{(2)} \le \theta_j^{(2)*} + \eta_n) \prod_{j\notin\boldsymbol{\gamma}^{(2)*}} \pi_\theta^{(2)}(\theta_j^{(2)*} - \eta_n' \le \theta_j^{(2)} \le \theta_j^{(2)*} + \eta_n')\\
&\ge (2\underline{\pi}_\theta^{(2)}\eta_n)^{r_{n2}} \left\{\pi_\theta^{(2)}(\theta_1^{(2)} \in [-\eta_n', \eta_n'])\right\}^{J_{n2}},
\end{aligned}
$$

and thus, according to Conditions B.1.1 and B.1.2 in Section A.3, we have that

$$
\begin{aligned}
-\log\Pi^{(2)}(\mathcal{S}^{(2)}) &\le r_{n2}\left\{\log\frac{1}{2} + \log\frac{1}{\underline{\pi}_\theta^{(2)}} + \log\frac{1}{\eta_n}\right\} + J_{n2}\log\left\{\frac{1}{\pi_\theta^{(2)}(\theta_1^{(2)} \in [-\eta_n', \eta_n'])}\right\}\\
&\le r_{n2}\left\{\text{constant} + H_{n2}\log n + \log\overline{L}_{n2} + \log\left(\frac{1}{\epsilon_n^2}\right) + \log H_{n2}\right.\\
&\quad \left. + H_{n2}\log(U_2E_{n2}) + H_{n2}\log\left(\frac{r_{n2}}{H_{n2}}\right)\right\} + J_{n2}\log\left(\frac{J_{n2}}{J_{n2}-1}\right)\\
&\asymp r_{n2}H_{n2}\log n + r_{n2}\log\overline{L}_{n2},
\end{aligned}
$$

where the last asymptotic equality follows from Condition A.3 in Section A.2 and the fact that $\log(1/\epsilon_n) = O(\log n)$. Thus, $(\text{II}) \ge \Pi(\mathcal{S}^{(2)}) \ge \exp(-\mathcal{M}_3 n\epsilon_n^2)$ holds if $r_{n2}H_{n2}\log n + r_{n2}\log\overline{L}_{n2} \le U_4 n\epsilon_n^2$ for some sufficiently small positive constant $U_4$.

Now we consider the third term in (23):

$$
\begin{aligned}
\mathbb{E}_{\mathbf{x}}\|\mathbf{B}_0^\top\psi_2(\mathbf{x};\boldsymbol{\theta}^{(2)*}) - \mathbf{B}_0^\top\psi_2(\mathbf{x})\|_1 &= \sum_{j=1}^K \mathbb{E}_{\mathbf{x}}\left|\boldsymbol{\beta}_{j0}^\top\{\psi_2(\mathbf{x};\boldsymbol{\theta}^{(2)*}) - \psi_2(\mathbf{x})\}\right|\\
&\le KF_2\mathbb{E}_{\mathbf{x}}\left|\psi_2(\mathbf{x};\boldsymbol{\theta}^{(2)*}) - \psi_2(\mathbf{x})\right|\\
&\le KF_2\varpi_{n2},
\end{aligned}
$$

where the second inequality follows from Condition A.4.2 in Section A.2 and the last inequality follows from the definition of the true model given in (14). Thus, we can take $\epsilon_n^2 = O(\varpi_{n2})$ so that (III) $\approx 1$ for large enough $n$.

Similar discussion can be applied to $(\text{I})'$, $(\text{II})'$ and $(\text{III})'$ in (23) and we can obtain that

$$
\begin{aligned}
(\text{I})' &\ge \left[\inf_{\boldsymbol{\theta}\in\mathcal{F}_n}\Pi\left\{\boldsymbol{\alpha}_1 : \mathbb{E}_{\mathbf{x}}|\boldsymbol{\alpha}_1^\top\psi_1(\mathbf{x};\boldsymbol{\theta}^{(1)}) - \boldsymbol{\alpha}_{10}^\top\psi_1(\mathbf{x};\boldsymbol{\theta}^{(1)})| \le \frac{\Delta_n}{6K}\left|\psi_1(\mathbf{x};\boldsymbol{\theta}^{(1)})\right|\right\}\right]^K\\
&\ge \exp\{-K(\mathcal{M}_1' + \mathcal{M}_2')n\epsilon_n^2\};\\
(\text{II})' &\ge \Pi(\boldsymbol{\theta}^{(1)} : \mathcal{S}^{(1)}) \ge \exp(-\mathcal{M}_3' n\epsilon_n^2);\\
(\text{III})' &\approx 1 \text{ for large enough } n,
\end{aligned}
$$

where $\mathcal{M}_1'$, $\mathcal{M}_2'$ and $\mathcal{M}_3'$ are constants, and $\mathcal{S}^{(1)}$ can be similarly defined as in (24). Since the discussion above holds for any $k \in [K]$ and $r \in [R]$, by choosing proper $\mathcal{M}_1$, $\mathcal{M}_2$, $\mathcal{M}_3$, $\mathcal{M}_1'$, $\mathcal{M}_2'$ and $\mathcal{M}_3'$, Condition (c) of Lemma 3 can be verified.

$\square$

### B.2.3 Verification of Condition (a) in Lemma 3

*Proof.* Condition (a) in Lemma 3 can be verified if we can find a parameter set $\Theta_n \subset \Theta$ such that $\log N(\epsilon_n/36, \mathcal{P}_n, d_n) \leq n\epsilon^2$, where $\mathcal{P}_n$ represents the set of all densities that can be represented by model (2) in Section 3.1 with parameters in $\Theta_n$. Consider the following parameter space:

$$\Theta_n = \mathcal{S}_{n1} \otimes \mathcal{S}_{n2} \otimes \mathcal{B}_n \otimes \mathcal{A}_n,$$

where

$$\mathcal{S}_{n1} = \{\boldsymbol{\theta}^{(1)} : |\theta_j^{(1)}| \leq I_n^{(1)}, |\boldsymbol{\gamma}_\theta^{(1)}| = |\{j : |\theta_j^{(1)}| \geq \delta'_{n1}\}| \leq k_{n1}r_{n1}, |\boldsymbol{\gamma}_\theta^{(1)}|_{\text{in}} \leq k'_{n1}s_{n1}\};$$

$$\mathcal{S}_{n2} = \{\boldsymbol{\theta}^{(2)} : |\theta_j^{(2)}| \leq I_n^{(2)}, |\boldsymbol{\gamma}_\theta^{(2)}| = |\{j : |\theta_j^{(2)}| \geq \delta'_{n2}\}| \leq k_{n2}r_{n2}, |\boldsymbol{\gamma}_\theta^{(2)}|_{\text{in}} \leq k'_{n2}s_{n2}\};$$

$$\mathcal{B}_n = \{\mathbf{B}^{(k)} = (\boldsymbol{\beta}_1^{(k)}, ..., \boldsymbol{\beta}_K^{(k)}) \text{ for } k \in [K] : |\beta_{l,j}^{(k)}| \leq b_n, \text{ for } k, l \in [K] \text{ and } j \in [M^{(1)}]\};$$

$$\mathcal{A}_n = \{\mathbf{A}^{(r)} = (\boldsymbol{\alpha}_1^{(r)}, ..., \boldsymbol{\alpha}_K^{(r)}) \text{ for } r \in [R] : |\alpha_{l,j}^{(r)}| \leq a_n, \text{ for } r \in [R], l \in [K], \text{ and } j \in [M^{(2)}]\}.$$

Here, $|\boldsymbol{\gamma}_\theta^{(1)}|_{\text{in}}$ and $|\boldsymbol{\gamma}_\theta^{(2)}|_{\text{in}}$ denote the input dimensions of the sparse network structures $\boldsymbol{\gamma}_\theta^{(1)}$ and $\boldsymbol{\gamma}_\theta^{(1)}$, respectively, and $I_n^{(1)}$, $I_n^{(2)}$, $\delta'_{n1}$, $\delta'_{n2}$, $k_{n1}(\leq n/r_{n1})$, $k_{n2}(\leq n/r_{n2})$, $k'_{n1}(\leq n/s_{n1})$, $k'_{n2}(\leq n/s_{n2})$, $b_n$ and $a_n$ are positive constants whose values will be described later.

The parameter space $\Theta_n$ can be covered by a set of $L_\infty$ balls of the form $(\varrho_j - \rho_n, \varrho_j + \rho_n)_{j=1}^{W_n}$, where for the $j$th coordinate, $\varrho_j$ is the center which lies inside $\Theta_n$, $\rho_n$ is the radius, and $W_n = J_{n1} + J_{n2} + K^2 M^{(1)} + RKM^{(2)}$ is the number of all involved parameters. It can be verified that the number of balls needed to cover the parameter space is upper bounded by

$$\mathcal{K}_n \triangleq \left\{\sum_{j=1}^{k_{n1}r_{n1}} \chi^{(1)}(j)\left(\frac{I_n^{(1)}}{\rho_n} + 1\right)^j\right\} \cdot \left\{\sum_{j=1}^{k_{n2}r_{n2}} \chi^{(2)}(j)\left(\frac{I_n^{(2)}}{\rho_n} + 1\right)^j\right\}$$

$$\cdot \left(\frac{a_n}{\rho_n} + 1\right)^{K^2 M^{(1)}} \cdot \left(\frac{b_n}{\rho_n} + 1\right)^{RKM^{(2)}}$$

$$\triangleq \mathcal{K}_{n1} \cdot \mathcal{K}_{n2} \cdot \mathcal{K}_{n3} \cdot \mathcal{K}_{n4}, \tag{25}$$

where for $l = 1, 2$, $\chi^{(l)}(j)$ denotes the number of all valid networks with exactly $j$ connections and no more than $k'_{nl}s_{nl}$ inputs.

Then we consider the number of $d_n$-balls needed to cover the set of all densities, $\mathcal{P}_n$. Consider two parameters in $\Theta_n$: $\boldsymbol{\theta}_u \triangleq (\boldsymbol{\theta}_u^{(1)\top}, \boldsymbol{\theta}_u^{(2)\top}, \mathbf{B}_u^{(1)\top}, \ldots, \mathbf{B}_u^{(K)\top}, \mathbf{A}_u^{(1)\top}, \ldots, \mathbf{A}_u^{(R)\top})^\top$ and $\boldsymbol{\theta}_v \triangleq (\boldsymbol{\theta}_v^{(1)\top}, \boldsymbol{\theta}_v^{(2)\top}, \mathbf{B}_v^{(1)\top}, \ldots, \mathbf{B}_v^{(K)\top}, \mathbf{A}_v^{(1)\top}, \ldots, \mathbf{A}_v^{(R)\top})^\top$, which satisfy that

(i) there exists a network structure $\boldsymbol{\gamma}^{(1)}$ such that $|\boldsymbol{\gamma}^{(1)}| \leq k_{n1}r_{n1}$, $|\boldsymbol{\gamma}^{(1)}|_{\text{in}} \leq k'_{n1}s_{n1}$, $|\theta_{u,j}^{(1)} - \theta_{v,j}^{(1)}| \leq \rho_n$ for $j \in \boldsymbol{\gamma}^{(1)}$, $\theta_{u,j}^{(1)} \leq \delta'_{n1}$ and $\theta_{v,j}^{(1)} \leq \delta'_{n1}$ for $j \notin \boldsymbol{\gamma}^{(1)}$;

(ii) there exists a network structure $\boldsymbol{\gamma}^{(2)}$ such that $|\boldsymbol{\gamma}^{(2)}| \leq k_{n2}r_{n2}$, $|\boldsymbol{\gamma}^{(2)}|_{\text{in}} \leq k'_{n2}s_{n2}$, $|\theta_{u,j}^{(2)} - \theta_{v,j}^{(2)}| \leq \rho_n$ for $j \in \boldsymbol{\gamma}^{(2)}$, $\theta_{u,j}^{(2)} \leq \delta'_{n2}$ and $\theta_{v,j}^{(2)} \leq \delta'_{n2}$ for $j \notin \boldsymbol{\gamma}^{(2)}$;

(iii) $|\beta_{ul,j}^{(k)} - \beta_{vi,j}^{(k)}| \leq \rho_n$ for $k, l \in [K]$ and $j \in [M^{(1)}]$;

(iv) $|\alpha_{ul,j}^{(r)} - \alpha_{vl,j}^{(r)}| \leq \rho_n$ for $r \in [R], l \in [K]$, and $j \in [M^{(2)}]$,

where $s_{n1}$ and $s_{n2}$ are defined in Condition A.3.1 in Appendix A.2, and $\rho_n$ is a positive constant whose values will be discussed later.

Let $f_{\boldsymbol{\theta}_u}^{(k)}$ and $f_{\boldsymbol{\theta}_v}^{(k)}$ denote the corresponding densities in $\mathcal{P}_n$ for $k \in [K]$. By (21) in Result 1 and the fact that $d(f_1, f_2) \leq \sqrt{K(f_1, f_2)}$ in Appendix B.1.1, we obtain that $d_n^2(\boldsymbol{\theta}_u, \boldsymbol{\theta}_v) \leq \max_{k \in [K], r \in [R]} U_5 \mathbb{E}_\mathbf{x} \|\boldsymbol{\omega}_u^{(k,r)} - \boldsymbol{\omega}_v^{(k,r)}\|_1$ for some positive constant $U_5$, where $\boldsymbol{\omega}_u^{(k,r)} = \mathbf{B}_u^{(k)\top} \psi_2(\mathbf{x}; \boldsymbol{\theta}_u^{(2)}) + \mathbf{A}_u^{(r)\top} \psi_1(\mathbf{x}; \boldsymbol{\theta}_u^{(1)})$ and $\boldsymbol{\omega}_v^{(k,r)} = \mathbf{B}_v^{(k)\top} \psi_2(\mathbf{x}; \boldsymbol{\theta}_v^{(2)}) + \mathbf{A}_v^{(r)\top} \psi_1(\mathbf{x}; \boldsymbol{\theta}_v^{(1)})$

for $k \in [K]$ and $r \in [R]$. For ease of presentation, we omit $k$ and $r$ in $\boldsymbol{B}_u^{(k)}, \boldsymbol{B}_v^{(k)}, \boldsymbol{A}_u^{(r)}, \boldsymbol{A}_v^{(r)}, \boldsymbol{\omega}_u^{(k,r)}$ and $\boldsymbol{\omega}_v^{(k,r)}$, and further obtain that

$$
\begin{aligned}
\|\boldsymbol{\omega}_u - \boldsymbol{\omega}_v\|_1 &\le \|\mathbf{B}_u^\top \psi_2(\mathbf{x};\boldsymbol{\theta}_u^{(2)}) - \mathbf{B}_v^\top \psi_2(\mathbf{x};\boldsymbol{\theta}_v^{(2)})\|_1 + \|\mathbf{A}_u^\top \psi_1(\mathbf{x};\boldsymbol{\theta}_u^{(1)}) - \mathbf{A}_v^\top \psi_1(\mathbf{x};\boldsymbol{\theta}_v^{(1)})\|_1 \\
&\le \|\mathbf{B}_u^\top \psi_2(\mathbf{x};\boldsymbol{\theta}_u^{(2)}) - \mathbf{B}_u^\top \psi_2(\mathbf{x};\boldsymbol{\theta}_v^{(2)})\|_1 + \|\mathbf{B}_u^\top \psi_2(\mathbf{x};\boldsymbol{\theta}_v^{(2)}) - \mathbf{B}_v^\top \psi_2(\mathbf{x};\boldsymbol{\theta}_v^{(2)})\|_1 \\
&\quad + \|\mathbf{A}_u^\top \psi_1(\mathbf{x};\boldsymbol{\theta}_u^{(1)}) - \mathbf{A}_u^\top \psi_1(\mathbf{x};\boldsymbol{\theta}_v^{(1)})\|_1 + \|\mathbf{A}_u^\top \psi_1(\mathbf{x};\boldsymbol{\theta}_v^{(1)}) - \mathbf{A}_v^\top \psi_1(\mathbf{x};\boldsymbol{\theta}_v^{(1)})\|_1 \\
&= \sum_{i=j}^{K} |\boldsymbol{\beta}_{uj}^\top \{\psi_2(\mathbf{x};\boldsymbol{\theta}_u^{(2)}) - \psi_2(\mathbf{x};\boldsymbol{\theta}_v^{(2)})\}| + \sum_{j=1}^{K} |(\boldsymbol{\beta}_{uj} - \boldsymbol{\beta}_{vj})^\top \psi_2(\mathbf{x};\boldsymbol{\theta}_v^{(2)})| \\
&\quad + \sum_{i=1}^{K} |\boldsymbol{\alpha}_{ui}^\top \{\psi_1(\mathbf{x};\boldsymbol{\theta}_u^{(1)}) - \psi_1(\mathbf{x};\boldsymbol{\theta}_v^{(1)})\}| + \sum_{i=1}^{K} |(\boldsymbol{\alpha}_{ui} - \boldsymbol{\alpha}_{vi})^\top \psi_1(\mathbf{x};\boldsymbol{\theta}_v^{(1)})| \\
&\le K b_n \|\psi_2(\mathbf{x};\boldsymbol{\theta}_u^{(2)}) - \psi_2(\mathbf{x};\boldsymbol{\theta}_v^{(2)})\|_1 + K\rho_n \|\psi_2(\mathbf{x};\boldsymbol{\theta}_v^{(2)})\| \\
&\quad + K a_n \|\psi_1(\mathbf{x};\boldsymbol{\theta}_u^{(1)}) - \psi_1(\mathbf{x};\boldsymbol{\theta}_v^{(1)})\|_1 + K\rho_n \|\psi_1(\mathbf{x};\boldsymbol{\theta}_v^{(1)})\|,
\end{aligned}
$$

where the preceding four boundness assumptions for $\boldsymbol{\theta}_u$ and $\boldsymbol{\theta}_v$, together with the definition of $\mathcal{A}_n$ and $\mathcal{B}_n$, are used.

Similar to the proof in Appendix B.2.2, by using Lemmas 1 and 2, we can further obtain that $d_n^2(\boldsymbol{\theta}_u, \boldsymbol{\theta}_v) \le (\epsilon_n/36)^2$ if we choose $\rho_n, \delta'_{n1}$ and $\delta'_{n2}$ in the preceding derivations as follows:

(i) $\delta'_{n1} = U_6 \epsilon_n^2 / b_n J_{n1} (U'_6 I_n^{(1)})^{H_{n1}} (k_{n1} r_{n1}/H_{n1})^{H_{n1}}$ for some constants $U_6 > 0$ and $U'_6 > 1$;

(ii) $\delta'_{n2} = U_7 \epsilon_n^2 / a_n J_{n2} (U'_7 I_n^{(2)})^{H_{n2}} (k_{n2} r_{n2}/H_{n2})^{H_{n2}}$ for some constants $U_7 > 0$ and $U'_7 > 1$;

(iii) $\rho_n = \min \Big\{ U_{8,1} \epsilon_n^2 / b_n H_{n1} (U'_{8,1} I_n^{(1)})^{H_{n1}} (k_{n1} r_{n1}/H_{n1})^{H_{n1}},$

$\qquad\qquad U_{8,2} \epsilon_n^2 / a_n H_{n2} (U'_{8,2} I_n^{(2)})^{H_{n2}} (k_{n2} r_{n2}/H_{n2})^{H_{n2}} \Big\}$

for some constants $U_{8,1}, U_{8,2} > 0$ and $U'_{8,1}, U'_{8,2} > 1$.

Thus, the log covering number of $\mathcal{P}_n$, $\log N(\epsilon_n/36, \mathcal{P}_n, d_n)$, can be upper bounded by $\log \mathcal{K}_n = \log \mathcal{K}_{n1} + \log \mathcal{K}_{n2} + \log \mathcal{K}_{n3} + \log \mathcal{K}_{n4}$, with $\mathcal{K}_{nj}$ given in (25) for $j = 1, 2, 3, 4$.

Considering the fact that $\log \chi^{(1)}(j) \le \log \binom{p_n}{k'_{n1} s_{n1}} \binom{k'_{n1} s_{n1} + H_{n1} \overline{L}_{n1}}{j} \le k'_{n1} s_{n1} \log(p_n) + j \log(k'_{n1} s_{n1} + H_{n1} \overline{L}_{n1}^2) \le k'_{n1} s_{n1} \log(p_n) + k_{n1} r_{n1} \log\{H_{n1}(k'_{n1} s_{n1} + \overline{L}_{n1})^2\}$, and by choosing $\log I_n^{(1)} = O(\log n), \log I_n^{(2)} = O(\log n), a_n \asymp \sqrt{n\epsilon_n^2}$ and $b_n \asymp \sqrt{n\epsilon_n^2}$, we have that

$\log \mathcal{K}_{n1}$

$$
\begin{aligned}
&\le \log(k_{n1} r_{n1}) + \log \chi^{(1)}(k_{n1} r_{n1}) + k_{n1} r_{n1} \log\left(\frac{I_n^{(1)}}{\rho_n} + 1\right) \\
&\le \log(k_{n1} r_{n1}) + k'_{n1} s_{n1} \log(p_n) + k_{n1} r_{n1} \log H_{n1} + 2 k_{n1} r_{n1} \log(k'_{n1} s_{n1} + \overline{L}_{n1}) \\
&\quad + k_{n1} r_{n1} \Big\{ \text{constant} + \log I_n^{(1)} + \log b_n + \log H_{n1} + H_{n1} \log(U'_{7,1} I_n^{(1)}) + H_{n1} \log(k_{n1} r_{n1}/H_{n1}) \\
&\quad + \log a_n + \log H_{n2} + H_{n2} \log(U'_{7,2} I_n^{(2)}) + H_{n2} \log(k_{n2} r_{n2}/H_{n2}) + \log \frac{1}{\epsilon_n^2} \Big\} \\
&= k'_{n1} s_{n1} \log(p_n) + k_{n1} r_{n1} \cdot O\left( H_{n1} \log n + H_{n2} \log n + \log \overline{L}_{n1} + \log \overline{L}_{n2} \right),
\end{aligned}
$$

where the last equality holds since $k_{n1} r_{n1} \le n$, $k_{n2} r_{n2} \le n$, $k'_{n1} s_{n1} \le n$ and $k'_{n1} s_{n1} \le n$. By choosing $k'_{n1}$ and $k_{n1}$ such that $k_{n1} r_{n1} \left( H_{n1} \log n + H_{n2} \log n + \log \overline{L}_{n1} + \log \overline{L}_{n2} \right) \asymp k'_{n1} s_{n1} \log(p_n) \asymp n\epsilon_n^2$, we have $n\epsilon_n^2 = O\{r_{n1} H_{n1} \log n + r_{n1} H_{n2} \log n + r_{n1} \log \overline{L}_{n1} + r_{n1} \log \overline{L}_{n2} + s_{n1} \log(p_n)\}$. By applying similar discussion on $\log \mathcal{K}_{n2}$, $\log \mathcal{K}_{n3}$ and $\log \mathcal{K}_{n4}$, we further obtain that $\log N(\mathcal{P}_n, \epsilon_n) \le \log \mathcal{K}_n \le n\epsilon_n^2$, where the sequence

$\{\epsilon_n\}_{n=1}^{\infty}$ satisfies that $n\epsilon_n^2 = O\{(r_{n1} + r_{n2})(H_{n1} + H_{n2})\log n + (r_{n1} + r_{n2})(\log \overline{L}_{n1} + \log \overline{L}_{n2}) + (s_{n1} + s_{n2})\log p_n\}$.

$\square$

### B.2.4 Verification of Condition (b) in Lemma 3

*Proof.* Since $\Pi(\Theta_n^c) \leq \Pi(\mathcal{S}_{n1}^c) + \Pi(\mathcal{S}_{n2}^c) + \Pi(\mathcal{B}_n^c) + \Pi(\mathcal{A}_n^c)$, we first examine $\Pi(\mathcal{B}_n^c)$ and $\Pi(\mathcal{A}_n^c)$. For any $b_n > 0$, by the prior assumption given in (6), we have that

$$\Pi(\mathcal{B}_n^c) = \Pi\left\{\mathbf{B}^{(k)} = (\boldsymbol{\beta}_1^{(k)}, ..., \boldsymbol{\beta}_K^{(k)}) \text{ for } k \in [K] : |\beta_{l,j}^{(k)}| \leq b_n \text{ for } l, k \in [K], \text{ and } j \in [M^{(2)}]\right\}^c$$

$$= \Pi(\cup_{k,l \in [K], j \in [M^{(2)}]} |\beta_{l,j}^{(k)}| > b_n)$$

$$\leq \sum_{k=1}^{K} \sum_{l=1}^{K} \sum_{j=1}^{M^{(2)}} \pi_B^{(k)}(|\beta_{l,j}^{(k)}| > b_n)$$

$$= K \sum_{k=1}^{K} \sum_{j=1}^{M^{(2)}} \pi_B^{(k)}(|\beta_{1,j}^{(k)}| > b_n),$$

where $\pi_B^{(k)}(\cdot)$ denotes the measure induced by the prior distribution of $\mathbf{B}^{(k)}$ as defined in Section 3.2.

Applying the bounds on the Mills ratio [76] for the standard normal distribution: $\frac{1}{x} - \frac{1}{x^3} < \frac{1-\Phi(x)}{\varphi(x)} < \frac{1}{x} - \frac{1}{x^3} + \frac{3}{x^5}$ for $-\infty < x < \infty$, where $\Phi(\cdot)$ and $\varphi(\cdot)$ denote the CDF and probability density function (PDF) of the standard normal distribution, respectively, we further obtain that, by taking $b_n = \max_{k \in [K]} \sqrt{5\overline{B}_n^{(k)} n\epsilon_n^2}$,

$$\pi_B^{(k)}(|\beta_{1,j}^{(k)}| > b_n) < \frac{2\varphi\left(b_n / \sqrt{(\boldsymbol{\Sigma}_\beta^{(k)})_{j,j}}\right)}{b_n / \sqrt{(\boldsymbol{\Sigma}_\beta^{(k)})_{j,j}}} = \frac{2}{\sqrt{2\pi b_n^2 / (\boldsymbol{\Sigma}_\beta^{(k)})_{j,j}}} \cdot \exp\left\{-\frac{1}{2}(b_n^2 / (\boldsymbol{\Sigma}_\beta^{(k)})_{j,j})\right\}$$

$$\leq \frac{2}{\sqrt{2\pi b_n^2 / \overline{B}_n^{(k)}}} \cdot \exp\left\{-\frac{1}{2}(b_n^2 / \overline{B}_n^{(k)})\right\} \prec \exp\left(-\frac{5}{2}n\epsilon_n^2\right)$$

where $A_{j,j}$ represents the $(j, j)$ element of matrix $A$, the first step is due to the Mills ratio bounds, the second step comes from the definition of the PDF of the standard normal distribution, the third step is due to the fact that the sum of the eigenvalues of a matrix equals the trace of the matrix and the positivity of $\left(\boldsymbol{\Sigma}_\beta^{(k)}\right)_{j,j}$, and the last step comes from the choice of $b_n$. Therefore, it follows that $\Pi(\mathcal{B}_n^c) < K^2 M^{(2)} \exp(-\frac{5}{2}n\epsilon_n^2)$ for large enough $n$.

Similarly, we can obtain that $\Pi(\mathcal{A}_n^c) < K^2 M^{(1)} \exp(-\frac{5}{2}n\epsilon_n^2)$ for large enough $n$ by taking $a_n = \max_{r \in [R]} \sqrt{5\overline{A}_n^{(r)} n\epsilon_n^2}$.

We then consider $\Pi(\mathcal{S}_{n1}^c)$, which can be upper bounded by

$$\Pi(\mathcal{S}_{n1}^c) \leq \Pi(\boldsymbol{\theta}^{(1)} : \cup_{j=1}^{J_{n1}} |\theta_j^{(1)}| > I_n^{(1)}) + \Pi(\boldsymbol{\theta}^{(1)} : |\boldsymbol{\gamma}_\theta^{(1)}| = |\{j : |\theta_j^{(1)}| \geq \delta_{n1}'\}| > k_{n1} r_{n1})$$

$$+ \pi(\boldsymbol{\theta}^{(1)} : |\boldsymbol{\gamma}_\theta^{(1)}|_{\text{in}} > k_{n1}' s_{n1})$$

$$= J_{n1} \pi_\theta^{(1)}(|\theta_1^{(1)}| > I_n^{(1)}|) + \mathbb{P}\{\text{Binomial}(J_{n1}, \nu_n) > k_{n1} r_{n1}\} \tag{26}$$

$$+ \mathbb{P}\{\text{Binomial}(p_n L_1^{(1)}, \nu_n) > k_{n1}' s_{n1}\}$$

where $\nu_n = 1 - \pi_\theta^{(1)}([-\delta_{n1}', \delta_{n1}']) \leq \frac{1}{J_{n1}} \exp\{(H_{n1} + H_{n2})\log n + \log \overline{L}_{n1} + \log \overline{L}_{n2} + \log p_n\}$ according to Condition B.1.2 in Section A.3. For the first term in (26), we have that $J_{n1}\pi_\theta^{(1)}(|\theta_1^{(1)}| > I_n^{(1)}|) \prec \exp\{-(2 + \epsilon_3^{(k)})n\epsilon_n^2\}$ following Condition B.1.3 in Section A.3, where $\epsilon_3^{(k)} > 0$ is a

constant. The second term can be upper bounded by

$$\mathbb{P}\{\text{Binomial}(J_{n1}, \nu_n) > k_{n1}r_{n1}\}$$

$$= \sum_{j=k_{n1}r_{n1}+1}^{J_{n1}} \binom{J_{n1}}{j} \nu_n^j (1-\nu_n)^{J_{n1}-j}$$

$$\leq \sum_{j=k_{n1}r_{n1}+1}^{J_{n1}} J_{n1}^j \nu_n^j (1-\nu_n)^{J_{n1}-j} \leq J_{n1}(J_{n1}\nu_n)^{k_{n1}r_{n1}}$$

$$\leq J_{n1} \exp\left[-S_0^{(1)} k_{n1} r_{n1} \{(H_{n1}+H_{n2})\log n + \log \overline{L}_{n1} + \log \overline{L}_{n2} + \log p_n\}\right]$$

$$\prec \exp\left\{-S_0^{(1)} n\epsilon_n^2\right\},$$

where the last inequality is due to the choice of $k_{n1}$ in Appendix B.2.3, and $S_0^{(1)} > 2$ is a constant given in Condition B.1.2 in Section A.3.

Similarly, for the third term in (26) we have that

$$\mathbb{P}\{\text{Binomial}(p_n L_1^{(1)}, \nu_n) > k_{n1}' s_{n1}\}$$

$$= \sum_{j=k_{n1}'s_{n1}+1}^{p_n L_1^{(1)}} j\binom{p_n L_1^{(1)}}{j} \nu_n^j (1-\nu_n)^{p_n L_1^{(1)}-j}$$

$$\leq \sum_{j=k_{n1}'s_{n1}+1}^{p_n L_1^{(1)}} (p_n L_1^{(1)})^j \nu_n^j (1-\nu_n)^{p_n L_1^{(1)}-j} = p_n L_1^{(1)} (J_{n1}\nu_n)^{k_{n1}'s_{n1}} \left(\frac{p_n L_1^{(1)}}{J_{n1}}\right)^{k_{n1}'s_{n1}}$$

$$\leq p_n L_1^{(1)} \exp\left[-S_0^{(1)} k_{n1}' s_{n1} \{(H_{n1}+H_{n2})\log n + \log \overline{L}_{n1} + \log \overline{L}_{n2} + \log p_n\}\right.$$

$$\left. - k_{n1}' s_{n1} \log\left(\frac{J_{n1}}{p_n L_1^{(1)}}\right)\right]$$

$$\prec \exp\{-S_0^{(1)} n\epsilon_n^2\}.$$

Thus, Condition (b) in Lemma 3 is verified.

$$\square$$

## B.3   Proof of Theorem 2

We now prove Theorem 2. The proof of the upper bound adapts the techniques in [46], and the proof of the lower bound employs a general mechanism for lower bounding the Bayes risk [40], which is tighter than the Fano's inequality [44].

*Proof.* **Upper Bound.** The Bayes risk can be written as

$$\Re_{\text{Bayes}} = \inf_{\overline{\mathbf{y}}} \left\{ \sum_{\mathbf{y} \in [K]^{\bar{n}}} \hbar(\mathbf{y}) \mathbb{E}(\mathcal{L}(\overline{\mathbf{y}}, \mathbf{y})|\mathbf{y}; \boldsymbol{\tau}) \right\}$$

$$= \inf_{\overline{\mathbf{y}}} \left\{ \sum_{\mathbf{y} \in [K]^{\bar{n}}} \hbar(\mathbf{y}) \frac{1}{\bar{n}} \sum_{i=1}^{\bar{n}} \mathbb{P}(\overline{y}_i \neq y_i | y_i; \boldsymbol{\tau}) \right\}$$

$$= \inf_{\overline{\mathbf{y}}} \left\{ \frac{1}{\bar{n}} \sum_{i=1}^{\bar{n}} \sum_{\mathbf{y} \in [K]^{\bar{n}}} \hbar(\mathbf{y}) \mathbb{P}(\overline{y}_i \neq y_i | y_i; \boldsymbol{\tau}) \right\}$$

$$= \inf_{\overline{\mathbf{y}}} \left\{ \frac{1}{\bar{n}} \sum_{i=1}^{\bar{n}} \sum_{g=1}^{K} \hbar_{i,g} \mathbb{P}(\overline{y}_i \neq g | y_i = g; \boldsymbol{\tau}) \right\}.$$

We first consider the situation $g = 1$:

$$\mathbb{P}(\overline{y}_i \neq 1 | y_i = 1; \boldsymbol{\tau}) \leq \sum_{g'=2}^{K} \mathbb{P}(\overline{y}_i = g' | y_i = 1; \boldsymbol{\tau}).$$

For each $g' \geq 2$, considering the inference procedure of $\overline{y}_i$, we have that

$$\mathbb{P}(\overline{y}_i = g' | y_i = 1; \boldsymbol{\tau})$$

$$\leq \mathbb{P}\left( \frac{\hbar_{i,g'}}{\hbar_{i,1}} \prod_{r=1}^{R} \prod_{l=1}^{K} \left( \frac{\tau_{i,g'l}^{(r)}}{\tau_{i,1l}^{(r)}} \right)^{\mathbb{I}(y_i^{(r)}=l)} > \Omega \,\middle|\, y_i = 1; \boldsymbol{\tau} \right)$$

$$= \mathbb{P}\left( \left(\frac{\hbar_{i,g'}}{\Omega \hbar_{i,1}}\right)^{\lambda} \prod_{r=1}^{R} \prod_{l=1}^{K} \left( \frac{\tau_{i,g'l}^{(r)}}{\tau_{i,1l}^{(r)}} \right)^{\lambda \mathbb{I}(y_i^{(r)}=l)} > 1 \,\middle|\, y_i = 1; \boldsymbol{\tau} \right)$$

$$\leq \min_{\lambda \geq 0} \left(\frac{\hbar_{i,g'}}{\Omega \hbar_{i,1}}\right)^{\lambda} \prod_{r=1}^{R} \mathbb{E}\left\{ \prod_{l=1}^{K} \left( \frac{\tau_{i,g'l}^{(r)}}{\tau_{i,1l}^{(r)}} \right)^{\lambda \mathbb{I}(y_i^{(r)}=l)} \,\middle|\, y_i = 1; \boldsymbol{\tau} \right\}$$

$$= \min_{\lambda \geq 0} \left(\frac{\hbar_{i,g'}}{\Omega \hbar_{i,1}}\right)^{\lambda} \prod_{r=1}^{R} \sum_{l=1}^{K} \left( \tau_{i,g'l}^{(r)} \right)^{\lambda} \left( \tau_{i,1l}^{(r)} \right)^{1-\lambda}$$

$$\leq \exp\left\{ -R \left\{ -\min_{0\leq\lambda\leq 1} \frac{1}{R} \left\{ -\lambda \log \frac{\Omega \hbar_{i,1}}{\hbar_{i,g'}} + \sum_{r=1}^{R} \log \left\{ \sum_{l=1}^{K} \left( \tau_{i,g'l}^{(r)} \right)^{\lambda} \left( \tau_{i,1l}^{(r)} \right)^{1-\lambda} \right\} \right\} \right\} \right\},$$

where the inequality in the third line follows from Markov's inequality. Denote $C_{gg'}^{(i)} = -\min_{0\leq\lambda\leq 1} \frac{1}{R} \left\{ -\lambda \log \frac{\Omega \hbar_{i,g}}{\hbar_{i,g'}} + \sum_{r=1}^{R} \log \left\{ \sum_{l=1}^{K} \left( \tau_{i,gl}^{(r)} \right)^{1-\lambda} \left( \tau_{i,g'l}^{(r)} \right)^{\lambda} \right\} \right\}$ and $I_{\Omega}^{(g)}(\hbar_i, \boldsymbol{\tau}_i) = \min_{g'\neq g} C_{gg'}^{(i)}$, we further obtain that

$$\mathbb{P}(\overline{y}_i = l | y_i = 1; \boldsymbol{\tau}) \leq \exp\left\{ -R C_{1g'}^{(i)} \right\}$$

$$\leq \exp\left\{ -R \min_{g'\neq 1} C_{gg'}^{(i)} \right\} = \exp\left\{ -R I_{\Omega}^{(1)}(\hbar_i, \boldsymbol{\tau}_i) \right\}.$$

Thus, we can obtain the upper bound of the Bayes risk:

$$\Re_{\text{Bayes}} \leq \frac{K-1}{\bar{n}} \sum_{i=1}^{\bar{n}} \sum_{g=1}^{K} \hbar_{i,g} e^{-R I_{\Omega}^{(g)}(\hbar_i, \boldsymbol{\tau}_i)}.$$

**Lower Bound.** Using Markov's inequality, we obtain that

$$\Re_{\text{Bayes}} = \inf_{\overline{y}} \left\{ \sum_{\mathbf{y}\in[K]^{\bar{n}}} \hbar(\mathbf{y}) \mathbb{E}(\mathcal{L}(\overline{\mathbf{y}}, \mathbf{y}) | \mathbf{y}; \boldsymbol{\tau}) \right\}$$

$$\geq \inf_{\overline{y}} \left\{ \sum_{\mathbf{y}\in[K]^{\bar{n}}} \hbar(\mathbf{y}) \frac{1}{\bar{n}} \mathbb{P}\left( \mathcal{L}(\overline{\mathbf{y}}, \mathbf{y}) > \frac{1}{\bar{n}} | \mathbf{y}; \boldsymbol{\tau} \right) \right\}$$

$$= \frac{1}{\bar{n}} \inf_{\overline{y}} \left\{ \sum_{\mathbf{y}\in[K]^{\bar{n}}} \hbar(\mathbf{y}) \mathbb{P}(\overline{\mathbf{y}} \neq \mathbf{y} | \mathbf{y}; \boldsymbol{\tau}) \right\}$$

$$= \frac{1}{\bar{n}} \inf_{\overline{y}} \left[ \sum_{\mathbf{y}\in[K]^{\bar{n}}} \hbar(\mathbf{y}) \mathbb{E}\left\{ \mathbf{1}(\overline{\mathbf{y}} \neq \mathbf{y}) | \mathbf{y}; \boldsymbol{\tau} \right\} \right]$$

$$\triangleq \frac{1}{\bar{n}} \overline{\Re},$$

where the second step is due to Markov's inequality. Using Lemma 7, we can obtain that

$$I_f(\hbar, \mathcal{P}) \geq \phi_f(\overline{\mathfrak{R}}, \Re_0),$$

where $\mathcal{P}$ is the set of distributions of $\{\tilde{\mathbf{y}}_i\}_{i=1}^{\bar{n}}$ induced by $\mathbf{y}$, and for 0-1 loss, $\Re_0$ has the expression in [40]: $\Re_0 = 1 - \sup_{\mathbf{a} \in [K]^{\bar{n}}} \hbar(B(\mathbf{a}))$ with $B(\mathbf{a}) = \{\mathbf{y} \in [K]^{\bar{n}} : \mathbf{1}(\mathbf{y} \neq \mathbf{a}) = 0\}$. Then, it can be easily verified that $\Re_0 = 1 - \prod_{i=1}^{\bar{n}} \max_{k \in [K]} \hbar_{i,k}$.

Let $g(t) = \phi_f(t, \Re_0)$, which is, by Lemma 6, non-increasing for $t \in [0, \Re_0]$, convex, and continuous in $t$. Using the convexity of $g(t)$, we obtain that for every $t \in (0, \Re_0]$

$$\phi_f(\overline{\mathfrak{R}}, \Re_0) \geq \phi_f(t, \Re_0) + \phi'_f(t-, \Re_0)(\overline{\mathfrak{R}} - t),$$

where $\phi'_f(t-, \Re_0)$ denotes the left derivative of $x \mapsto \phi_f(x, \Re_0)$ at $x = t$. Then we can obtain that

$$\overline{\mathfrak{R}} \geq t + \frac{\phi_f(\overline{\mathfrak{R}}, \Re_0) - \phi_f(t, \Re_0)}{\phi'_f(t-, \Re_0)} \geq t + \frac{I_f(\hbar, \mathcal{P}) - \phi_f(t, \Re_0)}{\phi'_f(t-, \Re_0)},$$

where the inequalities come from the fact that $I(\hbar, \mathcal{P}) \geq \phi_f(\overline{\mathfrak{R}}, \Re_0)$ by Lemma 7 and that $\phi'_f(t-, \Re_0) \leq 0$ due to the non-increasing function $g(t)$ over $t \in [0, \Re_0]$. By taking $f(t) = t \log t$ and $t = \frac{\Re_0}{1+\Re_0}$, we obtain that

$$\overline{\mathfrak{R}} \geq 1 + \frac{I(\hbar, \mathcal{P}) + \log(1 + \Re_0)}{\log(1 - \Re_0)},$$

where $I(\hbar, \mathcal{P})$ is the mutual information of $\hbar$ and $\mathcal{P}$.

Let $Y$ and $Z$ denote two independent random variables such that $Y \sim \hbar$ and $Z \sim \mathcal{P}$, and let $D_{KL}(\cdot \| \cdot)$ denote the KL divergence of the associated distributions. For ease of notation, we use $P(Y)$, $P(Z)$, $P(Y, Z)$ and $P(Z|Y)$ to denote the $\hbar$, $\mathcal{P}$, the joint distribution of $Y$ and $Z$, and the conditional distribution of $Z$, given $Y$. Then $I(\hbar, \mathcal{P})$ can be evaluated as follows [43]:

$$
\begin{aligned}
I(\hbar, \mathcal{P}) &= I(Y; Z) \\
&= D_{KL}(P(Y, Z) \| P(Y)P(Z)) \\
&= \sum_{Y \in [K]^{\bar{n}}, Z \in [K]^{R\bar{n}}} P(Y)P(Z|Y) \log \left\{ \frac{P(Y)P(Z|Y)}{P(Y)P(Z)} \right\} \\
&= \sum_{Y \in [K]^{\bar{n}}} P(Y) D_{KL}(P(Z|Y) \| P(Z)) \\
&= \sum_{Y \in [K]^{\bar{n}}} P(Y) D_{KL}(P(Z|Y) \| \sum_{Y' \in [K]^{\bar{n}}} P(Z|Y')P(Y')).
\end{aligned}
$$

Using the log sum inequality in Lemma 4, we further obtain that

$$
\begin{aligned}
I(\hbar, \mathcal{P}) &= \sum_{Y \in [K]^{\bar{n}}} P(Y) \sum_{Y' \in [K]^{\bar{n}}} P(Y') D_{KL}(P(Z|Y) \| P(Z|Y')) \\
&= \sum_{Y \in [K]^{\bar{n}}} \sum_{Y' \in [K]^{\bar{n}}} P(Y)P(Y') \left\{ \sum_{i=1}^{\bar{n}} \sum_{r=1}^{R} D_{KL}(P(Z_i^{(r)}|Y) \| P(Z_i^{(r)}|Y')) \right\} \\
&= \sum_{Y \in [K]^{\bar{n}}} \sum_{Y' \in [K]^{\bar{n}}} \sum_{i=1}^{\bar{n}} \sum_{r=1}^{R} P(Y)P(Y') D_{KL} \left( \tau_{i,Y_i*}^{(r)} \| \tau_{i,Y'_i*}^{(r)} \right) \\
&= \sum_{i=1}^{\bar{n}} \sum_{r=1}^{R} \sum_{g=1}^{K} \sum_{g'=1}^{K} P(Y_i = g)P(Y'_i = g') D_{KL} \left( \tau_{i,g*}^{(r)} \| \tau_{i,g'*}^{(r)} \right) \\
&= \sum_{i=1}^{\bar{n}} \sum_{r=1}^{R} \sum_{g=1}^{K} \sum_{g'=1}^{K} \hbar_{i,g} \hbar_{i,g'} D_{KL} \left( \tau_{i,g*}^{(r)} \| \tau_{i,g'*}^{(r)} \right) \\
&\triangleq \bar{n} \overline{D}_{KL}(\hbar, \tau).
\end{aligned}
$$

Combining the discussion above, we obtain that

$$\Re_{\text{Bayes}} \geq \frac{1}{\bar{n}} \left[ 1 - \frac{\overline{D}_{KL}(\hbar, \boldsymbol{\tau}) + \frac{1}{\bar{n}} \log(2 - \prod_{i=1}^{\bar{n}} \max_{k \in [K]} \hbar_{i,k})}{\left\{ \sum_{i=1}^{\bar{n}} \log(\max_{k \in [K]} \hbar_{i,k}) \right\} / \bar{n}} \right].$$

$\square$

### B.4 Proof of Corollary 3

*Proof.* For $k \in [K]$, $r \in [R]$, and consider the following set:

$$\mathcal{S}^c = \left\{ \boldsymbol{\theta} : \cup_{k \in [K], r \in [R]} \left\{ \max_{\tilde{y} \in [K], \mathbf{x} \in \overline{\mathcal{D}}_{\mathbf{x}}} \left| f^{(k,r)}(\tilde{y} | \boldsymbol{\omega}^{(k,r)}) - f_0^{(k,r)}(\tilde{y} | \boldsymbol{\omega}_0^{(k,r)}) \right| > M_n \epsilon_n \right\} \right\},$$

where $\boldsymbol{\theta}$, $\epsilon_n$ and $M_n$ satisfy the conditions in Theorem 1, $\boldsymbol{\omega}^{(k,r)}$ and $\boldsymbol{\omega}_0^{(k,r)}$ depend on $\mathbf{x}$, and $\overline{\mathcal{D}}_{\mathbf{x}}$ represents the set of observations of the instances in $\overline{\mathcal{D}}$. Then, according to Theorem 1 and using the union bound, we have that

$$\Pi(\mathcal{S}^c | \overline{\mathcal{D}}_0) \leq \sum_{k \in [K], r \in [R]} \Pi \left\{ \boldsymbol{\theta} : \max_{\tilde{y} \in [K], \mathbf{x} \in \overline{\mathcal{D}}_{\mathbf{x}}} \left| f^{(k,r)}(\tilde{y} | \boldsymbol{\omega}^{(k,r)}) - f_0^{(k,r)}(\tilde{y} | \boldsymbol{\omega}_0^{(k,r)}) \right| > M_n \epsilon_n \Big| \overline{\mathcal{D}}_0 \right\}$$

$$\leq \sum_{k \in [K], r \in [R]} \Pi \left( \boldsymbol{\theta} : \| f^{(k,r)} - f_0^{(k,r)} \|_{\text{TV}} > U_9 M_n \epsilon_n \Big| \overline{\mathcal{D}}_0 \right)$$

$$\leq \sum_{k \in [K], r \in [R]} \Pi \left( \boldsymbol{\theta} : d(f^{(k,r)}, f_0^{(k,r)}) > \sqrt{2} U_9 M_n \epsilon_n \Big| \overline{\mathcal{D}}_0 \right) \longrightarrow 0$$

in $\mathbb{P}_0^n$ probability as $n \to \infty$, where $\| \cdot \|_{\text{TV}}$ denotes the total variation distance of the associated distributions as defined in Example 1, and $U_9$ is a positive constant. Here, the second inequality is due to the definition of the total variation distance of discrete distributions and the fact that $f(\mathbf{x}) > \varsigma_2$ for $\mathbf{x} \in \overline{\mathcal{D}}_{\mathbf{x}}$, and the third inequality follows the (i) in Lemma 5. Thus, $\Pi(\mathcal{S} | \overline{\mathcal{D}}_0) \longrightarrow 1$ in $\mathbb{P}_0^n$ probability as $n \to \infty$, where $\mathcal{S}$ can be written as

$$\mathcal{S} = \left\{ \boldsymbol{\theta} : \max_{l \in [K], \mathbf{x} \in \overline{\mathcal{D}}_{\mathbf{x}}} \left| \overline{\tau}_{kl}^{(r)}(\mathbf{x}) - \tau_{kl}^{(r)}(\mathbf{x}) \right| \leq M_n \epsilon_n, k \in [K], r \in [R] \right\}.$$

We now consider the Bayes risk $\Re_{\text{Bayes}}$, which, according to the proof of Theorem 2, can be written as $\Re_{\text{Bayes}} = \inf_{\overline{\mathbf{y}}} \left\{ \frac{1}{\bar{n}} \sum_{i=1}^{\bar{n}} \sum_{g=1}^{K} \hbar_{i,g} \mathbb{P}(\overline{y}_i \neq g | y_i = g; \boldsymbol{\tau}) \right\}$. We first consider $g = 1$:

$$\mathbb{P}(\overline{y}_i \neq 1 | y_i = 1; \boldsymbol{\tau})$$

$$\leq \sum_{g'=2}^{K} \mathbb{P}(\overline{y}_i = g' | y_i = 1; \boldsymbol{\tau})$$

$$\leq \sum_{g'=2}^{K} \mathbb{P} \left\{ \left( \frac{\hbar_{i,g'}}{\Omega \hbar_{i,1}} \right) \prod_{r=1}^{R} \prod_{l=1}^{K} \left( \frac{\overline{\tau}_{i,g'l}^{(r)}}{\overline{\tau}_{i,1l}^{(r)}} \right)^{\mathbf{1}(y_i^{(r)}=l)} > 1 \Big| y_i = 1; \boldsymbol{\tau} \right\}$$

$$= \sum_{g'=2}^{K} \mathbb{P} \left\{ \left( \frac{\hbar_{i,g'}}{\Omega \hbar_{i,1}} \right) \prod_{r=1}^{R} \prod_{l=1}^{K} \left( \frac{\tau_{i,g'l}^{(r)}}{\tau_{i,1l}^{(r)}} \right)^{\mathbf{1}(y_i^{(r)}=l)} \cdot \prod_{r=1}^{R} \prod_{l=1}^{K} \left( \frac{\tau_{i,1l}^{(r)}}{\overline{\tau}_{i,1l}^{(r)}} \cdot \frac{\overline{\tau}_{i,g'l}^{(r)}}{\tau_{i,g'l}^{(r)}} \right)^{\mathbf{1}(y_i^{(r)}=l)} > 1 \Big| y_i = 1; \boldsymbol{\tau} \right\}.$$

If $\{\overline{\tau}_{k*}^{(r)}(\cdot)\}_{k \in [K], r \in [R]} \in \mathcal{S}$, using Taylor's expansion, we have that

$$\log \left\{ \prod_{r=1}^{R} \prod_{l=1}^{K} \left( \frac{\tau_{i,1l}^{(r)}}{\overline{\tau}_{i,1l}^{(r)}} \cdot \frac{\overline{\tau}_{i,g'l}^{(r)}}{\tau_{i,g'l}^{(r)}} \right)^{\mathbf{1}(y_i^{(r)}=l)} \right\}$$

$$= \sum_{r=1}^{R} \sum_{k=1}^{K} \left( \log \frac{\overline{\tau}_{i,g'l}^{(r)}}{\tau_{i,g'l}^{(r)}} - \log \frac{\overline{\tau}_{i,1l}^{(r)}}{\tau_{i,1l}^{(r)}} \right) \mathbf{1}(y_i^{(r)} = l) \leq \kappa M_n \epsilon_n,$$

for some constant $\kappa > 0$ and large enough $n$. For any $\epsilon > 0$, we have that $\kappa M_n \epsilon_n < \epsilon$ by taking $M_n = \epsilon/(2\kappa M_n \epsilon_n)$. Thus, we further obtain that

$$\mathbb{P}(\overline{y}_i \neq 1 | y_i = 1; \boldsymbol{\tau}) \leq \sum_{g'=2}^{K} \mathbb{P}\left( \exp(\epsilon) \left( \frac{\hbar_{i,g'}}{\Omega \hbar_{i,1}} \right) \prod_{r=1}^{R} \prod_{l=1}^{K} \left( \frac{\tau_{i,g'l}^{(r)}}{\tau_{i,1l}^{(r)}} \right)^{\mathbf{1}(y_i^{(r)}=l)} > 1 \Big| y_i = 1; \boldsymbol{\tau} \right)$$

$$\leq (K-1)\exp\left\{ -R I_\Omega^{(1)}(\hbar_i, \boldsymbol{\tau}_i) + \epsilon \right\}$$

for $\{\overline{\tau}_{k*}^{(r)}(\cdot)\}_{k\in[K],r\in[R]} \in \mathcal{S}$, where in the second step, we use Markov's inequality and the definition of $I_\Omega^{(1)}(\hbar_i, \boldsymbol{\tau}_i)$. Similar results also hold for $g = 2, ..., K$. Hence, according to Theorem 1, we have that

$$\Pi\left\{ \overline{\boldsymbol{\tau}} : \Re_{\text{Bayes}} \leq \frac{K-1}{\overline{n}} \sum_{i=1}^{\overline{n}} \sum_{g=1}^{K} \hbar_{i,g} e^{-R I_\Omega^{(g)}(\hbar_i, \boldsymbol{\tau}_i)+\epsilon} \Big| \overline{\mathcal{D}}_0 \right\} \geq \Pi(\mathcal{S}|\overline{\mathcal{D}}_0),$$

where $\Pi(\mathcal{S}|\overline{\mathcal{D}}_0) \longrightarrow 1$ in $\mathbb{P}_0^n$ probability as $n \to \infty$. Thus, the proof is completed.

$\square$

## C  Implementation details and additional experimental results

### C.1  Implementation details

**Dataset description.**  We assess the effectiveness of our method on three image datasets with synthetic annotations, MNIST [47], CIFAR-10 [48], and CIFAR-100 [48], and two datasets with human annotations, CIFAR-10N [49] and LabelMe [50, 51]. MNIST consists of $28 \times 28$ grayscale images with 10 classes, containing 60,000 training images and 10,000 test images. CIFAR10 has 10 classes of $32 \times 32 \times 3$ color images, with 50,000 images used for training and 10,000 for testing. CIFAR100 also consists of 50,000 training images and 10,000 test images, whose size is $32 \times 32 \times 3$, but with 100 fine-grained classes. We further consider two additional datasets with human annotations, CIFAR-10N [49] and LabelMe [50, 51]. For each instance in CIFAR10, CIFAR-10N provides three independent human annotated noisy labels, with the aggregation of three noisy labels by majority voting being $9.03\%$. LabelMe is an image classification dataset consists of 10,000 training images, 500 validation images, and 1,188 test images. For images in the training set, LabelMe has noisy and incomplete labels provided by a total of $R = 59$ annotators, with each image being labeled by an average of 2.547 annotations. For all the datasets except LabelMe, we leave out 10% of the training data as a noisy validation set.

**Experiment setup.**  The network structure for the MNIST dataset is chosen to be Lenet-5 [52]. We choose ResNet-18 [2] for CIFAR-10 and CIFAR-10N, and ResNet-34 architecture [2] for CIFAR-100. As in [53], we employ the pretrained VGG-16 network, followed by a fully connected layer and a softmax output layer for the LabelMe dataset, using 50% dropout. We take the batch size to be 128 for all the datasets. For MNIST, we use the SGD optimizer with momentum 0.9, weight decay $5 \times 10^{-4}$, and an initial learning rate of $10^{-2}$. The learning rate is divided by 10 at the 40th epoch, and we set 80 epochs in total, in which the first 20 epochs are used to warm up the model on the noisy dataset and to determine anchor points. For CIFAR10, CIFAR100, CIFAR10N and LabelMe, the Adam optimizer [77] is utilized with the weight decay $5 \times 10^{-4}$. For CIFAR10, CIFAR100 and CIFAR10N, the initial learning rate is set to be $10^{-3}$, and the network is trained for 120, 150, and 120 epochs for CIFAR10, CIFAR100 and CIFAR10N, repectively, with the first 30 epochs used as the warm-up stage. The model is trained on LabelMe for 100 epochs, with an initial learning rate $5 \times 10^{-4}$ and first 20 epochs as the warm-up stage.

**Baselines.**  We compare the proposed method with the following state-of-art methods: (1) CE (Clean), which trains the network with the standard cross entropy loss on the clean datasets; (2) CE (MV), which trains the network using the labels from majority voting; (3) CE (EM) [9], which obtains the aggregated labels utilizing the EM algorithm; (4) DoctorNet [54], which models the annotators individually and then learns averaging weights for combining them; (5) GCE [55], which generalizes

the mean absolute error and the cross entropy loss to combat errors in training labels; (6) Co-teaching [56], which trains two networks and cross-trains on instances with small loss values; (6) Co-teaching+ [57], which bridges the "Update by Disagreement" strategy with the Co-teaching method; (7) BLTM [17], which directly models the transition matrix from Bayes optimal labels to noisy labels and learns a classifier to predict Bayes optimal labels; (8) MBEM [11], which alternates in rounds between estimating annotator quality from disagreement with the current model and updating the model by optimizing the a loss function that accounts for the current estimate of worker quality; (9) CrowdLayer [53], which concatenates the classifier with multiple annotator-specific layers and simultaneously learns the parameters; (10) TraceReg [8], which uses a loss function similar to CrowdLayer [53], but adds a regularization to establish identifiability of the confusion matrices and the classifier; (11) Max-MIG [7], which jointly aggregates the noisy crowdsourced labels and trains the classifier; (12) CoNAL [58], which decomposes the annotation noise into common and individual confusions; (13) GeoCrowdNet (F) [13]; and (14) GeoCrowdNet (W) [13], which are two regularized variants of the coupled cross-entropy minimization to enhance the identifiability of the confusion matrices. Among these methods, GCE and Co-teaching are strong baselines dealing with single noisy label issue, and we adapt them to the multiple annotations setting by utilizing the majority vote labels for loss computation.

**Implementation details.** We first warm up the base models on the noisy dataset with majority vote labels, obtain the set of anchor points $\overline{\mathcal{D}}_0$, and train the sparse Bayesian model on $\overline{\mathcal{D}}_0$ by maximizing the log posterior distribution of network parameters and excluding non-informative parameters with low posterior inclusion probability. With the noise transition model trained, we then iteratively implement the label correction algorithm (10) and update the base models, where linearly increase the threshold $\Omega_t$ in the training process. Specifically, let $ER$ denoted the estimated average noise rate, and let $r_0$ and $r_1$ represent two prespecified constants with $r_1 \geq r_0 > 0$, which determine the magnitude of $\Omega_t$ at the beginning and the end of the training process. In the $t$th epoch, we set the threshold $\Omega_t$ to be $\Omega_t = (1 - ER) \cdot (r_0 + t \cdot \frac{r_1 - r_0}{T})$ in the experiment. The implementation procedure of the proposed method is summarized in Algorithm 1.

---

**Algorithm 1:** Annotator-Specific Instance-Dependent Label Noise Learning via Sparse Bayesian Network and Pairwise LRT

---

**Input** : Noisy training data $\mathcal{D} = \{\mathbf{x}_i, \tilde{\mathbf{y}}_i\}_{i=1}^N$;

1 // Warm Up the Base Models $h_1$ and $h_2$; Collect Anchor Points $\overline{\mathcal{D}}_0$ [26]
2 // Sections 3.1 and 3.2:  Train Sparse Transition Matrices
3 Maximize the log posterior distribution (12) of network parameters and obtain the MAP $\widehat{\boldsymbol{\theta}}$;
4 Calculate the posterior inclusion probability $\mathbb{P}(\gamma_k^{(j)} = 1|\widehat{\boldsymbol{\theta}})$ using (13); if $\mathbb{P}(\gamma_k^{(j)} = 1|\widehat{\boldsymbol{\theta}}) < 0.5$, zero out the corresponding parameter;
5 Fine tune the sparse network and obtain the noise transition model $f$;
6 // Section 3.3:  Pairwise Likelihood Ratio Test and Update the Base Models
7 **for** $epoch\ t = 1, ..., T$ **do**
8      Update the linearly increasing threshold $\Omega_t$ for pairwise LRT;
9      **for** *each instance* $\mathbf{x}_i$ **do**
10          Use $f_1(\mathbf{x}_i)$ as the prior of $\mathbf{x}_i$; if $\overline{y}_i$ satisfies (10) with threshold $\Omega_t$, put $\{\mathbf{x}_i, \overline{y}_i\}$ in $\overline{D}_{t,1}$;
11          Use $f_2(\mathbf{x}_i)$ as the prior of $\mathbf{x}_i$; if $\overline{y}_i$ satisfies (10) with threshold $\Omega_t$, put $\{\mathbf{x}_i, \overline{y}_i\}$ in $\overline{D}_{t,2}$;
12      **end for**
13      Update $f_1$ using $\overline{D}_{t,2}$;
14      Update $f_2$ using $\overline{D}_{t,1}$;
15 **end for**

**Output** : $h_1$, $h_2$ and $f$.

---

## C.2 Additional experimental results.

**Accuracy and number of selected labels.** To validate the effectiveness of the proposed label correction method, we plot the accuracy and the number of corrected labels using criterion (10) in Figures 3 and 4 for all the considered datasets, where the solid line is the accuracy (left y-axis) and the dashed line is the number (right y-axis) of corrected labels.

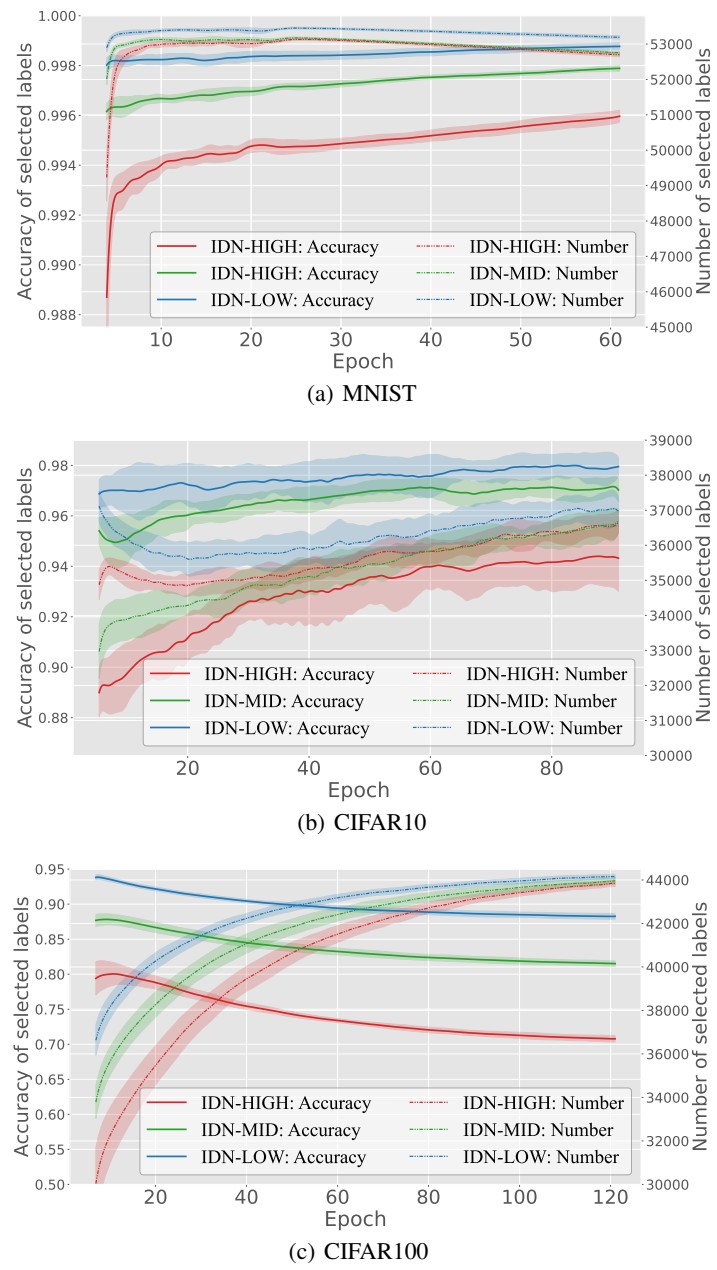

(a) MNIST

(b) CIFAR10

(c) CIFAR100

Figure 3: The accuracy and the number of corrected labels using the label correction algorithm (10) on synthetic noisy datasets. The error bar for standard deviation has been shaded.

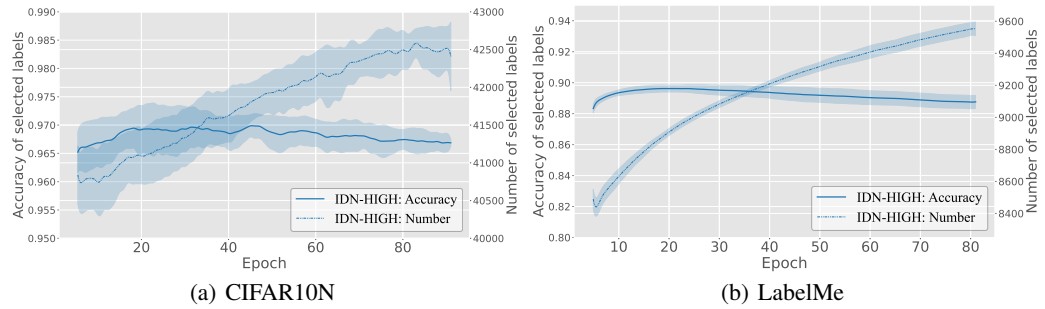

|              (a) CIFAR10N              |              (b) LabelMe              |

Figure 4: The accuracy and the number of corrected labels using the label correction algorithm (10) on real-world noisy datasets. The error bar for standard deviation has been shaded.

**Classification accuracy on MNIST.** The test accuracy on CIFAR10, CIFAR100, CIFAR10, and LabelMe are provided in Section 5. Here we present the average test accuracy on the MNIST dataset in Table 2, where the two highest accuraries are bold faced. Clearly, that the proposed method achieves the best performance.

Table 2: Average accuracy of learning MNIST dataset

|                      | IDN-LOW | IDN-MID | IDN-HIGH |
|----------------------|---------|---------|----------|
| CE (Clean)           |         | $99.14_{\pm0.10}$ |          |
| CE (MV)              | $98.59_{\pm0.13}$ | $97.97_{\pm0.13}$ | $96.60_{\pm0.52}$ |
| CE (EM) [9]          | $98.49_{\pm0.11}$ | $75.84_{\pm0.97}$ | $96.78_{\pm0.52}$ |
| DoctorNet [54]       | $98.17_{\pm0.12}$ | $97.36_{\pm0.23}$ | $95.32_{\pm0.51}$ |
| GCE [55]             | $99.02_{\pm0.15}$ | $98.51_{\pm0.24}$ | $98.05_{\pm0.42}$ |
| Co-teaching [56]     | $98.85_{\pm0.11}$ | $\mathbf{98.61_{\pm0.18}}$ | $\mathbf{98.23_{\pm0.21}}$ |
| Co-teaching+ [57]    | $98.64_{\pm0.10}$ | $98.33_{\pm0.10}$ | $\mathbf{97.67_{\pm0.44}}$ |
| BLTM [17]            | $98.69_{\pm0.06}$ | $98.11_{\pm0.09}$ | $96.40_{\pm0.81}$ |
| MBEM [11]            | $98.66_{\pm0.07}$ | $98.24_{\pm0.05}$ | $97.46_{\pm0.21}$ |
| CrowdLayer [53]      | $97.29_{\pm0.41}$ | $94.88_{\pm0.92}$ | $90.51_{\pm2.47}$ |
| TraceReg [8]         | $98.68_{\pm0.05}$ | $97.96_{\pm0.18}$ | $96.70_{\pm0.57}$ |
| Max-MIG [7]          | $98.62_{\pm0.06}$ | $97.97_{\pm0.05}$ | $96.46_{\pm0.26}$ |
| CoNAL [58]           | $98.60_{\pm0.09}$ | $97.89_{\pm0.06}$ | $96.03_{\pm0.73}$ |
| GeoCrowdNet (F) [13] | $\mathbf{98.98_{\pm0.02}}$ | $97.70_{\pm0.71}$ | $96.91_{\pm0.94}$ |
| GeoCrowdNet (W) [13] | $97.33_{\pm0.13}$ | $94.74_{\pm0.67}$ | $90.79_{\pm0.97}$ |
| Ours                 | $\mathbf{99.13_{\pm0.05}}$ | $\mathbf{98.98_{\pm0.11}}$ | $\mathbf{98.80_{\pm0.07}}$ |

**Hyperparameter analysis on the CIFAR100 dataset.** We conduct sensitivity analyses about hyperparameters $r_0$ and $r_1$ on the CIFAR100 dataset, where we choose $r_0$ from $\{3, 5, 10, 15, 20\}$ and $r_1$ from $\{200, 250, 300\}$. As discussed in the implementation details in Section C.1, $r_0$ and $r_1$ determine the magnitude of the threshold $\Omega_t$ at the beginning and the end of the training process, respectively. With higher values of $r_0$ and $r_1$, the accuracy of the corrected labels using algorithm (10) will increase accordingly, but the number of corrected labels will decrease. As shown in Table 3, with different choices of $r_0$ and $r_1$, the proposed method consistently outperforms all the compared methods.

**Classification accuracy on CIFAR100 with varying number of annotators.** Figure 5 shows the average accuracy on CIFAR100 with the number of annotators varying from 5 to 100, which further demonstrate the superiority of the proposed method under different settings.

Table 3: Average accuracy of learning the CIFAR100 dataset with different hyperparameters.

| | | IDN-LOW | IDN-MID | IDN-HIGH |
|---|---|---|---|---|
| $r_0 = 3$ | $r_1 = 200$ | $58.88_{\pm 1.18}$ | $52.98_{\pm 1.02}$ | $45.50_{\pm 1.30}$ |
| | $r_1 = 250$ | $58.19_{\pm 0.92}$ | $52.04_{\pm 1.77}$ | $46.28_{\pm 2.03}$ |
| | $r_1 = 300$ | $58.16_{\pm 0.57}$ | $53.03_{\pm 0.82}$ | $45.48_{\pm 1.97}$ |
| $r_0 = 5$ | $r_1 = 200$ | $59.01_{\pm 0.24}$ | $52.75_{\pm 1.38}$ | $45.76_{\pm 2.02}$ |
| | $r_1 = 250$ | $58.58_{\pm 0.62}$ | $52.76_{\pm 0.99}$ | $46.59_{\pm 1.23}$ |
| | $r_1 = 300$ | $59.34_{\pm 0.66}$ | $52.88_{\pm 1.68}$ | $47.04_{\pm 1.74}$ |
| $r_0 = 10$ | $r_1 = 200$ | $59.39_{\pm 0.44}$ | $53.23_{\pm 1.12}$ | $47.10_{\pm 1.78}$ |
| | $r_1 = 250$ | $59.75_{\pm 0.74}$ | $53.74_{\pm 0.54}$ | $47.27_{\pm 1.79}$ |
| | $r_1 = 300$ | $59.06_{\pm 0.94}$ | $53.73_{\pm 0.93}$ | $48.09_{\pm 1.47}$ |
| $r_0 = 15$ | $r_1 = 200$ | $59.43_{\pm 0.56}$ | $54.46_{\pm 0.37}$ | $47.40_{\pm 1.23}$ |
| | $r_1 = 250$ | $59.38_{\pm 1.22}$ | $54.48_{\pm 0.87}$ | $48.59_{\pm 0.90}$ |
| | $r_1 = 300$ | $59.36_{\pm 0.33}$ | $54.93_{\pm 0.63}$ | $48.29_{\pm 1.85}$ |
| $r_0 = 20$ | $r_1 = 200$ | $59.81_{\pm 0.55}$ | $54.88_{\pm 0.60}$ | $49.44_{\pm 1.30}$ |
| | $r_1 = 250$ | $59.38_{\pm 0.62}$ | $55.48_{\pm 0.75}$ | $49.39_{\pm 1.48}$ |
| | $r_1 = 300$ | $60.14_{\pm 0.87}$ | $55.10_{\pm 0.88}$ | $49.07_{\pm 1.22}$ |

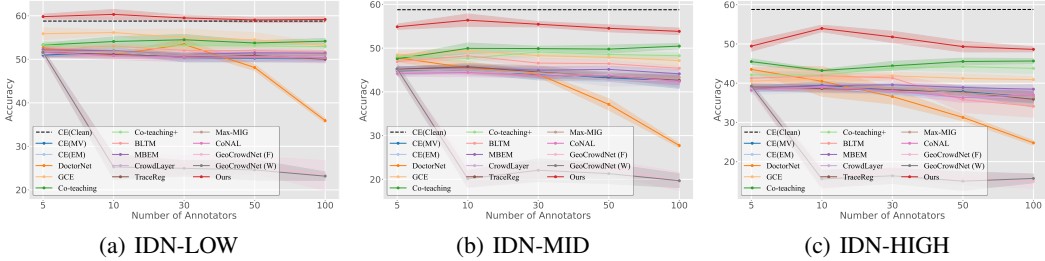

(a) IDN-LOW     (b) IDN-MID     (c) IDN-HIGH

Figure 5: Average accuracy of learning CIFAR-100 dataset with varying number of annotators. The error bar for standard deviation has been shaded.

**Average estimation error.** For synthetic noisy datasets CIFAR10 and CIFAR100, we compare the average estimation error of the proposed method with six competing methods, CrowdLayer [53], TraceReg [8], GeoCrowdNet (F) [13], GeoCrowdNet (W) [13], MBEM [11], and BLTM [17]. The definition of the average estimation error and the results on CIFAR10 with 5 annotators are given in the Ablation study of Section 5. In Figures 6 and 7, we respectively present the average estimation error on the validation set of CIFAR10 and CIFAR100 with varying numbers of annotators, where the results of CrowdLayer, TraceReg, GeoCrowdNet (F) and GeoCrowdNet (W) overlap in some subfigures of Figure 7. The proposed method outperforms all the compared methods with lower average estimation error for each annotator in most of the cases, further demonstrating the effectiveness of the proposed sparse Bayesian model.

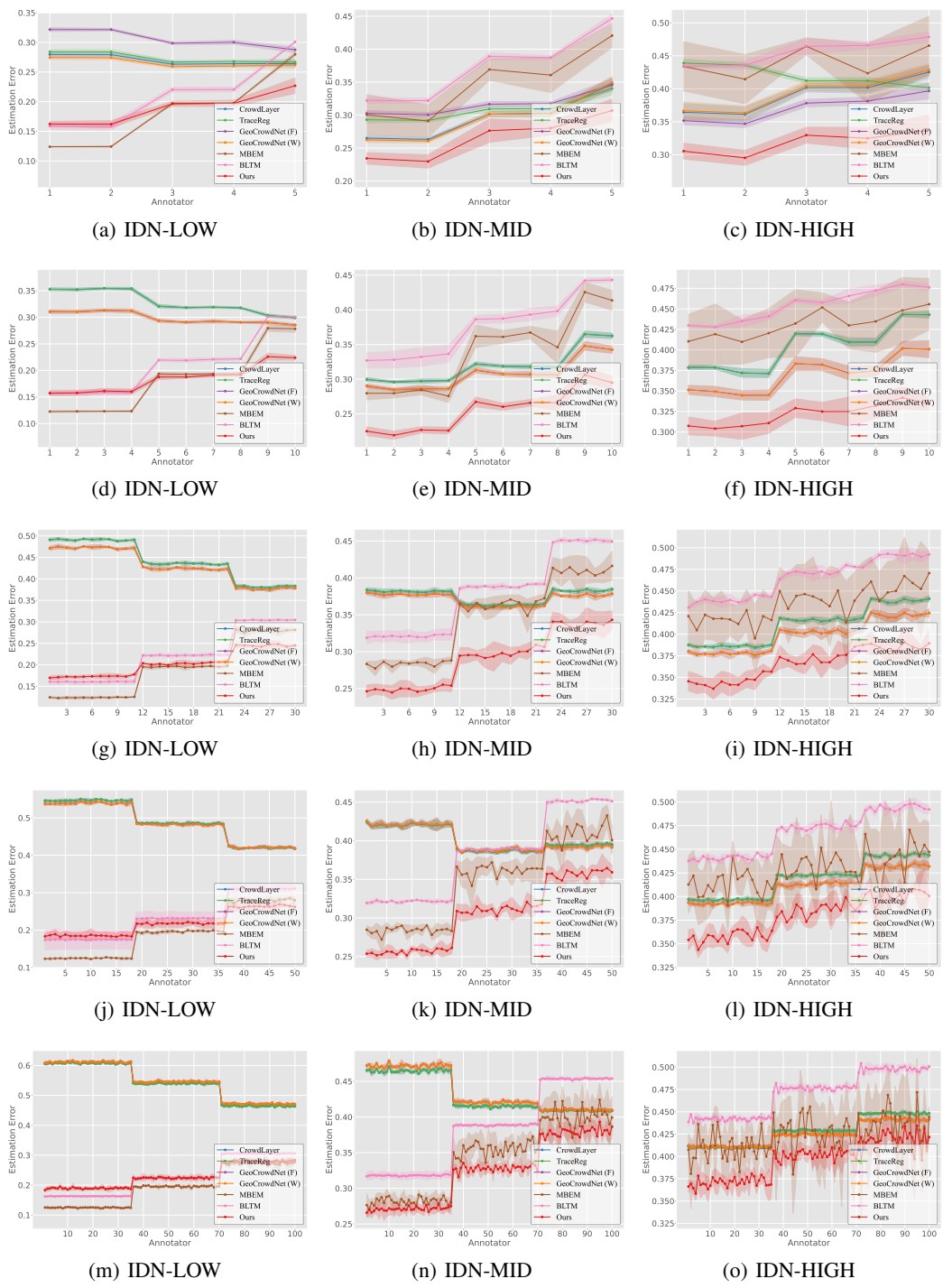

Figure 6: Average estimation error of annotator-specific instance-dependent noise transition matrices on CIFAR10. From the first to the fifth row, the number of annotators is 5, 10, 30, 50 and 100 respectively. Standard errors are represented by shaded regions.

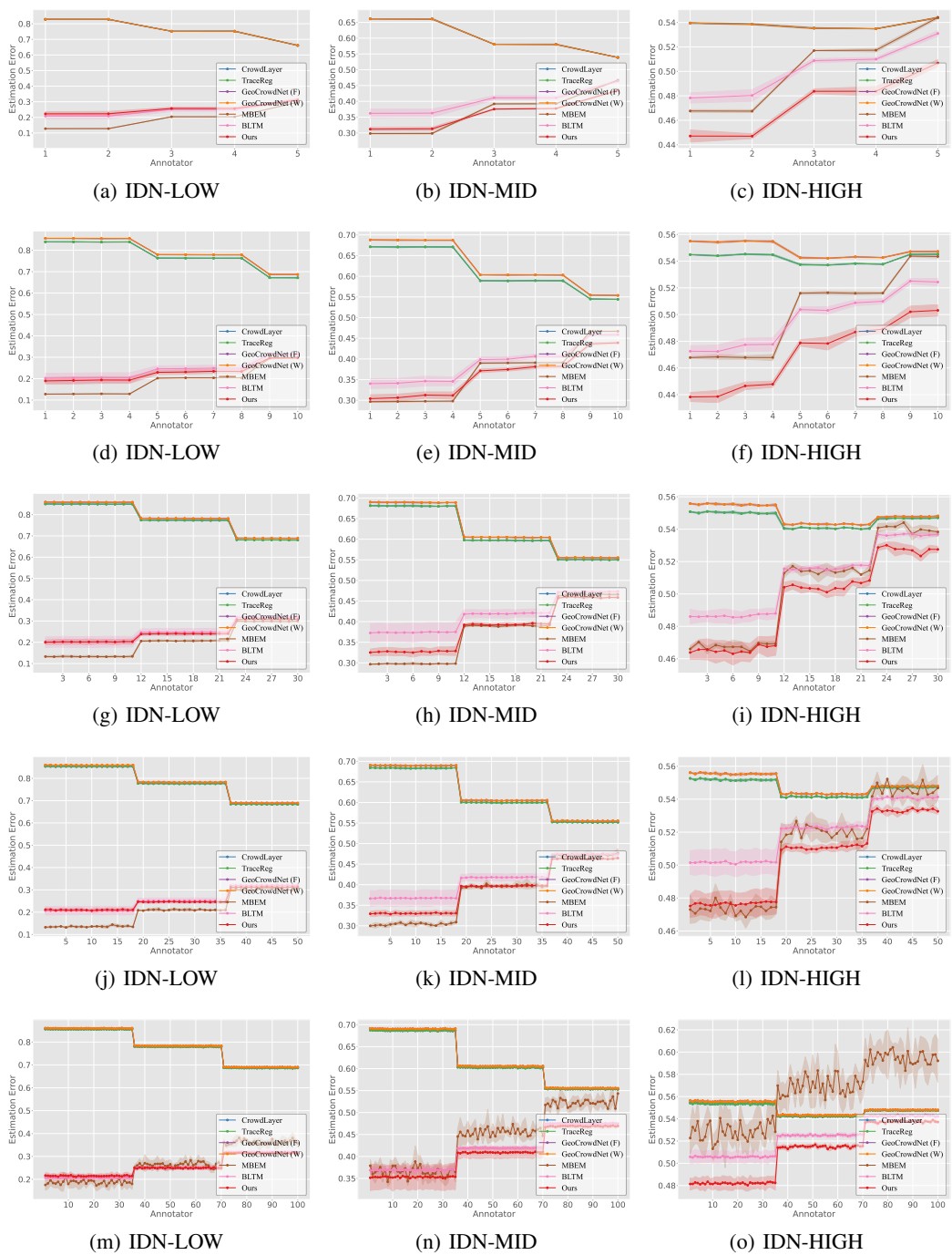

Figure 7: Average estimation error of annotator-specific instance-dependent noise transition matrices on CIFAR100. From the first to the fifth row, the number of annotators is 5, 10, 30, 50 and 100 respectively. Standard errors are represented by shaded regions.

