# OpenReview forum: "Label Correction of Crowdsourced Noisy Annotations with an Instance-Dependent Noise Transition Model"
_NeurIPS.cc/2023/Conference — NeurIPS 2023 poster_

### Official Review · Reviewer_ysQF · 2023-06-28

**Soundness:** 3 good
**Presentation:** 2 fair
**Contribution:** 3 good
**Rating:** 6
**Confidence:** 3

**Summary:**

The authors formulate the noise transition model in a Bayesian framework and design a new label correction algorithm. The authors further formulate the label correction process as a hypothesis testing problem and propose a novel algorithm to infer the true label from the noisy annotations based on the pairwise likelihood ratio test (LRT). The experimental results on benchmark and real-world datasets validate the effectiveness of the proposed approach.

**Strengths:**

1. This paper provides a posterior-concentration theorem, which guarantees the posterior consistency of the noise transition model in terms of the Hellinger distance.

2. This paper formulates the label correction process as a hypothesis testing problem and propose a novel algorithm to infer the true label from the noisy annotations based on the pairwise likelihood ratio test (LRT).

3. This paper presents extensive experiments on benchmark and real-world datasets to validate the effectiveness of the proposed approach.


**Weaknesses:**

The performance of the proposed approach outperforms other algorithms on three image datasets with synthetic annotations, but does not have significant advantages on two datasets with human annotations, CIFAR-10N and LabelMe. The authors do not provide a reasonable explanation for these results. In addition, some of the author's statements in the abstract need to be considered, such as “ Learning an instance-dependent noise transition model, however, is challenging and remains less explored.”.  To my knowledge, there are many algorithms proposed to solve this problem.

**Questions:**

1. From Table 1, the performance of the proposed approach outperforms other algorithms on three image datasets with synthetic annotations, but does not have significant advantages on two datasets with human annotations, CIFAR-10N and LabelMe. The authors have less analytical discussion of the experimental results. The authors should give a fuller discussion of the experimental results and add other datasets with human annotations to verify the validity of the proposed approach.

2. The authors claim that “Learning an instance-dependent noise transition model, however, is challenging and remains less explored.”. But to my knowledge, there are many algorithms proposed to solve this problem. The authors should notice t and cite these papers.

3. The proof section of the text is relatively substantial, but the author's explanation of the motivation of the article is slightly lacking.


**Limitations:**

1. From Table 1, the performance of the proposed approach outperforms other algorithms on three image datasets with synthetic annotations, but does not have significant advantages on two datasets with human annotations, CIFAR-10N and LabelMe. The authors have less analytical discussion of the experimental results. The authors should give a fuller discussion of the experimental results and add other datasets with human annotations to verify the validity of the proposed approach.

2. The authors claim that “Learning an instance-dependent noise transition model, however, is challenging and remains less explored.”. But to my knowledge, there are many algorithms proposed to solve this problem. The authors should notice t and cite these papers.

3. The proof section of the text is relatively substantial, but the author's explanation of the motivation of the article is slightly lacking.

---

> ### Author Rebuttal · Authors · 2023-08-10
>
> Thanks so much for your valuable comments on our paper. Below we address each question in detail.
>
> 1.As shown in Table 1 of our paper, the proposed method outperforms the baseline methods in most cases. We also display the results from Table 1 in Figure 1 of the PDF attachment, which may help demonstrate the benefits of the proposed method more clearly. Concerning the reason why the advantage of our approach is more pronounced on the datasets with synthetic annotations than on the real dataset, it may be pertinent to that we strictly follow the instance-dependent assumption in generating the annotations. To further verify the effectiveness of our method, we now add additional experiments on the MUSIC dataset and the results are shown in Table 2.
>
> 2.Thanks so much for your suggestion. In preparing a revision, we will take your suggestion and include this background information and an explanation of how our work differs from earlier efforts in our paper. Related references will be added too. In addition, to highlight the advantages of the proposed approach, we have now conducted additional empirical analyses by comparing with baseline methods [A-C] which employ instance-dependent matrix; the results are presented in Table 3 in the attached PDF.
>
> 3.Thanks for this thoughtful remark. The motivation and the advantages of the proposed approach can be summarized as follows.
>
> (1) Utilizing such a sparse Bayesian model enables us to consistently estimate the instance-dependent noise transition matrix with limited number of anchor points more. As shown in Figure 1 on page 9 of our paper, the average estimation error of the proposed method is apparently smaller than that of the baseline methods. We now also present the average estimation error under different annotator number settings in Figure 2 of the attached PDF.
>
> (2) Most of existing works related to the estimation of instance-dependent matrix are heuristic and lack theoretical guarantees. [D] made some theoretical progress to justify the use of the trace regularisation, extending the work of [E] in the case of instance-independent noise matrix. However, the theory in [D] only holds for individual samples rather than the population setting. Our work first theoretically characterizes the distance between the noise transition model and the true instance-dependent noise transition matrix in terms of the Hellinger distance, which largely closes the theoretical gaps in the consistency of the noise transition model.
>
> (3) The posterior consistency of the noise transition model enables us to conduct further analysis. In particular, based on the posterior consistency result, we propose an algorithm to infer the true label utilizing the noise transition model, and provide an information-theoretic bound on the Bayes error of the label inference method.
>
> **References**
>
> [A] Learning from Multiple Annotator Noisy Labels via Sample-wise Label Fusion. ECCV 2022
>
> [B] Part-dependent label noise: Towards instance-dependent label noise. NeurIPS, 2020.
>
> [C] Estimating instance-dependent bayes-label transition matrix using a deep neural network. ICML, 2022.
>
>
> [D] Disentangling human error from the ground truth in segmentation of medical images. NeurIPS, 2021.
>
> [E] Learning from noisy labels by regularized estimation of annotator confusion. CVPR, 2019.

---

### Official Review · Reviewer_uiTb · 2023-07-06

**Soundness:** 3 good
**Presentation:** 3 good
**Contribution:** 3 good
**Rating:** 6
**Confidence:** 3

**Summary:**

The paper introduces a method for modelling input-dependent label noise for classification in the presence of multiple annotators. The generative process of annotation is modelled as a hierarchical Bayesian model in which the observed labels are noisy versions of the latent true labels that are corrupted through a noise transition model that is dependent on the input as well as the annotator. The authors propose a method for inferring the true labels based on the approximated noise model  and demonstrate benefits on classification benchmarks with both synthetic and real annotation noise.

**Strengths:**

- The idea of modelling labeling noise as a function of both input and annotator is much more realistic than the majority of related works which drop the input dependence or ignore the differences between annotators.
- They propose a practical algorithm with demonstrable efficacy on classification benchmarks with both synthetic  & real label noise.
- The authors provide a sound theoretical justification for the above algorithm; 1) the concentration theorem shows the "tightness" of the approximation of the noise transition model; 2) formulating inference of true labels as a statistical test gives an information-theoretic bound on the error.




**Weaknesses:**

-  The empirical analysis lacks insights into which components of the annotation model are actually important e.g., how sensitive is the model to a different specification of the prior distribution.
-  There seems to be no comparison with other works that employ instance-specific noise transition models --- e.g., [18] seems to propose a much simpler model that simultaneously learn the instance-specific confusion matrix per annotator and the true label distribution.

**Questions:**

- how does the algorithm scale with the number of annotators?
- how does the accuracy of the method depend on the "sparsity" of labels (e.g., if we had only one label per image, would it still be possible to model the input-dependent noise)?

---

> ### Author Rebuttal · Authors · 2023-08-10
>
> Thanks for reading our paper and providing meaningful feedback. Below we address each question in detail.
>
> **Weaknesses**
>
> 1.The performance of the proposed method is relatively robust to the prior distribution as long as it satisfies the conditions listed in Appendix A.3. Specifically, for computational considerations, we consider the prior $\theta\sim \lambda_n N(0, \sigma^2_{1n})+(1-\lambda_n)N(0, \sigma^2_{0n})$ for each parameter $\theta$, which satisfies the conditions in  Appendix A.3 if the values of $\lambda_n$ and $\sigma_{0n}$ are small enough and $\sigma_{1n}$ is taken a proper value. We provide experimental results with different choices of $\lambda_n$ in Table 1, from which we can observe that the model performance is relatively stable with respect to different values of $\lambda_n$ if $\lambda_n$ is within a reasonable range.
>
> 2.Thanks so much for pointing this out. The reference [18] in our paper is intended for segmentation tasks so that we do not include it as a baseline method in our paper. But your suggestion that we should compare with other works employing instance-specific noise matrix is insightful. We have now added more baseline methods Multi-AnT[A], Part-T[B], and BLTM[C] to highlight the advantages of the proposed approach, and the results are presented in Table 3 in the attached PDF.
>
> **Questions**
>
> 1.In the network structure of our experiments, each annotator corresponds to a linear fully-connected layer, connected after the shared representation layer, and thus accounts for a small percentage of the overall parameters. We consider different settings on the CIFAR10 dataset with the number of annotators set to 10, 20, and 30, respectively. The performance of the proposed method under different settings can be found in Table 4, with the total training hours listed below.
>
> Number of Annotators | IDN-LOW | IDN-MID | IDN-HIGH
> |     ---      |   ---  |   ---  |   ---  |
> |       5       |   2.59  |   2.64  |   2.65  |
> |       10      |   2.67  |   2.62  |   2.63  |
> |       20      |   2.65  |   2.65  |   2.67  |
> |       30      |   2.67  |   2.67  |   2.71  |
>
> 2.To evaluate the methods under incomplete labeling, for each data item, an annotator labels it with probability 0.2.
>
> **References**
>
> [A] Learning from Multiple Annotator Noisy Labels via Sample-wise Label Fusion. ECCV 2022
>
> [B] Part-dependent label noise: Towards instance-dependent label noise. NeurIPS, 2020.
>
> [C] Estimating instance-dependent bayes-label transition matrix using a deep neural network. ICML, 2022.

---

### Official Review · Reviewer_wDkZ · 2023-07-07

**Soundness:** 3 good
**Presentation:** 3 good
**Contribution:** 2 fair
**Rating:** 5
**Confidence:** 3

**Summary:**

The paper deals with the noisy annotation problem with crowdsourcing data. Instead of traditional transition matrices, the paper proposed to learn an instance-dependent transition model that uses a Bayesian network with guaranteed consistent posterior to approximate the noise in labels. The paper then proposes to infer the true label from the noisy annotations and improve the performance based on the corrected labels.

**Strengths:**

1. The paper is clearly motivated and deals with a potentially meaningful problem for supervised learning with noisy annotations.
2. The paper provides a detailed theoretical analysis of the proposed Bayesian model and an information-theoretic bound on the Bayes error.
3. The experiments on MNIST and CIFAR datasets show that the proposed method outperforms the baselines most of the time.


**Weaknesses:**

1. The assumption that an instance-dependent noise model can be learned with limited training data is quite strong. It seems that the proposed method is still associated with preset thresholds like existing methods, which is particularly difficult to set in a noisy problem with multiple annotators.
2. The conditions for Theorem 1 are not listed in the main paper. I think some remarks are necessary for people to have a fair understanding of the proposed theorems.
3. The proposed method does not always outperform baselines, and the advantage is not significant. In some cases, the biggest difference appears in the IDN-MID case, which is not thoroughly investigated or explained.


**Questions:**

1. Can there be a more convincing claim that an instance-dependent noise model can be learned with limited training data?
2. Is the main theoretical result asymptotic? Can there be a clearer message that indicates the dependency on the number of data samples?
3. Can there be more analysis on the reasons for the change of performances with the number of annotators on some datasets?


**Limitations:**

Although the paper does mention limitations and extensions, the impact of modeling annotators and instance-dependent noise model is not adequately addressed. The noisy problem is often associated with biases or fairness concerns, so some careful consideration needs to be made.

---

> ### Author Rebuttal · Authors · 2023-08-10
>
> Thanks for assessing our paper and providing meaningful comments. Below we address each question in detail.
>
> **Weaknesses**
>
> 1.Thanks for your question.
>
> (1) As shown in Appendix A.1 and Appendix A.2, we assume that the underlying noise transition probability can be approximated by a sparse model. This assumption is supported by empirical studies [A-C] which explore different methods on compressing neural networks without impacting performance. Besides, theoretical works [D-E] in approximation theory have established theories guaranteeing uniform approximation rates for a broad family of function classes. Additionally, as shown in Figure 1 on page 9 of our paper, the average estimation error of the proposed method is apparently smaller than that of the baseline methods, which in turn provides empirical evidence for our assumption.
>
> (2) The threshold value in the proposed label correction method is directly related to accuracy of the inferred labels. In our experiment, we start with a smaller value, which is chosen from {5, 10, 15, 20} and linearly increases the value in the training process.
>
> 2.Thank you for this suggestion. We will add remarks on these conditions when preparing a revision shortly. Specifically, the conditions to guarantee the posterior consistency can be roughly categorized into the three parts.
>
> (1) Part 1: the hypothesis set we consider is a class of DNNs.
>
> (2) Part 2: the underlying noise transition probability can be approximated by a sparse model.
>
> (3) Part 3: the prior we use need to satisfy some conditions.
>
> 3.Thanks for your comment. As shown in Table 1 of our paper, the proposed method outperforms the baseline methods in most cases. On the CIFAR-10 and CIFAR-100 datasets, the performance of our method is always the best, but the second-best method is always changing, which may be the reason why the biggest difference is sometimes seen in the IND-MID case. We display the results from Table 1 in Figure 1 of the PDF attachment, which may help to demonstrate the benefits of the proposed method more clearly.
>
> **Questions**
>
> 1. Please see the first point in **Weaknesses**.
>
> 2. Yes. The main theoretical result is asymptotic. To investigate the dependency on the number of data samples $n$, we may look at the conditions on the prior in Appendix A.3. For computational considerations, we may consider the prior $\theta\sim \lambda_n N(0, \sigma^2_{1n})+(1-\lambda_n)N(0, \sigma^2_{0n})$, where the values of $\lambda_n$, $\sigma_{1,n}$ and $\sigma_{0,n}$ depends on $n$, and it can be verified that this prior satisfies the conditions in  Appendix A.3 if the values of $\lambda_n$, $\sigma_{1n}$ and $\sigma_{0n}$ are properly chosen. In particular, the value of $\lambda_n$ is related to the sparsity of the model and we require it to satisfy that $\lambda_n =O(1/J_{nk}[n^{H_{n1}+H_{n2}}(L_{n1}+L_{n2}p_{n})]^{c})$ for some positive constant $c$ and $k=1, 2$, which should be chosen by considering the network structure and the number of data points $n$. The meaning of the notations in the preceding formula is the same as in the paper. This procedure sheds some light on the consideration of the dependency on $n$.
>
> 3. Thanks for your question. We now also consider different settings on the CIFAR10 dataset with the number of annotators set to 10, 20, and 30, respectively. To evaluate the methods under incomplete labeling, for each data item, an annotator labels it with probability 0.2. The performance of the proposed method under different settings can be found in Table 4 and Figure 2, which further demonstrates the effectiveness of the proposed method.
>
> **References**
>
> [A] Learning both weights and connections for efficient neural network. NeurIPS, 2015.
>
> [B] Model compression and acceleration for deep neural networks: The principles, progress, and challenges. IEEE Signal Processing Magazine, 2018.
>
> [C] Parameter efficient training of deep convolutional neural networks by dynamic sparse reparameterization. ICML, 2019.
>
> [D] Optimal approximation with sparsely connected deep neural networks. SIAM Journal on Mathematics of Data Science, 2019
>
> [E] Optimal approximation of piecewise smooth functions using deep ReLU neural networks. Neural Networks, 2018.

---

> > ### Comment · Reviewer_wDkZ · 2023-08-20
> > **Response to Authors**
> >
> > Thank you for providing the rebuttal. I appreciate the answers to address the clarity of conditions. However, although I don't agree with Reviewer i77Y about asking for extremely sparse cases, I do agree with some concerns about the practicalness of the proposed method. I'd like to keep the rating for now to see if the strongly negative opinions persist.

---

> > > ### Author Response · Authors · 2023-08-21
> > > **Thank you for the comments.**
> > >
> > > Thank you for the comments. We present additional experimental results in Tables 1-2 to further illustrate the benefits of the proposed approach.
> > >
> > > *  We conduct more experiments with varying number of annotators (10, 30, 50, 100) and each instance only has one label. Under these settings, we compare our approach with different baseline methods: (1) Instance-dependent method -- "Multi-AnT"; (2) Label correction methods --  "EM" and "MBEM"; and (3) Methods utilizing two networks -- "Co-teaching" and "Co-teaching+". The results are shown in Table 1, where "EM+" and "MBEM+" are the results obtained by replacing the label inference process in our method with EM and MBEM methods, respectively.
> > >
> > > *  In Table 2, we present the accuracy of the inferred labels of our method and EM/MBEM method, where the numbers in the brackets of our method are the amount of data points whose labels are inferred and used for training. The results in Table 2 show the high accuracy of the inferred labels with our method and further exhibit the advantages of the proposed label inference method.
> > >
> > > We hope our response can address your concerns.
> > >
> > > #### Experimental results.
> > >
> > >
> > > **Table 1:** Average accuracy of learning the CIFAR10 dataset with different number of annotators.
> > >
> > > |  |  | Ours | Multi-AnT | MBEM | MBEM+ | EM+ | Co-teaching | Co-teaching+ |
> > > | :---- | :---- | :---- | :---- | :---- | :---- | :---- | :---- | :---- |
> > > | IDN-LOW  | 10 | $87.34_{\pm 0.27}$ | $85.23_{\pm 0.42}$ | $82.60_{\pm 0.36}$ | $82.58_{\pm 0.24}$ | $81.02_{\pm 0.88}$ | $84.23_{\pm 0.56}$ | $84.32_{\pm 0.60}$ |
> > > |          | 30 | $86.94_{\pm 1.15}$ | $84.36_{\pm 0.58}$ | $82.18_{\pm 0.62}$ | $82.19_{\pm 0.21}$ | $80.88_{\pm 0.48}$ | $83.69_{\pm 0.54}$ | $84.27_{\pm 0.40}$ |
> > > |          | 50 | $86.78_{\pm 0.68}$ | $84.53_{\pm 0.75}$ | $82.00_{\pm 0.33}$ | $81.85_{\pm 0.53}$ | $80.18_{\pm 0.32}$ | $84.39_{\pm 0.82}$ | $84.90_{\pm 0.50}$ |
> > > |          | 100 | $86.67_{\pm 0.35}$ | $84.49_{\pm 0.32}$ | $81.71_{\pm 0.45}$ |$81.34_{\pm 0.52}$ | $80.18_{\pm 0.42}$ | $84.49_{\pm 0.50}$ | $84.34_{\pm 0.73}$ |
> > > | IDN-MID  | 10 | $86.58_{\pm 0.62}$ | $81.14_{\pm 1.10}$ | $77.96_{\pm 0.32}$ | $78.68_{\pm 0.52}$ | $76.61_{\pm 0.41}$ | $80.97_{\pm 0.82}$ | $81.28_{\pm 0.67}$ |
> > > |          | 30 | $85.67_{\pm 0.55}$ | $80.74_{\pm 0.50}$ | $77.61_{\pm 0.48}$ | $78.37_{\pm 0.51}$ | $76.51_{\pm 0.32}$ | $81.14_{\pm 0.70}$ | $81.72_{\pm 0.40}$ |
> > > |          | 50 | $85.68_{\pm 0.41}$ | $80.06_{\pm 0.66}$ | $77.46_{\pm 0.74}$ | $78.17_{\pm 0.17}$ | $76.28_{\pm 0.63}$ | $81.47_{\pm 0.75}$ | $81.19_{\pm 0.38}$ |
> > > |          | 100 | $84.99_{\pm 0.59}$ | $79.26_{\pm 0.42}$ | $76.62_{\pm 0.28}$ | $78.06_{\pm 0.73}$ | $75.24_{\pm 0.66}$ | $81.65_{\pm 0.21}$ | $81.30_{\pm 0.39}$ |
> > > | IDN-HIGH | 10 | $84.68_{\pm 0.37}$ | $76.64_{\pm 0.68}$ | $71.98_{\pm 1.36}$ | $73.95_{\pm 1.86}$ | $66.82_{\pm 2.19}$ | $76.47_{\pm 1.24}$ | $77.45_{\pm 1.11}$ |
> > > |          | 30 | $83.05_{\pm 0.41}$ | $73.57_{\pm 0.99}$ | $72.73_{\pm 0.76}$ | $75.32_{\pm 1.15}$ | $70.52_{\pm 0.80}$ | $77.57_{\pm 0.76}$ | $77.89_{\pm 0.06}$ |
> > > |          | 50 | $82.66_{\pm 0.33}$ | $73.78_{\pm 1.13}$ | $72.85_{\pm 0.65}$ | $74.51_{\pm 0.82}$ | $70.69_{\pm 0.75}$ | $77.79_{\pm 0.74}$ | $77.87_{\pm 0.88}$ |
> > > |          | 100 | $81.71_{\pm 1.19}$ | $71.47_{\pm 1.37}$ | $72.70_{\pm 0.90}$ | $74.08_{\pm 0.94}$ | $70.73_{\pm 0.79}$ | $78.25_{\pm 0.77}$ | $78.50_{\pm 0.45}$ |
> > >
> > > **Table 2:** Average accuracy of the inferred labels.
> > >
> > > |  |  | Ours | MBEM/EM |
> > > | :---- | :---- | :---- | :---- |
> > > | IDN-LOW  | 10 | $97.98_{\pm 0.22}$ (39417) | $80.26_{\pm 0.51}$ |
> > > |          | 30 | $98.01_{\pm 0.23}$ (38271) | $78.20_{\pm 1.58}$ |
> > > |          | 50 | $97.30_{\pm 0.59}$ (39104) | $79.37_{\pm 0.17}$ |
> > > |          | 100 | $97.33_{\pm 0.37}$ (38478) | $77.35_{\pm 2.68}$ |
> > > | IDN-MID  | 10 | $97.70_{\pm 0.42}$ (36579) | $62.81_{\pm 0.37}$ |
> > > |          | 30 | $96.65_{\pm 0.38}$ (37316) | $62.22_{\pm 0.19}$ |
> > > |          | 50 | $96.21_{\pm 0.37}$ (38110) | $61.67_{\pm 0.38}$ |
> > > |          | 100 | $96.65_{\pm 0.37}$ (35528) | $61.61_{\pm 0.30}$ |
> > > | IDN-HIGH | 10 | $95.18_{\pm 0.65}$ (37394) | $48.66_{\pm 0.37}$ |
> > > |          | 30 | $94.03_{\pm 0.47}$ (37040) | $49.08_{\pm 0.21}$ |
> > > |          | 50 | $93.51_{\pm 0.30}$ (36498) | $48.89_{\pm 0.25}$ |
> > > |          | 100 | $94.13_{\pm 0.42}$ (34347) | $48.85_{\pm 0.12}$ |

---

### Official Review · Reviewer_PGxM · 2023-07-07

**Soundness:** 4 excellent
**Presentation:** 3 good
**Contribution:** 3 good
**Rating:** 6
**Confidence:** 2

**Summary:**

This paper focuses on the problem of estimating annotator-specific instance-dependent transition matrices. The authors propose a solution based on a family of Bayesian estimators to approximate these matrices, supported by theoretical foundations. Through rigorous mathematical proofs, the authors demonstrate that their estimator can effectively approximate transition matrices under claimed mild conditions. Additionally, they introduce a novel label correction method with bounded Bayes error. To evaluate the effectiveness of their proposed method, the authors conduct experiments using three commonly used benchmarks and synthetic annotations. Overall, this paper has potential significant insights that could benefit the community. The experiment evaluation is not very convincing, but I'm tending to accept it at this stage.




**Strengths:**

There are several main theoritical contributions in this paper:

1. Authors proposed a sparse Bayesian approach to learn the annotator-specific instance-dependent transition matrices to model the relationship between label noise and crowdsourced annotators.

2. The proposed estimator is claimed to approxmiate true transition processs with guarantee (minimizes Hellinger distance).

3. Based on the derived theorem, the authors also provided a novel correction algorithm named that leverages likelihood ratio test, an upper-bound of the Bayes error for the proposed algorithm is provided.

With the approxmiation guarantee and error-bound on the correction algorithm, the proposed method is therefore considered to have significant theoritical implications and considerable novelties, provided that the proof are rigourous and correct, which is under further checking.

**Weaknesses:**

1. The section numbers provided in the appendix do not correspond to the references in the main paper, where are assumptions A.1-A.4 in A.1?

2. Compared baselines are not known for their advantages in modeling instance-dependent label noise, methods such as PartT [1], BLTM [2] should be compared instead. Also, SOTA methods based on geometrical properties such as [3,4] should also be compared.

3. One of the main issues is that the use of sparse Bayesian approach is not strongly motivated, it remains unclear to me why using sparse Bayesian estimators is better or non-trivial compared with existing works.

[1] Xia, Xiaobo, et al. "Part-dependent label noise: Towards instance-dependent label noise." Advances in Neural Information Processing Systems 33 (2020): 7597-7610.

[2] Yang, Shuo, et al. "Estimating instance-dependent bayes-label transition matrix using a deep neural network." International Conference on Machine Learning. PMLR, 2022.

[3] Li, Xuefeng, et al. "Provably end-to-end label-noise learning without anchor points." International conference on machine learning. PMLR, 2021.

[4] Yong, L. I. N., et al. "A Holistic View of Label Noise Transition Matrix in Deep Learning and Beyond." The Eleventh International Conference on Learning Representations. 2022.



**Questions:**

1. Why are the results on CE trained with clean labels so low? This is inconsistent with prior works.

2. How are the sythentic annotations generated?



**Limitations:**

N.A.

---

> ### Author Rebuttal · Authors · 2023-08-10
>
> **Weaknesses**
>
> 1.Thanks so much for pointing this out. The assumptions A.1-A.4 are in Appendix A.2 and we will revise this error in the final version of our paper. We apologize for the misreference and thanks again for reading our paper carefully.
>
> 2.We have now added more baseline methods PartT [1], BLTM [2] and VolMinNet[3], and the results are presented in Table 3 in the attached PDF.
>
> 3.Thanks for this thoughtful remark. The motivation and the advantages of the proposed approach can be summarized as follows.
>
> (1) Utilizing such a sparse Bayesian model enables us to consistently estimate the instance-dependent noise transition matrix with limited number of anchor points. As shown in Figure 1 on page 9 of our paper, the average estimation error of the proposed method is apparently smaller than that of the baseline methods. We also present the average estimation error under different annotator number settings in Figure 2 of the attached PDF.
>
> (2) Most of existing works related to the estimation of instance-dependent matrix are heuristic and lack theoretical justifications. [A] made some theoretical progress to justify the use of the trace regularisation, which extends the work of [B] for the case of instance-independent noise matrix. However, the theory in [A] only holds for individual samples, rather than the population setting. Our work first theoretically characterizes the distance between the noise transition model and the true instance-dependent noise transition matrix in terms of the Hellinger distance, which largely closes the theoretical gaps in the consistency of the noise transition model.
>
> (3) The posterior consistency of the noise transition model enables us to conduct further analysis. In particular, based on the posterior consistency result, we propose an algorithm to infer the true label utilizing the noise transition model, and provide an information-theoretic bound on the Bayes error of the label inference method.
>
> **Questions**
>
> 1. This could be attributed to the different choices of the network structure and the number of training epochs. In particular, for CIFAR-10 and CIFAR-100 datasets, we choose ResNet-18 and ResNet-34 architecture and train the network for 120 and 150 epochs respectively. And we train the model from scratch instead of using pre-trained weights.
>
> 2. We generate three groups of annotators with varying expertise as described on page 8 of our paper, where 5 annotators are in each group. For each annotator, we generate the instance-dependent noisy annotations according to Algorihtm 2 in [C].
>
> **References**
>
> [A] Disentangling human error from the ground truth in segmentation of medical images. NeurIPS, 2021.
>
> [B] Learning from noisy labels by regularized estimation of annotator confusion. CVPR, 2019.
>
> [C] Xia, Xiaobo, et al. "Part-dependent label noise: Towards instance-dependent label noise." Advances in Neural Information Processing Systems 33 (2020): 7597-7610.

---

### Official Review · Reviewer_i77Y · 2023-07-07

**Soundness:** 2 fair
**Presentation:** 3 good
**Contribution:** 2 fair
**Rating:** 3
**Confidence:** 4

**Summary:**

This paper proposes a new method for label correction of crowdsourced data via an instance-dependent noise transition model. It parameterizes the instance-dependent noise transition matrices by a Bayesian network. A theoretical analysis that guarantees the posterior consistency of noise transition matrices is provided.   A new label correction way based on the instance-dependent noise transition matrices is also provided. Experiments show the proposed method can achieve better performance than some existing methods.

**Strengths:**

1. Estimating the instance-dependent noise transition matrices for crowdsourcing is an important topic.
2. This paper provides some theoretical analyses to understand and guarantee some useful properties of the proposed method.
3. Experiments on synthetic and real crowdsourced datasets verified the effectiveness of the proposed methods.

**Weaknesses:**

1.  General instance-dependent noise matrix is not new for crowdsourcing. For example, several prior methods [1-5] have studied it in the early period. Recently, this topic receives more attention [6-8], due to the increasing demand for large-scale well-annotated datasets. I think the paper should include these background and existing works, and discuss the difference with them.

2.  The conditions to guarantee the posterior consistency of the noise transition model seems very complex. How can we know such conditions are met in real scenes? Besides, since the anchor point assumption is strong in label-noise learning [9-11], the requirement of the proposed method that a set of anchor points can be used for training the noise transition model seems much more difficult.

3.  The experiments are insufficient in terms of the following aspects:
- The number of annotators is usually large, while the experiments on synthetic data only assume 5 annotators, which is not a typical setting.
- Also due to the small number of annotators, the annotations are not sparse for each annotator as usual case.
- I suggest the authors compare the proposed method with some baselines that model instance-dependent noise matrices [6-8], and more truth inference baselines [12-14] to show its advantage more clearly.
- Since the experimental datasets are all image classification datasets, it would be better to do some experiments on the dataset from other modalities, e.g. Music dataset [15].

4.  Lack of ablation discussion on label correction way. How to use noise transition for inferring clean labels has been explored in previous works [16,13], I think it is necessary to discuss and compare the proposed label correction way with the existing truth inference methods with noise transition matrix to when to use such label correction way.

[1] The multidimensional wisdom of crowds. NeurIPS 2010

[2] Exploiting structure in crowdsourcing tasks via latent factor models. Citeseer. 2010

[3] Modeling annotator expertise: Learning when everybody knows a bit of something. AISTATS 2010

[4] Learning from multiple annotators with varying expertise. MLJ 2014

[5] Learning to Predict from Crowdsourced Data. UAI 2014

[6] Disentangling human error from the ground truth in segmentation of medical images. NeurIPS 2020

[7] Learning from Multiple Annotator Noisy Labels via Sample-wise Label Fusion. ECCV 2022

[8] Beyond confusion matrix: learning from multiple annotators with awareness of instance features. MLJ 2023

[9] Are Anchor Points Really Indispensable in Label-Noise Learning? NeurIPS 2019

[10] Provably End-to-end Label-noise Learning without Anchor Points. ICML 2021

[11] Estimating Noise Transition Matrix with Label Correlations for Noisy Multi-Label Learning. NeurIPS 2022

[12] Bayesian classifier combination. AISTATS 2012

[13] Learning from noisy singly-labeled data. ICLR 2018

[14]  Exploiting worker correlation for label aggregation in crowdsourcing. ICML 2019

[15] Gaussian process classification and active learning with multiple annotators. In ICML 2014

[16] Maximum likelihood estimation of observer error-rates using the em algorithm. Journal of the Royal Statistical Society, 1979

**Questions:**

See above.

**Limitations:**

The authors have discussed the limitations at the end of the main body, while I think the demand of a set of anchor points is difficult to meet in real-world scenes.

---

> ### Author Rebuttal · Authors · 2023-08-10
>
> Thank you for providing such a comprehensive review.
>
> 1.We appreciate your thoughtful remarks. We will include this background and an explanation of how our work differs from earlier works in our paper. Specifically, existing methods investigate the human annotation process and propose different models to estimate the instance-dependent noise matrix. [1-5] use traditional classification models such as logistic regression, and [6-8] are works in the context of large datasets and deep models. However, [1-5] and [7-8] are heuristic and lack theoretical guarantees in estimating instance-dependent noise matrix.  [6] makes some theoretical progress to justify the use of the trace regularisation, and extends the work of [A] in the case of instance-independent noise matrix. However, the theory in [6] only holds for individual samples, rather than the population setting.  In contrast to existing papers, our work first theoretically characterizes the distance between the noise transition model and the true instance-dependent noise transition matrix, which largely complements the theoretical gaps in the consistency of the noise transition model. Moreover, based on the posterior consistency result, we propose an algorithm to infer the true label utilizing the noise transition model.
>
> 2.Thank you for your comments. We will add remarks on these conditions for clarifications in our paper. Specifically, the conditions to guarantee the posterior consistency can be roughly categorized into the three parts.
>
> (1) Part 1: the hypothesis set we consider is a class of DNNs.
>
> (2) Part 2: the underlying noise transition probability can be approximated by a sparse model.
>
> (3) Part 3: the prior we use should satisfy some conditions.
>
> The condition (1) is a general setting in literature. Below we illustrate how conditions (2) and (3) are met in real scenes.
>
> Regarding (2), existing works [B-D] empirically show that large DNNs contain a large number of redundant parameters and propose various methods on compressing neural networks without impacting performance. Moreover, theoretical works [E-G] in approximation theory establish theories guaranteeing uniform approximation rates for a broad family of function classes.
>
> Concerning (3) we can consider the prior $\theta\sim \lambda_n N(0, \sigma^2_{1n})+(1-\lambda_n)N(0, \sigma^2_{0n})$ for each parameter $\theta$ in the training process. By using techniques such as Mill's ratio, we prove that this prior satisfies the conditions in  Appendix A.3 if the values of $\lambda_n$ and $\sigma_{0n}$ are small enough and $\sigma_{1n}$ is taken a proper value.
>
> The anchor point assumption is used in our paper for theoretical consideration, which, however,  can be relaxed to be the existence of a set of points belonging to the $k$th class with high probability (denoted $1-\delta_n$) for $k\in[K]$. We prove that a similar result to Theorem 1 still holds, with the distance $d_n$ multiplied by a term related to $1-\delta_n$. Moreover, we add an additional baseline method VolMinNet[10] which does not exploit anchor points and present the result in Table 3. We observe that our approach still outperform this baseline, which further verifies the effectiveness of the proposed method.
>
> 3.We take your suggestions and run additional experimentals. The results can be found in the attached PDF.
>
> (1) \& (2)  We now consider different settings on the CIFAR10 dataset with the number of annotators set to 10, 20, and 30, respectively. To evaluate the methods under incomplete labeling, for each data item, we consider the case where an annotator labels it with probability $p=0.2$. The performance of the proposed method under different settings can be found in Table 4 and Figure 2. Due to the time constraint, we have not obtained all results yet; we are still running some experiments. In the subsequent discussion period, we will give the results of the experiments with varying values of $p$.
>
> (3) The experimental results with additional baselines Multi-AnT[7] and MBEM [13] are shown in Table 3.
>
> (4) We now present the results on the MUSIC dataset in Table 2.
>
> 4.Thanks for your insightful comments and we will add this part in the related work of our paper. The EM[16] and MBEM[13] methods are proposed under the instance-independent transition matrix setting, while our method is developed in the instance-dependent case. In the process of truth inference, we all need to utilize the transition matrix, but the difference is that the EM and MBEM methods directly select the inferred label as the label value corresponding to the maximum label posterior distribution. Our method, however, conduct the pairwise likelihood ratio test between different label values and we select only labels with the likelihood ratio greater than a pre-specified threshold to train the classifier, thus greatly increasing the accuracy of the inferred labels. The efficiency of the proposed approach is further confirmed by the performance comparison with the EM and MBEM methods, as shown in Table 3.
>
> **References**
>
> [A] Learning from noisy labels by regularized estimation of annotator confusion. CVPR, 2019.
>
> [B] Optimal approximation with sparsely connected deep neural networks. SIAM Journal on Mathematics of Data Science, 2019.
>
> [C] Learning both weights and connections for efficient neural network. NeurIPS, 2015.
>
> [D] Model compression and acceleration for deep neural networks: The principles, progress, and challenges. IEEE Signal Processing Magazine, 2018.
>
> [E] Parameter efficient training of deep convolutional neural networks by dynamic sparse reparameterization. ICML, 2019.
>
> [F] Optimal approximation with sparsely connected deep neural networks. SIAM Journal on Mathematics of Data Science, 2019
>
> [G] Optimal approximation of piecewise smooth functions using deep ReLU neural networks. Neural Networks, 2018.

---

> > ### Comment · Reviewer_i77Y · 2023-08-19
> >
> > Thanks for the detailed response of authors. After carefully reading the response and checking the source codes, I still have following questions:
> >
> > 1. Expect for the existent of anchor points, how to collect a set of anchor points is unclear and has no guarantee. According to the source codes, this paper collected anchor points by selecting some points with the estimated noisy class posteriors exceeding a certain threshold. Even if anchor points exist in the training dataset, as far as I know, this way only has the guarantee when faced with instance-independent label noise [1,2]. While, when faced with instance-dependent label noise, without other assumptions, collecting a set of anchor points by such way is not guaranteed. Besides, according to the source codes, this work utilized the inferred true labels of anchor points for training the transition matrix models, which I didn’t find this practice explicitly in the paper. And this practice also raises more questions: could the accuracy of inferred true labels of anchor points be guaranteed, and if the inferred true labels is wrong, how it will influence the proposed method. I think the authors should largely improve their analysis to clarify these problems.
> >
> > 2. The annotations in the experiments are still not sparse as real-world cases. As seen in CIFAR10-N and CIFAR100-N, the number of annotations of each annotator is only about hundreds. Besides, could the proposed method work well when each instance only has one label, which is also a typical case in related works[3,4,5]. I more want to see the learning performance in CIFAR-100 with sparse annotations.
> >
> > 3. The proposed framework seems computationally complex, which utilizes two networks to model instance-dependent noise transition model. An analysis of computational complexity is suggested. It will be great to know how much we pay to improve the performance with new proposed method.
> >
> > 4. As for the label correction way, does the choice of the pre-specified threshold matter and how the proposed way perform better than that in MBEM and EM are still unclear. I suggest the author to conduct the experiments by replacing the proposed label correction way with that in MBEM and EM to show the effectiveness of the proposed label correction way directly.
> >
> > 5. Since the proposed method trains two classifiers to reciprocally perform the label correction, the comparison with other single-classifier methods will be unfair to some extent. The authors should conduct the ablation study of this factor.
> >
> > [1] Classification with noisy labels by importance reweighting. TPAMI 2015
> >
> > [2]  Making deep neural networks robust to label noise: A loss correction approach. CVPR 2017
> >
> > [3] Learning from noisy singly-labeled data. ICLR 2018
> >
> > [4] Learning From Noisy Labels By Regularized Estimation Of Annotator Confusion. CVPR 2019
> >
> > [5] Learning from crowds with sparse and imbalanced annotations. MLJ 2022

---

> > > ### Author Response · Authors · 2023-08-21
> > > **Thank you for the follow-up questions.**
> > >
> > > Thank you for the follow-up questions. Below are our responses and hope they can address your concerns and clarify the miscommunication.
> > >
> > > #### 1. The anchor point assumption.
> > >
> > > * As mentioned in our previous response (point 2), the anchor point assumption can be relaxed. More concretely (and thanks to your question, which motivates us to extend our analysis), we further extend our theoretical result (Theorem 1) to a more relaxed condition by re-defining the (_pseudo_) anchor point of class $k$ as $\mathbb{P}(y=k|\mathbf{x}) \ge 1-\delta_n$ and denoting $D_{\delta_n}$ as the set of pseudo anchor points accordingly, where $\delta_n$ characterizes the relaxation of our assumption. Then, we can modify Theorem 1 and theoretically prove that (detailed proof will be included in our revised manuscript) the following holds:
> > > $$\Pi[\theta\in\Theta:d_n(\theta, \theta_0)>M_n \epsilon_n+C\delta_n|D_{\delta_n}]\rightarrow 0$$
> > > in $\mathbb{P}^{n}_{\delta_n}$ probability with $C$ denoting a constant. From the modified theorem, we can observe that as $\delta_n$ converges slower, the Hellinger distance of the transition model and the true transition probability converges to zero at a slower rate. In other words, we theoretically justify that **the transition model will still converge even if collecting a set of anchor points is not guaranteed (with the cost of a slower rate)**.
> > >
> > > * On the empirical side, the accuracy of the pseudo anchor points for training the transition model is higher than 90% in all the cases (Table 1), and the theoretical analysis of relaxing the anchor point assumption is given in the above bullet point.
> > >
> > > * We do not use the inferred labels to train the transition matrix model. The inferred labels are only used for training the base classifiers (Line 559 - Line 743 in train_proposed_plus.py) when the training of the transition matrix (Line 419 - Line 555 in train_proposed_plus.py) is finished.
> > >
> > > #### 2. More sparse case.
> > >
> > > * We conduct more experiments with varying number of annotators (10, 30, 50, 100) and each instance only has one label. Under these settings, we compare our approach with different baseline methods, including instance-dependent method [A], label inference methods [B-C], and methods trained on two classifiers [D-E]. The results are given in Table 2, which exhibit the advantages of the proposed method.
> > >
> > > * To further investigate the influence of the sparsity of annotations on the proposed method, we also conduct experiments with different number of annotations (1, 3, 5, 7, 9) for each instance and the results are presented in Table 3. As we increase the number of annotations, the performance of the proposed method is similar to the training result with access to the true labels.
> > >
> > > * Since it is already the last two days of the discussion period, we don't have enough time to complete the experiments on the CIFAR100 dataset. All the results below are obtained on CIFAR10. We will add more experiments on the CIFAR100 dataset in the new version of our paper.
> > >
> > > #### 3 & 5. Use of two networks.
> > >
> > > * We don't use two networks to model instance-dependent noise transition model. The two networks are used as base classifiers and reciprocally provide prior information in the label inference method, and the training of the transition model is already finished at this stage. According to our records, the total training time of our method on cifar10 is about 2.8 hours, which is about twice as long as that of the majority vote method (~ 1.3 hours) 1.7 times as that of MBEM (~1.6 hours), and about the same as the training time of other methods using two metworks. The total training time for the above mentioned methods on CIFAR10 with 100 annotators is as follows.
> > >
> > > | CE(MV) | MBEM | Co-teaching | Co-teaching+ | Ours |
> > > | :---- | :---- | :---- | :---- | :---- |
> > > |  $1.31_{\pm 0.04}$ | $1.62_{\pm 0.03}$  | $2.56_{\pm 0.09}$  | $2.69_{\pm 0.07}$  | $2.78_{\pm 0.04}$  |
> > >
> > >
> > > * There are indeed other methods in the existing literature [D-E] that utilize two networks, and for a fairer comparison, we also use them as baseline methods, denoted "Co-teaching" and "Co-teaching+" in Table 2. Moreover, when we compare the label correction methods later, we also use two networks to train and reciprocally perform label correction ("MBEM+" and "EM+" in Table 2). As shown in Table 2, our method still outperforms the baseline methods when two networks are employed.

---

> > > > ### Author Response · Authors · 2023-08-21
> > > >
> > > > 4. Comparison with label correction methods.
> > > >
> > > > * As requested by the reviewer, we replace the proposed label correction method in our code with that in MBEM and EM, denoted "MBEM+" and "EM+", respectively. The experimental results are shown in Table 2. The performance of MBEM+ is better than that of MBEM in most cases, especially when the noise rate is relatively high, but still not as good as the proposed method. Note that we use two classifiers in the training of EM+ and MBEM+.
> > > >
> > > > * To further compare the proposed label correction method with EM/MBEM, we present the accuracy of the inferred labels in Table 4, where the numbers in the brackets of our method are the amount of data points whose labels are inferred and used for training. The results in Table 4 show the high accuracy of the inferred labels with our method and further demonstrate the advantages of the proposed approach.
> > > >
> > > > * In practice, we first take a smaller value for the threshold (from {5, 10, 15, 20}) and then linearly increase the value in the training process. Hyperparameter analysis on the CIFAR100 dataset can be found in Appendix C.2.
> > > >
> > > >
> > > > **Table 1:** Average accuracy of the collected pseudo anchor points.
> > > >
> > > > |  |  | Ours |
> > > > | :---- | :---- | :---- |
> > > > | IDN-LOW  | 10 | $93.77_{\pm 0.22}$ |
> > > > |          | 30 | $93.74_{\pm 0.49}$ |
> > > > |          | 50 | $93.98_{\pm 0.53}$ |
> > > > |          | 100 | $94.02_{\pm 0.32}$ |
> > > > | IDN-MID  | 10 | $95.33_{\pm 0.45}$ |
> > > > |          | 30 | $95.18_{\pm 0.55}$ |
> > > > |          | 50 | $95.78_{\pm 0.55}$ |
> > > > |          | 100 | $95.59_{\pm 0.24}$ |
> > > > | IDN-HIGH | 10 | $92.72_{\pm 1.45}$ |
> > > > |          | 30 | $94.24_{\pm 0.50}$ |
> > > > |          | 50 | $93.61_{\pm 0.61}$ |
> > > > |          | 100 | $93.55_{\pm 0.32}$ |
> > > >
> > > > **Table 2:** Average accuracy of learning the CIFAR10 dataset with different number of annotators.
> > > >
> > > > |  |  | Ours | Multi-AnT | MBEM | MBEM+ | EM+ | Co-teaching | Co-teaching+ |
> > > > | :---- | :---- | :---- | :---- | :---- | :---- | :---- | :---- | :---- |
> > > > | IDN-LOW  | 10 | $87.34_{\pm 0.27}$ | $85.23_{\pm 0.42}$ | $82.60_{\pm 0.36}$ | $82.58_{\pm 0.24}$ | $81.02_{\pm 0.88}$ | $84.23_{\pm 0.56}$ | $84.32_{\pm 0.60}$ |
> > > > |          | 30 | $86.94_{\pm 1.15}$ | $84.36_{\pm 0.58}$ | $82.18_{\pm 0.62}$ | $82.19_{\pm 0.21}$ | $80.88_{\pm 0.48}$ | $83.69_{\pm 0.54}$ | $84.27_{\pm 0.40}$ |
> > > > |          | 50 | $86.78_{\pm 0.68}$ | $84.53_{\pm 0.75}$ | $82.00_{\pm 0.33}$ | $81.85_{\pm 0.53}$ | $80.18_{\pm 0.32}$ | $84.39_{\pm 0.82}$ | $84.90_{\pm 0.50}$ |
> > > > |          | 100 | $86.67_{\pm 0.35}$ | $84.49_{\pm 0.32}$ | $81.71_{\pm 0.45}$ |$81.34_{\pm 0.52}$ | $80.18_{\pm 0.42}$ | $84.49_{\pm 0.50}$ | $84.34_{\pm 0.73}$ |
> > > > | IDN-MID  | 10 | $86.58_{\pm 0.62}$ | $81.14_{\pm 1.10}$ | $77.96_{\pm 0.32}$ | $78.68_{\pm 0.52}$ | $76.61_{\pm 0.41}$ | $80.97_{\pm 0.82}$ | $81.28_{\pm 0.67}$ |
> > > > |          | 30 | $85.67_{\pm 0.55}$ | $80.74_{\pm 0.50}$ | $77.61_{\pm 0.48}$ | $78.37_{\pm 0.51}$ | $76.51_{\pm 0.32}$ | $81.14_{\pm 0.70}$ | $81.72_{\pm 0.40}$ |
> > > > |          | 50 | $85.68_{\pm 0.41}$ | $80.06_{\pm 0.66}$ | $77.46_{\pm 0.74}$ | $78.17_{\pm 0.17}$ | $76.28_{\pm 0.63}$ | $81.47_{\pm 0.75}$ | $81.19_{\pm 0.38}$ |
> > > > |          | 100 | $84.99_{\pm 0.59}$ | $79.26_{\pm 0.42}$ | $76.62_{\pm 0.28}$ | $78.06_{\pm 0.73}$ | $75.24_{\pm 0.66}$ | $81.65_{\pm 0.21}$ | $81.30_{\pm 0.39}$ |
> > > > | IDN-HIGH | 10 | $84.68_{\pm 0.37}$ | $76.64_{\pm 0.68}$ | $71.98_{\pm 1.36}$ | $73.95_{\pm 1.86}$ | $66.82_{\pm 2.19}$ | $76.47_{\pm 1.24}$ | $77.45_{\pm 1.11}$ |
> > > > |          | 30 | $83.05_{\pm 0.41}$ | $73.57_{\pm 0.99}$ | $72.73_{\pm 0.76}$ | $75.32_{\pm 1.15}$ | $70.52_{\pm 0.80}$ | $77.57_{\pm 0.76}$ | $77.89_{\pm 0.06}$ |
> > > > |          | 50 | $82.66_{\pm 0.33}$ | $73.78_{\pm 1.13}$ | $72.85_{\pm 0.65}$ | $74.51_{\pm 0.82}$ | $70.69_{\pm 0.75}$ | $77.79_{\pm 0.74}$ | $77.87_{\pm 0.88}$ |
> > > > |          | 100 | $81.71_{\pm 1.19}$ | $71.47_{\pm 1.37}$ | $72.70_{\pm 0.90}$ | $74.08_{\pm 0.94}$ | $70.73_{\pm 0.79}$ | $78.25_{\pm 0.77}$ | $78.50_{\pm 0.45}$ |

---

> > > > > ### Author Response · Authors · 2023-08-21
> > > > >
> > > > > **Table 3:** Average accuracy of the proposed method on the CIFAR10 dataset over different total number of annotators and different amount of annotations received per data point.
> > > > >
> > > > > |  |  | 1 | 3 | 5 | 7 | 9 |
> > > > > | :---- | :---- | :---- | :---- | :---- | :---- | :---- |
> > > > > | IDN-LOW  | 10 | $87.34_{\pm 0.27}$ | $88.36_{\pm 0.32}$ | $88.47_{\pm 0.17}$ | $88.63_{\pm 0.43}$ | $88.75_{\pm 0.21}$ |
> > > > > |          | 30 | $86.94_{\pm 1.15}$ | $88.13_{\pm 0.31}$ | $88.63_{\pm 0.34}$ | $89.00_{\pm 0.51}$ | $89.01_{\pm 0.60}$ |
> > > > > |          | 50 | $86.78_{\pm 0.68}$ | $88.05_{\pm 0.41}$ | $88.72_{\pm 0.24}$ | $88.94_{\pm 0.43}$ | $89.21_{\pm 0.30}$ |
> > > > > | IDN-MID  | 10 | $86.58_{\pm 0.62}$ | $87.41_{\pm 0.21}$ | $88.10_{\pm 0.19}$ | $88.02_{\pm 0.31}$ | $87.93_{\pm 0.33}$ |
> > > > > |          | 30 | $85.67_{\pm 0.55}$ | $87.55_{\pm 0.52}$ | $87.88_{\pm 0.52}$ | $88.25_{\pm 0.44}$ | $88.35_{\pm 0.10}$ |
> > > > > |          | 50 | $85.68_{\pm 0.41}$ | $87.48_{\pm 0.49}$ | $87.80_{\pm 0.22}$ | $88.50_{\pm 0.30}$ | $88.53_{\pm 0.46}$ |
> > > > > | IDN-HIGH | 10 | $84.68_{\pm 0.37}$ | $86.77_{\pm 0.52}$ | $87.63_{\pm 0.43}$ | $87.68_{\pm 0.48}$ | $87.48_{\pm 0.35}$ |
> > > > > |          | 30 | $83.05_{\pm 0.41}$ | $87.11_{\pm 0.26}$ | $87.75_{\pm 0.38}$ | $87.74_{\pm 0.46}$ | $88.20_{\pm 0.20}$ |
> > > > > |          | 50 | $82.66_{\pm 0.33}$ | $87.22_{\pm 0.52}$ | $87.48_{\pm 0.34}$ | $87.92_{\pm 0.58}$ | $88.00_{\pm 0.38}$ |
> > > > >
> > > > >
> > > > >
> > > > >
> > > > > **Table 4:** Average accuracy of the inferred labels.
> > > > >
> > > > > |  |  | Ours | MBEM/EM |
> > > > > | :---- | :---- | :---- | :---- |
> > > > > | IDN-LOW  | 10 | $97.98_{\pm 0.22}$ (39417) | $80.26_{\pm 0.51}$ |
> > > > > |          | 30 | $98.01_{\pm 0.23}$ (38271) | $78.20_{\pm 1.58}$ |
> > > > > |          | 50 | $97.30_{\pm 0.59}$ (39104) | $79.37_{\pm 0.17}$ |
> > > > > |          | 100 | $97.33_{\pm 0.37}$ (38478) | $77.35_{\pm 2.68}$ |
> > > > > | IDN-MID  | 10 | $97.70_{\pm 0.42}$ (36579) | $62.81_{\pm 0.37}$ |
> > > > > |          | 30 | $96.65_{\pm 0.38}$ (37316) | $62.22_{\pm 0.19}$ |
> > > > > |          | 50 | $96.21_{\pm 0.37}$ (38110) | $61.67_{\pm 0.38}$ |
> > > > > |          | 100 | $96.65_{\pm 0.37}$ (35528) | $61.61_{\pm 0.30}$ |
> > > > > | IDN-HIGH | 10 | $95.18_{\pm 0.65}$ (37394) | $48.66_{\pm 0.37}$ |
> > > > > |          | 30 | $94.03_{\pm 0.47}$ (37040) | $49.08_{\pm 0.21}$ |
> > > > > |          | 50 | $93.51_{\pm 0.30}$ (36498) | $48.89_{\pm 0.25}$ |
> > > > > |          | 100 | $94.13_{\pm 0.42}$ (34347) | $48.85_{\pm 0.12}$ |
> > > > >
> > > > > **Reference**
> > > > >
> > > > > [A] Gao, Zhengqi, et al. ”Learning from multiple annotator noisy labels via samplewise label fusion.” European Conference on Computer Vision. Cham: Springer Nature Switzerland, 2022.
> > > > >
> > > > > [B] Khetan, Ashish, Zachary C. Lipton, and Anima Anandkumar. ”Learning from noisy singly-labeled data.” The Ninth International Conference on Learning Representations. 2018.
> > > > >
> > > > > [C] Dawid, Alexander Philip, and Allan M. Skene. ”Maximum likelihood estimation of observer error-rates using the EM algorithm.” Journal of the Royal Statistical Society: Series C (Applied Statistics) 28.1 (1979): 20-28.
> > > > >
> > > > > [D] Han, Bo, et al. "Co-teaching: Robust training of deep neural networks with extremely noisy labels." Advances in neural information processing systems 31 (2018).
> > > > >
> > > > > [E] Yu, Xingrui, et al. "How does disagreement help generalization against label corruption?." International Conference on Machine Learning. PMLR, 2019.

---

### Author Rebuttal · Authors · 2023-08-10

Dear reviewers,

Thank you for providing feedback! We appreciate your comments and the time and effort you took to read, comprehend, and assess our paper. We value your comments and concerns regarding how to emphasize our motivations and clarify the assumptions more clearly (Reviewer i77Y, PGxM, wDkZ), as well as your suggestions that we compare with more baseline methods to highlight the advantages of the proposed method (Reviewer i77Y, PGxM, uiTb, ysQF). We carefully reviewed every one of your queries, concerns, and remarks. Here we addressed each review separately with a thorough response. In order to reflect our responses, we have also uploaded a one-page PDF with additional experimental results. The baseline methods used in the experiments are listed below.

We hope that our responses have adequately addressed all of the concerns raised. However, if further details, justifications, or results are needed, we would be pleased to provide them.



[1] Gao, Zhengqi, et al. Learning from multiple annotator noisy labels via sample-wise label fusion. European Conference on Computer Vision. Cham: Springer Nature Switzerland, 2022.

[2] Xia, Xiaobo, et al. Part-dependent label noise: Towards instance-dependent label noise. Advances in Neural Information Processing Systems 33 (2020): 7597-7610.

[3] Yang, Shuo, et al. Estimating instance-dependent bayes-label transition matrix using a deep neural network. International Conference on Machine Learning. PMLR, 2022.

[4] Dawid, Alexander Philip, and Allan M. Skene. Maximum likelihood estimation of observer error‐rates using the EM algorithm. Journal of the Royal Statistical Society: Series C (Applied Statistics) 28.1 (1979): 20-28.

[5] Khetan, Ashish, Zachary C. Lipton, and Anima Anandkumar. Learning from noisy singly-labeled data. The Ninth International Conference on Learning Representations. 2018.

[6] Li, Yuan, Benjamin Rubinstein, and Trevor Cohn. Exploiting worker correlation for label aggregation in crowdsourcing. International conference on machine learning. PMLR, 2019.

[7] Li, Xuefeng, et al. Provably end-to-end label-noise learning without anchor points. International conference on machine learning. PMLR, 2021.

[8] Ibrahim, Shahana, Tri Nguyen, and Xiao Fu. Deep learning from crowdsourced labels: Coupled cross-entropy minimization, identifiability, and regularization. ICLR, 2023.

---

### Comment · Area_Chair_Ynv3 · 2023-08-18
**Reviewer-Author Discussion Period**

Dear All,

Thank you reviewers for your hard work in evaluating this submission, and thank you authors for responding to the reviewers’ questions and concerns.

We are now entering the final phase of the discussion period, which will run until 21 Aug, and some of the authors' responses have to been acknowledged by all reviewers.

@Reviewers, if you have any follow up questions or comments on the rebuttal or the responses, now is the time to express them. At the very least, please acknowledge that you have read the authors’ response to your review.

Thank you everyone for making the review process a fruitful, constructive, and civil process.

AC

---

### Decision · Program_Chairs · 2023-09-21

**Decision:**

Accept (poster)

**Comment:**

This paper introduces a method for modelling input-dependent label noise for classification in the presence of multiple annotators. The generative process of annotation is modelled as a hierarchical Bayesian model in which the observed labels are noisy versions of the latent true labels that are corrupted through a noise transition model that is dependent on the input as well as the annotator. The authors propose a method for inferring the true labels based on the approximated noise model and demonstrate benefits on classification benchmarks with both synthetic and real annotation noise.

Specifically, the idea of modelling labeling noise as a function of both input and annotator is much more realistic than the majority of related works which drop the input dependence or ignore the differences between annotators. They propose a practical algorithm with demonstrable efficacy on classification benchmarks with both synthetic & real label noise. The authors provide a sound theoretical justification for the above algorithm: namely the concentration theorem shows the "tightness" of the approximation of the noise transition model; and formulating inference of true labels as a statistical test gives an information-theoretic bound on the error. However, there still have some rooms to be improved. For example, the relaxed condition and its analysis still need to be largely improved; the description of the anchor-point collection method and the transition-model learning process should be improved; and the roles of different modules were not fully decoupled in the ablation experiments.

Although the reviewers had some concerns and follow-up questions, the authors did a point-to-point rebuttal in details. The score of reviewer i77Y should have been increased after the rebuttal (but it wasn't). Meanwhile, the clarity and novelty are above the bar of NeurIPS. Therefore, this paper can be accepted as a poster.